# Optimal Batched (Generalized) Linear Contextual Bandit Algorithm

## Abstract

We study *batched* linear and generalized linear contextual bandits and introduce practical batched algorithms, aiming for methods that are both *practical* and *provably* optimal under limited adaptivity. For linear contextual bandits, we propose the first algorithm that attains minimax-optimal regret (up to polylogarithmic factors in $T$) in both small-$K$ and large-$K$ regimes using only $\mathcal{O}(\log \log T)$ batches, while our second algorithm removes the G-optimal design step—the dominant computational bottleneck—yet preserves the same order of statistical guarantees and achieves the lowest known runtime complexity. We then adapt to the generalized linear contextual bandits and design an algorithm that is fully free of curvature parameter $\kappa$: neither the algorithm requires knowledge of nor its regret bound depends on $\kappa$, and it retains $\mathcal{O}(\log \log T)$ batch complexity with near-optimal regret. Collectively, these results deliver the first batched linear contextual methods that are simultaneously minimax-optimal across all regimes and computationally efficient, and the first generalized linear method that is both statistically and computationally efficient while remaining fully $\kappa$-independent.

## 1 Introduction

Stochastic *linear contextual bandits* are a cornerstone of sequential decision-making, where an agent repeatedly selects actions from a time-varying, feature-based set and observes rewards generated by an unknown linear model (Abe & Long, 1999; Auer, 2002; Abe et al., 2003; Dani et al., 2008; Li et al., 2010; Chu et al., 2011; Abbasi-Yadkori et al., 2011; Li et al., 2019; Lattimore & Szepesvári, 2020; Kirschner et al., 2021). The resulting low-dimensional structure enables efficient generalization and has seen wide application, from recommender systems (Li et al., 2010) and inventory control (Jin et al., 2021a) to clinical trials and precision medicine (Lu et al., 2021).

In practice, however, *fully adaptive* algorithms—which update policies every round—are often infeasible due to computational or operational constraints. This motivates the study of *batched* linear contextual bandits, where updates occur at a small number of batch endpoints (Abbasi-Yadkori et al., 2011; Ruan et al., 2021; Hanna et al., 2023a;b; Zhang et al., 2025). While recent algorithms achieve strong theoretical guarantees with as few as $\mathcal{O}(\log \log T)$ batches, many still rely on computationally expensive G-optimal design or update more frequently than desirable, limiting their practical efficiency (see Table 3 and Figure 1). Recently, Yu & Oh (2025) proposed a computationally efficient batched algorithm to achieve minimax-optimal regret. However, their method is only applicable to non-contextual, fixed feature settings.

Beyond linear models, *generalized linear contextual bandits* allows expected rewards to follow a nonlinear link function (e.g., logistic or Poisson). This broadens applicability but introduces new challenges: existing analyses typically depend on instance dependent curvature parameters such as $\kappa$ (defined in Section 5.2), and a prior batched algorithm for generalized linear contextual bandits (Sawarni et al., 2024) require knowledge of $\kappa$, while still incurring loose regret bounds and high computational cost (see Table 2). In their method, the first batch size scales as $\kappa^{1/3}$, which can be prohibitively large in saturated regimes where $\kappa$ is large. Thus, beyond the difficulty of requiring prior knowledge of $\kappa$, developing algorithms that are entirely $\kappa$-free is a central challenge.

Consequently, practical algorithms with provable regret guarantees for both linear and generalized linear contextual bandits remain elusive. This shortfall is also evident in our numerical experiments, where existing methods underperform (see Section 6). This motivates the following open research:

- For *linear contextual bandits*, can we design batched algorithms that achieve minimax-optimal regret in both small-$K$ and large-$K$ regimes with the minimal batch complexity? Can we also design a computationally more efficient algorithm avoiding G-optimal design procedure while still maintaining the minimax optimality in regret?

- As an extension, can we adapt the computationally efficient batched algorithm to *generalized linear* contextual bandits (hence, again not relying on G-optimal design) and show near-optimal regret, entirely free of $\kappa$ dependence, while still maintaining the minimal batch complexity?

Positive answers to these questions would unify theory and practice in linear and generalized linear contextual bandits, leading to algorithms that remain statistically optimal and computationally efficient under limited adaptivity. Our main contributions are summarized as follows:

- **Tightest regret bounds for batched linear contextual bandits.** We introduce BLCE-G, which combines near G-optimal design and arm elimination. It achieves the worst-case regret bound $\mathcal{O}(\sqrt{dT}(\sqrt{\log(KT)} \wedge \sqrt{d + \log T})\sqrt{\log d \log \log T})$, where $K$ is the number of arms, $d$ is the feature dimension, and $T$ is the horizon. This is the tightest known bound for batched linear contextual bandits. BLCE-G is the first algorithm to simultaneously match the minimax lower bounds $\Omega(d\sqrt{T})$ in the large-$K$ regime ($K \geq \Omega(e^d)$) and $\Omega(\sqrt{dT \log K})$ in the small-$K$ regime ($K \leq \mathcal{O}(e^d)$), up to logarithmic factors.

- **First minimax-optimal algorithm without G-optimal design.** We propose BLCE, which replaces the G-optimal design step with uncertainty-driven exploration combined with arm elimination. It still achieves the minimax-optimal regret with the lowest total time complexity $\mathcal{O}(Kd^2T \log \log T)$. To our knowledge, BLCE is the first theoretically optimal batched linear contextual bandit algorithm that avoids G-optimal design, which is the main computational bottleneck of the existing batched algorithms (Ruan et al., 2021; Hanna et al., 2023a;b; Zhang et al., 2025). Its guarantees also extend beyond conventional i.i.d. contexts, as discussed in Remark 1.

- **First $\kappa$-independent algorithm for generalized linear contextual bandits.** We develop BGLE, which extends BLCE to generalized linear contextual bandits. It achieves the worst-case regret bound $\widetilde{\mathcal{O}}(dRS\sqrt{T}/\sqrt{\hat{\kappa}}) + \widetilde{\mathcal{O}}((R^2 Se^{8RS}d^2 + R)T^{1/3})$, where $R$ is the upper-bound on rewards, $S$ is the norm-bound of the parameter $\|\theta^*\|_2 \leq S$, and $\hat{\kappa}$ is the expected inverse curvature at the optimal arm. This is the tightest known bound for batched generalized linear contextual bandits (Sawarni et al., 2024), and uniquely, it is entirely independent of $\kappa$ in both leading and transient terms. Here, $\kappa$ measures worst-case curvature and can diverge in saturated regimes, while $\hat{\kappa}$ reflects average curvature around the optimal arm. Unlike prior work (Sawarni et al., 2024), our algorithm requires no prior knowledge of $\kappa$ and inherits the efficiency of BLCE.

- **Batch Complexity.** While achieving favorable regret guarantees, all of our proposed algorithms only require the minimal batch complexity of $\mathcal{O}(\log \log T)$.

- **Practical Superiority.** Our experiments demonstrate that BLCE-G, BLCE, and BGLE consistently outperform prior batched linear and generalized linear contextual bandit algorithms across various instances, combining provable efficiency with strong empirical performance and substantially reduced runtime overhead.

## 2 RELATED WORK

A substantial literature on *batched bandits* spans from multi-armed to linear (contextual) models. Early work established near-optimal learning with few policy updates in the multi-armed setting (Perchet et al., 2016; Gao et al., 2019; Jin et al., 2021b;c), later extended to linear bandits under Gaussian-type features (Han et al., 2020) and adversarial features (Esfandiari et al., 2021), culminating in algorithms that achieve near-optimal regret with the minimal batch complexity $\mathcal{O}(\log \log T)$ (Ren et al., 2024; Yu & Oh, 2025). Although Yu & Oh (2025) attain minimax-optimal regret in both regimes, their analysis is restricted to non-contextual batched bandits and does not extend to the linear contextual setting. In linear *contextual* bandits, recent algorithms (Ruan et al.,

Table 1: Worst-case regret, batch complexity, and time complexity comparison in batched linear contextual bandits. **Both BLCE-G and BLCE achieve minimax-optimal regret**, matching the minimax lower bounds $\Omega(\sqrt{dT\log K} \wedge d\sqrt{T})$ (Dani et al., 2008; He et al., 2022) across all regimes while requiring only $\mathcal{O}(\log\log T)$ batches. Among existing approaches, **BLCE-G attains the tightest regret bound, whereas BLCE achieves the lowest time complexity**. Note that $\mathcal{T}_{\text{opt}}$ is the cost of one call to the linear optimization oracle, and Ruan et al. (2021); Hanna et al. (2023a;b); Zhang et al. (2025) do not use within-batch context information in their algorithms.

| Paper | Worst-Case Regret | Batches | Time Complexity |
|---|---|---|---|
| Abbasi-Yadkori et al. (2011) | $\mathcal{O}(d\sqrt{T}\log T)$ | $\mathcal{O}(d\log T)$ | $\mathcal{O}((Kd+d^2)T + Kd^3\log T)$ |
| Ruan et al. (2021) | $\mathcal{O}(\sqrt{dT\log(dKT)\log d}\log\log T)$ | $\mathcal{O}(\log\log T)$ | $\mathcal{O}(Kd^4 T(\log T + \log d))$ |
| Hanna et al. (2023b) | $\mathcal{O}(d\sqrt{T\log T}\log\log T)$ | $\mathcal{O}(\log\log T)$ | $\Omega(T^d)$ |
| Hanna et al. (2023a) | $\mathcal{O}(d^{3/2}\sqrt{T}\log T\log\log T)$ | $\mathcal{O}(\log\log T)$ | $\mathcal{O}(d^3\mathcal{T}_{\text{opt}}T\log d\log^3 T\log\log T)$ $+ \mathcal{O}(d^4\log d\log^3 T\log\log T)$ |
| Zhang et al. (2025) | $\mathcal{O}(\sqrt{dT\log(dKT)\log T}\,\log(dT)\log\log T)$ | $\mathcal{O}(\log\log T)$ | $\mathcal{O}(Kd^2 T\log\log T)$ $+\mathcal{O}(Kd^{7/2}\sqrt{T\log(dKT)\log T})$ |
| Algorithm 1 | $\min \begin{cases} \mathcal{O}(\sqrt{dT\log(KT)\log d}\log\log T) \\ \mathcal{O}(\sqrt{d(d+\log T)T}\log d\log\log T) \end{cases}$ | $\mathcal{O}(\log\log T)$ | $\mathcal{O}(Kd^2 T(d+\log\log T))$ |
| Algorithm 2 | $\min \begin{cases} \mathcal{O}(\sqrt{dT\log(KT)\log T}\log\log T) \\ \mathcal{O}(\sqrt{d(d+\log T)T}\log T\log\log T) \end{cases}$ | $\mathcal{O}(\log\log T)$ | $\mathcal{O}(Kd^2 T\log T)$ |

Table 2: Worst-case regret, batch complexity, and time complexity comparison in batched generalized linear contextual bandits. **BGLE attains the tightest regret bound** with only $\mathcal{O}(\log\log T)$ batches, is entirely $\kappa$-free in the regret bound, and attains the lowest time complexity.[1]

| Paper | Worst-Case Regret | Batches | Time Complexity |
|---|---|---|---|
| Sawarni et al. (2024) | $\mathcal{O}\big((RSd(\sqrt{d/\hat{\kappa}} \wedge \sqrt{R_{\dot{\mu}}\log d})\sqrt{\log T}\log\log T + R)\sqrt{T}\big)$ $+ \mathcal{O}\big((\kappa R_{\dot{\mu}}R^5 S^2)^{1/3}e^{2RS}d^2(\log T)^{2/3}\log\log T \cdot T^{\frac{1}{3}}\big)$ | $\mathcal{O}(\log\log T)$ | $\mathcal{O}(Kd^4 T\log T + \mathcal{C}_{\text{opt}}^{\text{tot}})$ $+ \mathcal{O}(Kd^5(\kappa R_{\dot{\mu}})^{1/3}e^{2RS}$ $(R^2 S^2 T\log^2 T)^{1/3})$ |
| Algorithm 3 | $\mathcal{O}\big(RS\sqrt{d(d+\log T)\log T\log\log T/\hat{\kappa}} \cdot \sqrt{T}\big)$ $+ \mathcal{O}\big((R^2 Se^{8RS}d(d+\log T)\log T\log\log T + \frac{R}{\log\log T})T^{\frac{1}{3}}\big)$ | $\mathcal{O}(\log\log T)$ | $\mathcal{O}(Kd^2 T\log\log T)$ $+ \mathcal{O}(\mathcal{C}_{\text{opt}}^{\text{tot}})$ |

2021; Hanna et al., 2023b; Zhang et al., 2025) achieve minimax-optimal regret under i.i.d. contexts with only $\mathcal{O}(\log\log T)$ batches, but rely on G-optimal design, ignore within-batch contexts, and achieve optimality in only one regime. Abbasi-Yadkori et al. (2011) require $\mathcal{O}(\log T)$ batches, while Hanna et al. (2023a) improve efficiency but obtain a suboptimal regret bound $\widetilde{\mathcal{O}}(d^{3/2}\sqrt{T})$. In contrast, our algorithm leverages within-batch contexts while maintaining both computational and statistical optimality, as discussed in Appendix E.

Batched generalized linear contextual bandits have been studied more recently. Sawarni et al. (2024) obtain $\widetilde{\mathcal{O}}(\sqrt{T})$ regret with $\mathcal{O}(\log\log T)$ batches but require prior knowledge of the parameter $\kappa$, which characterizes the worst-case curvature of the link function. Large values of $\kappa$ degrade the performance of UCB-based elimination, and in their method the first batch size scales as $\kappa^{1/3}$, which can be prohibitively large in saturated regimes, causing the algorithm to spend much of the horizon with no informative updates. Moreover, their regret bound is loose due to the leading term $\sqrt{d/\hat{\kappa}}$, preventing the optimal $\widetilde{\mathcal{O}}(d\sqrt{T})$ rate. Their reliance on the G-optimal based method of Ruan et al. (2021) also incurs significant runtime overhead.

In contrast, our proposed BGLE builds on BLCE, ensuring computational efficiency. Crucially, it removes any dependence on $\kappa$ in both leading and transient terms, eliminates the extraneous factor $d$ in $\sqrt{d/\hat{\kappa}}$, and achieves the optimal regret bound $\widetilde{\mathcal{O}}(d\sqrt{T})$.

---

[1] $\mathcal{C}_{\text{opt}}^{\text{tot}}$ denotes the total oracle cost of solving the log-loss minimization at batch boundaries. With $B$ batches, $\mathcal{C}_{\text{opt}}^{\text{tot}} = \sum_{\ell=1}^{B} \mathcal{C}_{\text{opt}}(\mathcal{T}_{\ell} - \mathcal{T}_{\ell-1}, d)$, where $\mathcal{C}_{\text{opt}}(n,d)$ is the cost of computing the unconstrained MLE from $n$ samples in $d$ dimensions.

## 3 PRELIMINARIES

### 3.1 NOTATIONS

For a set, $|\cdot|$ denotes its cardinality. For $x \in \mathbb{R}^d$, $\|x\|_2$ is the Euclidean norm, and for positive definite $H$, $\|x\|_H := \sqrt{x^\top H x}$. For a matrix, $\mathrm{tr}(\cdot)$ and $\det(\cdot)$ denote its trace and determinant. For $n \in \mathbb{N}$, we write $[n] := \{1, \ldots, n\}$, and use $\wedge$ for the minimum operator. For symmetric matrices $A, B$ of the same dimension, $A \preceq B$ (resp. $A \succeq B$) means $B - A$ (resp. $A - B$) is positive semidefinite. The indicator $\mathbb{1}_{\{E\}}$ equals 1 if the event $E$ occurs and 0 otherwise. Finally, the natural filtration is $\mathcal{F}_t := \sigma(\mathcal{A}_1, x_{1,a_1}, r_1, \ldots, \mathcal{A}_t, x_{t,a_t}, r_t)$ with $\mathcal{F}_0$ trivial.

### 3.2 PROBLEM SETTING: BATCHED LINEAR CONTEXTUAL BANDITS

We study the stochastic linear contextual bandit problem. At each round $t \in [T]$, the agent observes contextual features of $K$ arms $\mathcal{A}_t := \{x_{t,1}, \ldots, x_{t,K}\} \subseteq \mathbb{R}^d$ and selects one arm $x_{t,a_t} \in \mathcal{A}_t$, receiving reward $r_t = \langle x_{t,a_t}, \theta^* \rangle + \eta_t$, where $\theta^* \in \mathbb{R}^d$ is unknown and $\eta_t$ is independent $\sigma$-subgaussian noise. Following the convention in the batched contextual bandit literature (Ruan et al., 2021; Hanna et al., 2023b; Zhang et al., 2025), we assume that each feature set $\mathcal{A}_t$ is drawn i.i.d. from an unknown distribution $\mathcal{D}$ across rounds, while allowing arbitrary correlation among features presented within the same round. (In Remark 1, we discuss how our algorithm Algorithm 2 and its analysis further relax this i.i.d. assumption.)

The performance of the agent is measured by the *cumulative expected regret* defined as $\mathcal{R}(T) := \mathbb{E}\left[\sum_{t=1}^{T}\left(\langle x_t^*, \theta^* \rangle - \langle x_{t,a_t}, \theta^* \rangle\right)\right]$, where $x_t^* \in \mathrm{argmax}_{x \in \mathcal{A}_t} \langle x, \theta^* \rangle$ is the optimal arm. We make the following standard assumptions:

**Assumption 1.** $\|x\|_2 \leq 1$ *for all $x \in \mathcal{A}_t$ and $\|\theta^*\|_2 \leq 1$.*

**Assumption 2.** *The noise $\eta_t$ is a 1-subgaussian random variable for all $t \in [T]$.*

In the *batched* setting, the horizon $[T]$ is partitioned into $B$ disjoint batches,

$$\{1, \ldots, T\} = \underbrace{[\mathcal{T}_0 + 1, \ldots, \mathcal{T}_1]}_{\text{batch } 1} \cup \cdots \cup \underbrace{[\mathcal{T}_{B-1} + 1, \ldots, \mathcal{T}_B]}_{\text{batch } B} .$$

At time $t$, the agent may only use the reward information from all previous batches (while the contexts up to and including time $t$ are always observable). In particular, the rewards for all rounds within a batch become available only at the end of that batch.

### 3.3 PROBLEM SETTING: BATCHED GENERALIZED LINEAR CONTEXTUAL BANDITS

We next consider generalized linear contextual bandits, where rewards follow a one–parameter exponential family distribution. Given an arm $x \in \mathbb{R}^d$ and the unknown parameter $\theta^* \in \mathbb{R}^d$, the reward $r$ has density $p(r \,|\, x; \theta^*) = \exp\left(r\langle x, \theta^* \rangle - m(\langle x, \theta^* \rangle) + h(r)\right)\nu(\mathrm{d}r)$, with log-partition function $m$, base measure $\nu$, and link function $\mu(z) := m'(z)$. We adopt the following standard assumptions:

**Assumption 3.** $\|x\|_2 \leq 1$ *for all $x \in \mathcal{A}_t$, and $\|\theta^*\|_2 \leq S$ for a known constant $S > 0$.*

**Assumption 4.** *The log-partition function $m$ is convex and three times differentiable. Equivalently, $\dot{\mu} := m'' \geq 0$ and $m'''$ exists.*

At each round $t \in [T]$, the learner observes an arm set $\mathcal{A}_t = \{x_{t,1}, \ldots, x_{t,K}\} \subseteq \mathbb{R}^d$ and selects $x_{t,a_t} \in \mathcal{A}_t$. The reward $r_t$ is drawn from $p(r \,|\, x_{t,a_t}; \theta^*)$ with natural parameter $\langle x_{t,a_t}, \theta^* \rangle$, and satisfies $\mathbb{E}[r \,|\, x; \theta^*] = \mu(\langle x, \theta^* \rangle)$. The performance is measured by the *cumulative expected regret* $\mathcal{R}(T) = \mathbb{E}\left[\sum_{t=1}^{T}(\mu(\langle x_t^*, \theta^* \rangle) - \mu(\langle x_{t,a_t}, \theta^* \rangle))\right]$, where $x_t^* \in \mathrm{argmax}_{x \in \mathcal{A}_t} \mu(\langle x, \theta^* \rangle)$ is the optimal arm. Following Sawarni et al. (2024), we assume i.i.d. contexts drawn from an unknown distribution $\mathcal{D}$ and rewards almost surely supported on $[0, R]$, which implies that link function satisfies $|\ddot{\mu}(z)| \leq R\dot{\mu}(z)$ for all $z \in \mathbb{R}$. This self-concordance property of GLMs is crucial for our analysis.

## 4 BATCHED LINEAR CONTEXTUAL BANDIT ALGORITHMS

### 4.1 NOTE ON BATCH SCHEDULE

Three types of batch schedules are commonly considered for algorithms in the literature: the *static grid*, where batch boundaries are fixed in advance; the *adaptive grid*, where batch sizes are chosen adaptively at the beginning of each batch; and the *rare policy switch* schedule studied by Abbasi-Yadkori et al. (2011), which allows arbitrary changes subject to a limit on the total number of switches. Similar to Ruan et al. (2021); Hanna et al. (2023a;b); Zhang et al. (2025), we adopt the *static grid* for our algorithms.

### 4.2 PROPOSED ALGORITHMS

We propose two batched algorithms `BLCE-G` and `BLCE` for linear contextual bandits. Let us start with explanation on our first algorithm, `BLCE-G`, which stands for the *Batched Linear Contextual Bandit with Elimination and G-optmial design*, whose pseudocode is given in Algorithm 1. In the first batch, rounds are divided into two phases with ratio $c : (1 - c)$. During the first $c$-fraction, arms are sampled according to a near G-optimal design over $\mathcal{A}_t$ (Line 5), while in the remaining $(1 - c)$-fraction the algorithm selects the most informative direction with respect to the current Gram matrix (Line 7). To reduce computational and runtime cost, we adopt a relaxation of the G-optimal design, namely the near G-optimal design, which loosens the bound by at most a factor of two. Formally, for any arm set $X \subset \mathbb{R}^d$, there exists a design distribution $\mathcal{K}_X$ supported on $X$ such that

$$\max_{x \in X} x^\top (\mathbb{E}_{z \sim \mathcal{K}_X}[zz^\top])^{-1} x \le 2d \ .$$

As shown in Corollary 4, such a design can be computed in time $\mathcal{O}(Kd^3)$. For fair comparison, we also account for this cost when evaluating other algorithms that rely on G-optimal design (Ruan et al., 2021; Hanna et al., 2023b; Zhang et al., 2025). After each arm pull, the inverse Gram matrix is updated using the Sherman-Morrison formula (reducing the cost from $\mathcal{O}(d^3)$ to $\mathcal{O}(d^2)$) (Line 8). At the end of the first batch, we set $V_1 := H_{\mathcal{T}_1}$, compute the regression estimate $\hat{\theta}_1$, and reinitialize $H_{\mathcal{T}_1}$ for the next batch (Line 9).

For any batch $\ell \ge 2$, the algorithm eliminates suboptimal arms $\ell - 1$ times using the past estimates $\hat{\theta}_1, \ldots, \hat{\theta}_{\ell-1}$, yielding a nested sequence of feasible sets $\mathcal{A}_t^{(1)}, \ldots, \mathcal{A}_t^{(\ell-1)}$ (Line 13). The elimination threshold $\varepsilon_{t,k}$ for $k \in [\ell - 1]$ is defined as

$$\max_{y \in \mathcal{A}_t^{(k-1)}} \|y\|_{V_k^{-1}} \left( \sqrt{2 \log\left(|\mathcal{A}_t^{(k-1)}|(B-1)T^2\right)} + \sqrt{\lambda} \wedge 2\sqrt{\log\left(\frac{2^{6d-5}\pi d(B-1)^2}{15^{d-1}/T^2}\right)} + 2\sqrt{\lambda} \right) .$$

Within batch $\ell \ge 2$, the rounds are partitioned in the ratio $c^2 : c(1 - c) : (1 - c)$. In the first $c^2$-fraction, arms are sampled according to a near G-optimal design over $\mathcal{A}_t^{(\ell-1)}$ (Line 15). In the next $c(1 - c)$-fraction, the algorithm selects the most informative direction relative to the Gram matrix (Line 17). In the final $(1 - c)$-fraction, arms are chosen greedily with respect to the latest estimate (Line 19). As before, after each pull the Gram matrix is updated (Line 20), and at the end of the batch we set $V_\ell := H_{\mathcal{T}_\ell}$, compute $\hat{\theta}_\ell$, and reinitialize $H_{\mathcal{T}_\ell}$ for the next batch (Line 21).

Now, let us introduce our second algorithm, `BLCE`, which stands for *Batched Linear Contextual Bandit with Elimination*. The pseudocode of `BLCE` is given in Algorithm 2. Relative to Algorithm 1, `BLCE` eliminates the near G-optimal design segment and instead lengthens the uncertainty-driven exploration phase to occupy those rounds. To our knowledge, `BLCE` is the first batched linear contextual bandit algorithm that achieves theoretical optimality without relying on G-optimal design, which has traditionally been regarded as essential (Ruan et al., 2021; Hanna et al., 2023b; Zhang et al., 2025). Notably, both `BLCE-G` and `BLCE` avoid enforcing any fixed choice of $c \in (0, 1]$, thereby providing theoretical guarantees together with practical flexibility in balancing exploration and exploitation.

### 4.3 REGRET ANALYSIS FOR BATCHED LINEAR CONTEXTUAL BANDITS

**Theorem 1** (Regret of `BLCE-G`). *Consider running the `BLCE-G` algorithm for $T$ rounds with $K$ arms in $d$ dimensions. The worst-case cumulative regret satisfies*

$$\mathcal{R}(T) = \mathcal{O}\left(\sqrt{dT}(\sqrt{\log(KT)} \wedge \sqrt{d + \log T})\sqrt{\log d \log \log T}\right) = \widetilde{\mathcal{O}}\left(\sqrt{dT \log K} \wedge d\sqrt{T}\right) .$$

---

**Algorithm 1** `BLCE-G`

---

1: **Input:** Horizon $T$; batch end times $\mathcal{T}_1 = \left\lceil \frac{\sqrt{T}}{\log_2 \log_2 T} \right\rceil + 1$, $\mathcal{T}_\ell = \left( \mathcal{T}_{\ell-1} + \left\lceil \frac{T^{1-2^{-\ell}}}{\log_2 \log_2 T} \right\rceil + 2 \right) \wedge T$
     for $\ell \geq 2$; number of batches $B$, with $\mathcal{T}_B = T$; within-batch allocation rate $c$

---

2: **Initialize:** $\lambda \leftarrow \log(dT)$, $H_0 \leftarrow \lambda I$;
3: **for** $t \leftarrow 1, 2, \ldots, \mathcal{T}_1$ **do**
4:     **if** $t \leq \left\lceil c\sqrt{T}/\log_2 \log_2 T \right\rceil$ **then**
5:        Pull arm $x_{t,a_t} \sim \pi_{G'}(\mathcal{A}_t)$;
6:     **else**
7:        Pull arm $x_{t,a_t} \in \arg\max_{x \in \mathcal{A}_t} \|x\|_{H_{t-1}^{-1}}$;
8:     $H_t^{-1} \leftarrow H_{t-1}^{-1} - H_{t-1}^{-1} x_{t,a_t} x_{t,a_t}^\top H_{t-1}^{-1}/(1 + x_{t,a_t}^\top H_{t-1}^{-1} x_{t,a_t})$;
9: $V_1^{-1} \leftarrow H_{\mathcal{T}_1}^{-1}, \hat{\theta}_1 \leftarrow V_1^{-1} \sum_{t=1}^{\mathcal{T}_1} r_t x_{t,a_t}, H_{\mathcal{T}_1} \leftarrow \lambda I$;
10: **for** $\ell \leftarrow 2, \ldots, B$ **do**
11:     **for** $t \leftarrow \mathcal{T}_{\ell-1} + 1, \ldots, \mathcal{T}_\ell$ **do**
12:        **for** $k \leftarrow 1, \ldots, \ell - 1$ **do**
13:           $x_t^{(k)} \leftarrow \arg\max_{x \in \mathcal{A}_t^{(k-1)}} \langle x, \hat{\theta}_k \rangle, \mathcal{A}_t^{(k)} \leftarrow \left\{ x \in \mathcal{A}_t^{(k-1)} \,\middle|\, \langle \hat{\theta}_k, x_t^{(k)} - x \rangle \leq 2\varepsilon_{t,k} \right\}$;
14:        **if** $t \leq \mathcal{T}_{\ell-1} + \left\lceil c^2 T^{1-2^{-\ell}}/\log_2 \log_2 T \right\rceil$ **then**
15:           Pull arm $x_{t,a_t} \sim \pi_{G'}(\mathcal{A}_t^{(\ell-1)})$, and receive reward $r_t$;
16:        **else if** $t \leq \mathcal{T}_{\ell-1} + \left\lceil c^2 T^{1-2^{-\ell}}/\log_2 \log_2 T \right\rceil + \left\lceil c(1-c)T^{1-2^{-\ell}}/\log_2 \log_2 T \right\rceil$ **then**
17:           Pull arm $x_{t,a_t} \in \arg\max_{x \in \mathcal{A}_t^{(\ell-1)}} \|x\|_{H_{t-1}^{-1}}$;
18:        **else**
19:           Pull arm $x_{t,a_t} \in \arg\max_{x \in \mathcal{A}_t^{(\ell-1)}} \langle x, \hat{\theta}_{\ell-1} \rangle$;
20:        $H_t^{-1} \leftarrow H_{t-1}^{-1} - H_{t-1}^{-1} x_{t,a_t} x_{t,a_t}^\top H_{t-1}^{-1}/(1 + x_{t,a_t}^\top H_{t-1}^{-1} x_{t,a_t})$;
21:     $V_\ell^{-1} \leftarrow H_{\mathcal{T}_\ell}^{-1}, \hat{\theta}_\ell \leftarrow V_\ell^{-1} \sum_{t=\mathcal{T}_{\ell-1}+1}^{\mathcal{T}_\ell} r_t x_{t,a_t}, H_{\mathcal{T}_\ell} \leftarrow \lambda I$;

---

**Discussion of Theorem 1.** Theorem 1 shows that `BLCE-G` achieves minimax-optimal regret for fully adaptive linear contextual bandits with only $\mathcal{O}(\log \log T)$ batches, the lowest attainable batch complexity. A notable feature of this bound is that it covers both regimes. In the small-$K$ regime ($K \leq \mathcal{O}(e^d)$), the regret scales as $\widetilde{\mathcal{O}}(\sqrt{dT \log K})$, and in the large-$K$ regime ($K \geq \Omega(e^d)$), as $\widetilde{\mathcal{O}}(d\sqrt{T})$. Thus, `BLCE-G` provides *the tightest known performance guarantees for batched linear contextual bandits* and, to our knowledge, is the first algorithm to match the minimax lower bounds in both regimes. Moreover, it achieves the smallest regret bound within each regime among existing works (Abbasi-Yadkori et al., 2011; Ruan et al., 2021; Hanna et al., 2023a;b; Zhang et al., 2025).

**Theorem 2** (Regret of BLCE). *Consider running the BLCE algorithm for $T$ rounds with $K$ arms in $d$ dimensions. The worst-case cumulative regret satisfies*

$$\mathcal{R}(T) = \mathcal{O}\left( \sqrt{dT}(\sqrt{\log(KT)} \wedge \sqrt{d + \log T})\sqrt{\log T \log \log T} \right) = \widetilde{\mathcal{O}}\left( \sqrt{dT \log K} \wedge d\sqrt{T} \right).$$

**Discussion of Theorem 2.** Theorem 2 establishes that `BLCE` also achieves the minimax-optimal regret bound with only $\mathcal{O}(\log \log T)$ batches. Its key distinction lies in computational: by removing the G-optimal design step, `BLCE` significantly reduces both complexity and runtime yet retains the best-known regret guarantees. As shown in Table 3, this makes `BLCE` the first batched linear contextual bandit algorithm to combine minimax optimality across both regimes with *no reliance on G-optimal design*. Moreover, in the large-$K$ regime, `BLCE` attains the smallest regret bound among existing approaches (Abbasi-Yadkori et al., 2011; Hanna et al., 2023a;b).

**Remark 1.** *While prior work on batched linear contextual bandits typically assumes i.i.d. contexts (Ruan et al., 2021; Hanna et al., 2023a;b; Zhang et al., 2025), we show that this assumption can be relaxed to the following batch-wise conditions (for any $\ell \geq 1$)*

*(1)* $\mathrm{Law}(\mathcal{A}_t | \mathcal{F}_{\mathcal{T}_{\ell-1}}) \sim \mathcal{D}_{\ell-1}$ *for* $t \in [\mathcal{T}_{\ell-1} + 1, \mathcal{T}_{\ell+1}]$,
*(2)* $\mathcal{A}_t \perp\!\!\!\perp \{\mathcal{A}_s, x_{s,a_s}, r_s\}_{s=\mathcal{T}_{\ell-1}+1}^{\mathcal{T}_\ell} | \mathcal{F}_{\mathcal{T}_{\ell-1}}$ *for* $t \in [\mathcal{T}_\ell + 1, \mathcal{T}_{\ell+1}]$,

---

**Algorithm 2** BLCE

---

1: **Input:** Horizon $T$; batch end times $\mathcal{T}_1 = \left\lceil \frac{\sqrt{T}}{\log_2 \log_2 T} \right\rceil$, $\mathcal{T}_\ell = \left( \mathcal{T}_{\ell-1} + \left\lceil \frac{T^{1-2^{-\ell}}}{\log_2 \log_2 T} \right\rceil + 1 \right) \wedge T$

    for $\ell \geq 2$; number of batches $B$, with $\mathcal{T}_B = T$; within-batch allocation rate $c$

---

2: **Initialize:** $\lambda \leftarrow 1, H_0 \leftarrow \lambda I, b_0 \leftarrow \mathbf{0}$;
3: **for** $t \leftarrow 1, 2, \ldots, \mathcal{T}_1$ **do**
4:     Pull arm $x_{t,a_t} \in \arg\max_{x \in \mathcal{A}_t} \|x\|_{H_{t-1}^{-1}}$;
5:     $H_t^{-1} \leftarrow H_{t-1}^{-1} - H_{t-1}^{-1} x_{t,a_t} x_{t,a_t}^\top H_{t-1}^{-1} / (1 + x_{t,a_t}^\top H_{t-1}^{-1} x_{t,a_t})$;
6: $V_1^{-1} \leftarrow H_{\mathcal{T}_1}^{-1}, \hat{\theta}_1 \leftarrow V_1^{-1} \sum_{t=1}^{\mathcal{T}_1} r_t x_{t,a_t}, H_{\mathcal{T}_1} \leftarrow \lambda I$;
7: **for** $\ell \leftarrow 2, \ldots, B$ **do**
8:     **for** $t \leftarrow \mathcal{T}_{\ell-1} + 1, \ldots, \mathcal{T}_\ell$ **do**
9:         **for** $k \leftarrow 1, \ldots, \ell - 1$ **do**
10:             $x_t^{(k)} \leftarrow \arg\max_{x \in \mathcal{A}_t^{(k-1)}} \langle x, \hat{\theta}_k \rangle, \mathcal{A}_t^{(k)} \leftarrow \left\{ x \in \mathcal{A}_t^{(k-1)} \,\middle|\, \langle \hat{\theta}_k, x_t^{(k)} - x \rangle \leq 2\varepsilon_{t,k} \right\}$;
11:         **if** $t \leq \mathcal{T}_{\ell-1} + \left\lceil cT^{1-2^{-\ell}} / \log_2 \log_2 T \right\rceil$ **then**
12:             Pull arm $x_{t,a_t} \in \arg\max_{x \in \mathcal{A}_t^{(\ell-1)}} \|x\|_{H_{t-1}^{-1}}$;
13:         **else**
14:             Pull arm $x_{t,a_t} \in \arg\max_{x \in \mathcal{A}_t^{(\ell-1)}} \langle x, \hat{\theta}_{\ell-1} \rangle$;
15:         $H_t^{-1} \leftarrow H_{t-1}^{-1} - H_{t-1}^{-1} x_{t,a_t} x_{t,a_t}^\top H_{t-1}^{-1} / (1 + x_{t,a_t}^\top H_{t-1}^{-1} x_{t,a_t})$;
16:     $V_\ell^{-1} \leftarrow H_{\mathcal{T}_\ell}^{-1}, \hat{\theta}_\ell \leftarrow V_\ell^{-1} \sum_{t=\mathcal{T}_{\ell-1}+1}^{\mathcal{T}_\ell} r_t x_{t,a_t}, H_{\mathcal{T}_\ell} \leftarrow \lambda I$;

---

*(3) $\mathcal{A}_s \perp\!\!\!\perp \{\mathcal{A}_u, x_{u,a_u}, r_u\}_{u=\mathcal{T}_{\ell-1}+1}^{s-1} | \mathcal{F}_{\mathcal{T}_{\ell-1}}$ for $s \in (\mathcal{T}_{\ell-1} + 1, \mathcal{T}_\ell]$.*

*Given the history $\mathcal{F}_{\mathcal{T}_{\ell-1}}$, condition (1) requires batches $\ell$ and $\ell + 1$ to share the same conditional context law; condition (2) enforces that the contexts are conditionally independent of the contexts/actions/rewards realized in previous batch; and condition (3) imposes within-batch conditional independence of each context from earlier within-batch observations. These assumptions are strictly weaker than full i.i.d., requiring only (i) equality of the conditional context law across consecutive batches and (ii) conditional independence across and within batches. This relaxation offers greater modeling flexibility while preserving the guarantees established in Appendix B.*

### 4.4 Time-Complexity of Algorithms

The computational bottlenecks of BLCE-G are the near G-optimal design step and arm elimination. By Corollary 4, each call to the near G-optimal design costs $\mathcal{O}(Kd^3)$ operations, giving a total cost of $\mathcal{O}(Kd^3T)$. For arm elimination, computing $\varepsilon_{t,k}$ requires $\mathcal{O}(Kd^2)$ operations; since the number of elimination rounds $k$ is at most $\mathcal{O}(\log \log T)$, this step costs $\mathcal{O}(Kd^2T \log \log T)$. Thus, the overall complexity of BLCE-G is $\mathcal{O}(Kd^2T(d + \log \log T))$. For BLCE, the only bottleneck is arm elimination, which follows the same procedure as in BLCE-G, yielding a total complexity of $\mathcal{O}(Kd^2T \log \log T)$.

## 5 Extensions to Generalized Linear Contextual Bandits

### 5.1 Proposed Algorithm

Here, we propose BGLE (*Batched Generalized Linear Contextual Bandit with Elimination*), whose pseudocode is given in Algorithm 3. To extend our approach to the generalized linear setting, we build on the structure of Algorithm 2. In the first batch, the algorithm repeatedly pulls the most informative direction with respect to the current Gram matrix (Line 4) and updates its inverse via the Sherman–Morrison (Line 5). At the batch boundary, we set $V_1 := H_{\mathcal{T}_1}$, compute the MLE $\hat{\theta}_1$ for the per–round log-loss $\ell_t(\theta) = m(\langle x_{t,a_t}, \theta \rangle) - r_t \langle x_{t,a_t}, \theta \rangle$, and reinitialize $H_{\mathcal{T}_1}$ for the next batch (Line 6). For each batch $\ell \geq 2$, the Gram matrix is *weighted* by $\alpha_{t,\ell-1}(\lambda) \dot{\mu}(\langle x_{t,a_t}, \hat{\theta}_{\ell-1} \rangle)$, where

$$\alpha_{t,k}(\lambda) = \exp(-2RS) \mathbb{1}_{\{k=1\}} + \exp(-R(2S \wedge \|x_{t,a_t}\|_{V_k^{-1}} \beta(\lambda))) \mathbb{1}_{\{k \geq 2\}}$$

(Line 17). Beginning at batch $\ell \geq 3$, the algorithm performs $\ell - 2$ elimination rounds using the estimates $\hat{\theta}_2, \ldots, \hat{\theta}_{\ell-1}$, yielding nested feasible sets $\mathcal{A}_t^{(2)}, \ldots, \mathcal{A}_t^{(\ell-1)}$ (Line 11-12). Because no elimination is conducted with $\hat{\theta}_1$, we set $\mathcal{A}_t = \mathcal{A}_t^{(0)} = \mathcal{A}_t^{(1)}$. The elimination threshold $\varepsilon'_{t,k}(\lambda)$ for $k \in [\ell - 1] \setminus \{1\}$ is defined as

$$\max_{y \in \mathcal{A}_t^{(k-1)}} \|y\|_{V_k^{-1}} \left( 24RS(\sqrt{d + \log T} + R(d + \log T)/\sqrt{\lambda}) + 2S\sqrt{\lambda} \right),$$

which, under the choice $\lambda = R^2(d + \log T)$, simplifies to $\max_{y \in \mathcal{A}_t^{(k-1)}} \|y\|_{V_k^{-1}}(50RS\sqrt{d + \log T})$. Within each batch $\ell \geq 2$, the action selection strategy follows that of BLCE, splitting the batch in the ratio $c : (1 - c)$ between exploration and exploitation. The key difference is that arm selection is based on the *weighted* Gram matrix (Lines 14 and 16). At the end of batch $\ell$, we set $V_\ell := H_{\mathcal{T}_\ell}$, compute the MLE $\hat{\theta}_\ell$ for $\ell_t(\theta)$, and reinitialize $H_{\mathcal{T}_\ell}$ for the next batch (Line 18).

## 5.2 REGRET ANALYSIS FOR BATCHED GENERALIZED LINEAR CONTEXTUAL BANDITS

To analyze BGLE, we introduce parameters that capture problem non-linearity. For any arm set $\mathcal{A}$, let $x^* \in \arg\max_{x \in \mathcal{A}} \mu(\langle x, \theta^* \rangle)$ denote the optimal arm, and define

$$\kappa := \max_{\mathcal{A} \in \text{supp}(\mathcal{D})} \max_{x \in \mathcal{A}} \frac{1}{\dot{\mu}(\langle x, \theta^* \rangle)} \,, \quad \hat{\kappa} := \frac{1}{\mathbb{E}_{\mathcal{A} \sim \mathcal{D}}[\dot{\mu}(\langle x^*, \theta^* \rangle)]} \,, \quad R_{\dot{\mu}} := \max_{\mathcal{A} \in \text{supp}(\mathcal{D})} \dot{\mu}(\langle x^*, \theta^* \rangle) \,.$$

Here, $\kappa$ captures the worst-case curvature, $\hat{\kappa}$ the average inverse curvature at the optimal arm, and $R_{\dot{\mu}}$ the maximum derivative of the link function at optimal arms.

**Theorem 3** (Regret of BGLE). *Consider running the BGLE algorithm for $T$ rounds with $K$ arms in $d$ dimensions. The worst-case cumulative regret satisfies*

$$\mathcal{R}(T) = \mathcal{O}\left( RS\sqrt{d(d + \log T)T \log T \log\log T / \hat{\kappa}} \right) \qquad \text{(leading term)}$$

$$+ \mathcal{O}\left( (R^2 S e^{8RS} d(d + \log T) \log T \log\log T + R/\log\log T) T^{1/3} \right) \quad \text{(transient term)}$$

$$= \widetilde{\mathcal{O}}(RSd\sqrt{T}/\sqrt{\hat{\kappa}}) + \widetilde{\mathcal{O}}((R^2 S e^{8RS} d^2 + R)T^{1/3}) \,.$$

**Discussion of Theorem 3.** Theorem 3 shows that both the leading and transient terms of BGLE are $\kappa$-*free*, in sharp contrast to (Sawarni et al., 2024), whose transient term depends on $\kappa$. Since $\dot{\mu}(z) \to 0$ in saturation, $\kappa$ can grow arbitrarily large, so removing this dependence is a substantial improvement. Moreover, BGLE uses only $\mathcal{O}(\log\log T)$ batches, matching the lowest known batch complexity. Because $\frac{1}{\hat{\kappa}} \leq R_{\dot{\mu}}$, the leading term in our regret bound is strictly smaller than that of Sawarni et al. (2024); by eliminating the extraneous $d$ in $\sqrt{d/\hat{\kappa}}$, our bound attains a sharper dependence on $\hat{\kappa}$, thereby addressing the open question noted in that work. Finally, by building on the BLCE framework, BGLE inherits substantially lower computational complexity.

**Remark 2.** *The total time complexity of BGLE is $\mathcal{O}(Kd^2T \log\log T + \mathcal{C}_{\text{opt}}^{\text{tot}})$, where the first term comes from the BLCE, and the second from computing the MLE at batch boundaries.*

## 6 NUMERICAL EXPERIMENTS

We evaluate the performance of BLCE-G and BLCE over a horizon of $T = 10{,}000$ across 10 independent runs. At each round, $K$ arms are sampled i.i.d. from a $d$-dimensional uniform distribution, and the parameter $\theta^*$ is drawn from a $d$-dimensional normal distribution. We consider four $(K, d)$ pairs: $(1000, 5)$ and $(5000, 10)$, representing the large-$K$ regime, and $(50, 20)$ and $(100, 30)$, representing the small-$K$ regime. For comparison, we benchmark against state-of-the-art algorithms: RS-OFUL (Abbasi-Yadkori et al., 2011), BatchLinUCB-DG (Ruan et al., 2021), SoftBatch (Hanna et al., 2023b), and BatchLearning (Zhang et al., 2025). Hyperparameters are set consistently with theory, ensuring all choices satisfy the required conditions: BLCE-G and BLCE use within-batch allocation rate $c = 0.5$; RS-OFUL uses switching parameter $C = 1$; and SoftBatch employs discretization parameter $q = 1/(8\sqrt{d})$. Algorithms requiring G-optimal design are implemented using the same near G-optimal routine. Due to the substantial computational overhead reported in Table 1, we omit regret plots for the methods of Hanna et al. (2023b).

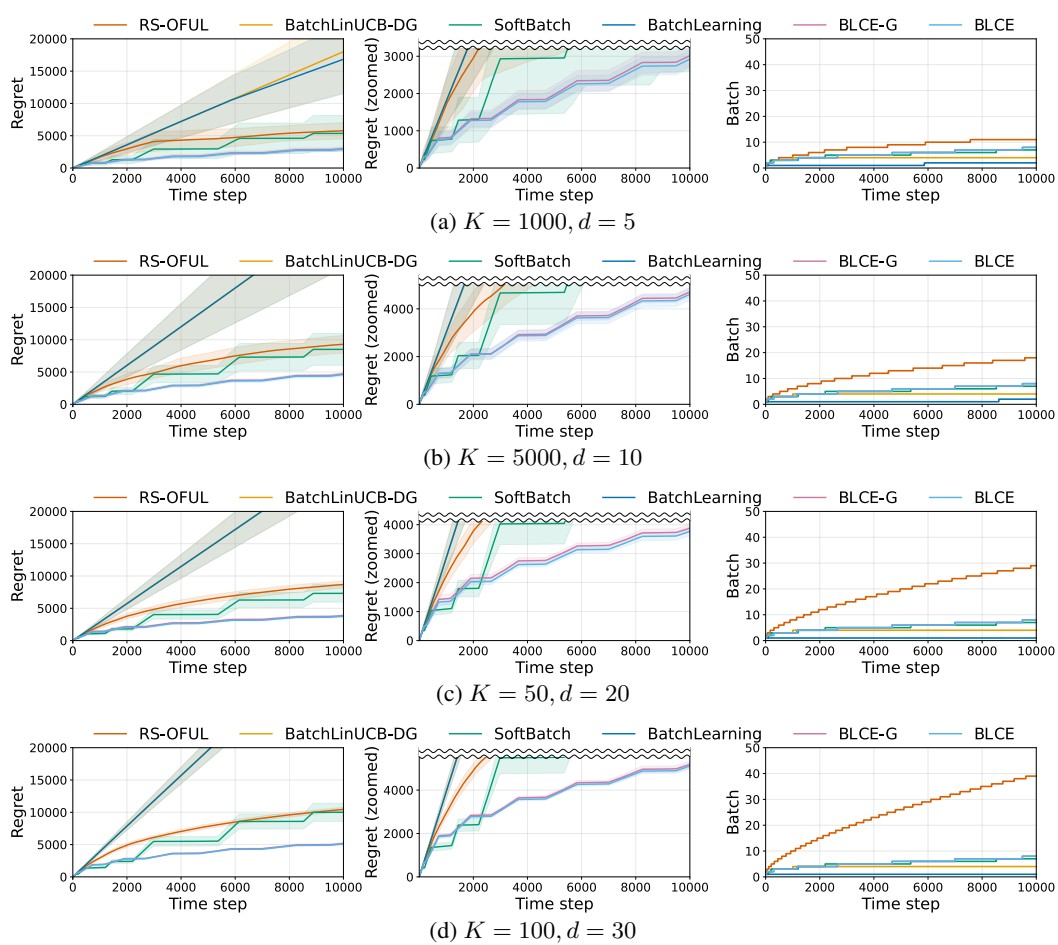

Figure 1: Regret, zoomed-in regret, and batch complexity over time for different values of $K$ and $d$.

We present three types of figures: (i) the average cumulative regret (solid line) with its standard deviation (shaded region) over 10 runs, (ii) zoomed-in views of regret curves to highlight the differences between BLCE-G and BLCE, and (iii) the average batch complexity across 10 runs, showing the frequency of policy updates. As shown in Figure 1, both BLCE-G and BLCE consistently outperform all baselines in both large-$K$ and small-$K$ regimes, achieving the lowest regret with greater stability. Runtime comparisons in Table 3 further show that our methods incur substantially lower computational cost; in particular, BLCE, which eliminates G-optimal design entirely, achieves the fastest runtime among optimal algorithms, comparable even to suboptimal baselines. Overall, these results demonstrate that BLCE-G and BLCE combine minimax-optimal regret with practical efficiency.

For generalized linear contextual bandits, BGLE similarly outperforms the baseline, achieving lowest regret, stable performance, and reduced computation. See Appendix D.2 for details.

Table 3: Average runtime (seconds) over 10 runs.

| $(K, d)$ | Suboptimal algorithms | | Optimal algorithms | | | | |
|---|---|---|---|---|---|---|---|
| | RS-OFUL | SoftBatch | BatchLinUCB-DG | Hanna et al. (2023b) | BatchLearning | BLCE-G | BLCE |
| $(1000, 5)$ | 0.85 | 1.18 | 290.87 | Exponential | 166.17 | 23.40 | 5.91 |
| $(5000, 10)$ | 4.15 | 13.17 | 1300.01 | Exponential | 621.09 | 40.27 | 12.83 |
| $(50, 20)$ | 0.42 | 1.74 | 1031.66 | Exponential | 45.85 | 2.26 | 1.06 |
| $(100, 30)$ | 0.61 | 3.50 | 2987.07 | Exponential | 77.01 | 3.70 | 1.62 |

REPRODUCIBILITY STATEMENT

All theoretical results in Section 4.3 are accompanied by complete proofs provided in the appendix, and the full set of employed assumptions is clearly specified in Section 3.2 and Section 3.3. The numerical experiments reported in Section 6 and additional experiments in Appendix D are fully reproducible: we provide the source code, along with implementation details, as supplementary material to facilitate verification and replication of our results.

USE OF LARGE LANGUAGE MODELS

Large Language Models (LLMs) were used solely as an assistive tool for writing. Specifically, we employed an LLM to improve clarity, grammar, and style of exposition. No part of the research ideation, algorithm design, theoretical analysis, or experimental results involved the use of LLMs. The authors take full responsibility for the content of the paper.

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

# Appendix

## A  PROOF OF THEOREM 1

**Lemma 1.** *(Yu & Oh, 2025) Let $V_\ell$ be the Gram matrix and $\hat{\theta}_\ell$ be the least squares estimator obtained from the contexts in the $\ell$-th batch ($\ell \geq 1$). Then, for any $x \in \mathbb{R}^d$ and $0 < \delta < 1$, the following inequality holds with probability at least $1 - \delta$*

$$|\langle x, \hat{\theta}_\ell - \theta^* \rangle| \leq \left( \sqrt{2\log\left(\frac{1}{\delta}\right)} + \sqrt{\lambda} \right) \|x\|_{V_\ell^{-1}} .$$

**Lemma 2.** *(Yu & Oh, 2025) Let $V_\ell$ be the Gram matrix and $\hat{\theta}_\ell$ be the least squares estimator obtained from the contexts in the $\ell$-th batch ($\ell \geq 1$). Then, for any $0 < \zeta, \delta < 1$ and $d \geq 2$, the following inequality holds with probability at least $1 - \delta$*

$$\|\hat{\theta}_\ell - \theta^*\|_{V_\ell} \leq \frac{\sqrt{(d-1)\log(\frac{4}{4\zeta^2-\zeta^4}) + 2\log(\frac{\sqrt{2\pi d}}{\delta})} + \sqrt{\lambda}}{1 - \zeta} .$$

Lemma 1 and Lemma 2 were originally proved in the linear bandit setting. However, the proofs do not rely on the non-contextual assumption that the feature vectors of arms remain fixed. Therefore, the results can be directly applied to the linear contextual bandit setting as well.

**Corollary 1.** *Let $V_\ell$ be the Gram matrix and $\hat{\theta}_\ell$ be the least squares estimator obtained from the contexts in the $\ell$-th batch ($\ell \geq 1$). Then, for any $0 < \delta < 1$ and $d \geq 2$, the following inequality holds with probability at least $1 - \delta$*

$$\|\hat{\theta}_\ell - \theta^*\|_{V_\ell} \leq 2\sqrt{\log\left(\frac{2^{6d-5}\pi d}{15^{d-1}\delta^2}\right)} + 2\sqrt{\lambda} .$$

*Proof.* Substituting $\zeta = 0.5$ into Lemma 2 yields the desired result. $\qquad\square$

**Lemma 3** (Good event). *Define the following quantities:*

$$\beta_{t,\ell}^{(1)}(\delta) := \sqrt{2\log\left(\frac{2|\mathcal{A}_t^{(\ell-1)}|(B-1)T}{\delta}\right)} + \sqrt{\lambda},$$

$$\beta_{t,\ell}^{(2)}(\delta) := 2\sqrt{\log\left(\frac{2^{6d-3}\pi d(B-1)^2}{15^{d-1}\delta^2}\right)} + 2\sqrt{\lambda},$$

$$\varepsilon_{t,\ell}(\delta) := \max_{y \in \mathcal{A}_t^{(\ell-1)}} \|y\|_{V_\ell^{-1}} \cdot \left( \beta_{t,\ell}^{(1)}(\delta) \wedge \beta_{t,\ell}^{(2)}(\delta) \right) .$$

*Here, $\delta$ is a constant in the interval $(0,1)$. Then, the following event $E$ holds with probability at least $1 - \delta$*

$$E := \bigcap_{\ell=1}^{B-1} \bigcap_{t=\mathcal{T}_\ell+1}^{T} \left\{ |\langle x, \hat{\theta}_\ell - \theta^* \rangle| \leq \varepsilon_{t,\ell}(\delta), \ \forall x \in \mathcal{A}_t^{(\ell-1)} \right\} .$$

*Proof.* Fix arbitrary $\ell$ and $t$. By Lemma 1, the following inequality holds for all $x \in \mathcal{A}_t^{(\ell-1)}$ with probability at least $1 - \frac{\delta}{2(B-1)T}$

$$|\langle x, \hat{\theta}_\ell - \theta^* \rangle| \leq \max_{y \in \mathcal{A}_t^{(\ell-1)}} \|y\|_{V_\ell^{-1}} \cdot \left( \sqrt{2\log\left(\frac{2|\mathcal{A}_t^{(\ell-1)}|(B-1)T}{\delta}\right)} + \sqrt{\lambda} \right) .$$

Applying a union bound over all $\ell$ and $t$, the following event holds with probability at least $1 - \frac{\delta}{2}$

$$\bigcap_{\ell=1}^{B-1} \bigcap_{t=\mathcal{T}_\ell+1}^{T} \left\{ |\langle x, \hat{\theta}_\ell - \theta^* \rangle| \leq \max_{y \in \mathcal{A}_t^{(\ell-1)}} \|y\|_{V_\ell^{-1}} \cdot \beta_{t,\ell}^{(1)}(\delta), \ \forall x \in \mathcal{A}_t^{(\ell-1)} \right\} . \tag{1}$$

Next, for fixed $\ell$, by the Cauchy-Schwarz inequality and Corollary 1, the following inequality holds for all $t$ and $x \in \mathcal{A}_t^{(\ell-1)}$ with probability at least $1 - \frac{\delta}{2(B-1)}$

$$|\langle x, \hat{\theta}_\ell - \theta^* \rangle| \leq \|x\|_{V_\ell^{-1}} \|\hat{\theta}_\ell - \theta^*\|_{V_\ell} \leq \max_{y \in \mathcal{A}_t^{(\ell-1)}} \|y\|_{V_\ell^{-1}} \cdot \left( 2\sqrt{\log\left(\frac{2^{6d-3}\pi d(B-1)^2}{15^{d-1}\delta^2}\right)} + 2\sqrt{\lambda} \right) .$$

A union bound over $\ell$ yields the following event with probability at least $1 - \frac{\delta}{2}$

$$\bigcap_{\ell=1}^{B-1} \bigcap_{t=\mathcal{T}_\ell+1}^{T} \left\{ |\langle x, \hat{\theta}_\ell - \theta^* \rangle| \leq \max_{y \in \mathcal{A}_t^{(\ell-1)}} \|y\|_{V_\ell^{-1}} \cdot \beta_{t,\ell}^{(2)}(\delta), \ \forall x \in \mathcal{A}_t^{(\ell-1)} \right\} . \tag{2}$$

Combining event (1) and event (2), we conclude that the following event holds with probability at least $1 - \delta$

$$\bigcap_{\ell=1}^{B-1} \bigcap_{t=\mathcal{T}_\ell+1}^{T} \left\{ |\langle x, \hat{\theta}_\ell - \theta^* \rangle| \leq \max_{y \in \mathcal{A}_t^{(\ell-1)}} \|y\|_{V_\ell^{-1}} \cdot \left( \beta_{t,\ell}^{(1)}(\delta) \wedge \beta_{t,\ell}^{(2)}(\delta) \right), \ \forall x \in \mathcal{A}_t^{(\ell-1)} \right\} .$$

$\square$

**Lemma 4.** *Let $E$ be the good event defined in Lemma 3 with $\delta = \frac{2}{T}$. Conditioned on $E$, the optimal arm $x_t^* \in \arg\max_{x \in \mathcal{A}_t} \langle x, \theta^* \rangle$ is never eliminated at any round $t$. In particular,*

$$x_t^* \in \mathcal{A}_t^{(\ell)}, \quad \text{for all } 1 \leq \ell \leq B-1 \text{ and } \mathcal{T}_\ell + 1 \leq t \leq T .$$

*Proof.* Fix $t \in [\mathcal{T}_s + 1, \mathcal{T}_{s+1}]$ for some $s \in [B-1]$. We show by induction on $\ell$ that $x_t^* \in \mathcal{A}_t^{(\ell)}$ for all $\ell \in [s]$.

**Base case** ($\ell = 1$). Since both $x_t^*$ and $x_t^{(1)}$ belong to $\mathcal{A}_t^{(0)}(= \mathcal{A}_t)$, we have

$$\begin{aligned}
\langle \hat{\theta}_1, x_t^{(1)} - x_t^* \rangle &= \langle \hat{\theta}_1 - \theta^*, x_t^{(1)} - x_t^* \rangle + \langle \theta^*, x_t^{(1)} - x_t^* \rangle \\
&\leq \langle \hat{\theta}_1 - \theta^*, x_t^{(1)} - x_t^* \rangle \\
&\leq |\langle \hat{\theta}_1 - \theta^*, x_t^{(1)} \rangle| + |\langle \hat{\theta}_1 - \theta^*, x_t^* \rangle| \\
&\leq 2\varepsilon_{t,1} ,
\end{aligned}$$

where the first inequality follows from the optimality of $x_t^*$ and the last from the definition of the good event $E$. Hence $x_t^* \in \mathcal{A}_t^{(1)}$.

**Inductive step.** Assume $x_t^* \in \mathcal{A}_t^{(\ell-1)}$ for some $\ell \in \{2, \ldots, s\}$. Since $x_t^{(\ell)} \in \mathcal{A}_t^{(\ell-1)}$, we similarly obtain

$$\begin{aligned}
\langle \hat{\theta}_\ell, x_t^{(\ell)} - x_t^* \rangle &= \langle \hat{\theta}_\ell - \theta^*, x_t^{(\ell)} - x_t^* \rangle + \langle \theta^*, x_t^{(\ell)} - x_t^* \rangle \\
&\leq \langle \hat{\theta}_\ell - \theta^*, x_t^{(\ell)} - x_t^* \rangle \\
&\leq |\langle \hat{\theta}_\ell - \theta^*, x_t^{(\ell)} \rangle| + |\langle \hat{\theta}_\ell - \theta^*, x_t^* \rangle| \\
&\leq 2\varepsilon_{t,\ell} ,
\end{aligned}$$

which shows that $x_t^* \in \mathcal{A}_t^{(\ell)}$. By induction, the claim holds for all $\ell \in [s]$, completing the proof. $\square$

**Lemma 5.** *(Abbasi-Yadkori et al., 2011) Let $\{x_1, \ldots, x_n\} \subset \mathbb{R}^d$ be a sequence of vectors such that $\|x_i\|_2 \leq 1$ for all $i \in [n]$. Let $H_0 \in \mathbb{R}^{d \times d}$ be a positive definite matrix, and define $H_t := H_0 + \sum_{i=1}^{t} x_i x_i^\top$. Then, the following inequality holds*

$$\sum_{t=1}^{n} \min\left\{ 1, \|x_t\|_{H_{t-1}^{-1}}^2 \right\} \leq 2\log\left(\frac{\det(H_n)}{\det(H_0)}\right) .$$

**Corollary 2.** *Let $\{x_1, \ldots, x_n\} \subset \mathbb{R}^d$ be a sequence of vectors such that $\|x_i\|_2 \leq 1$ for all $i \in [n]$. Suppose $\lambda \geq 1$, and define $H_0 := \lambda I$ and $H_t := \lambda I + \sum_{i=1}^{t} x_i x_i^\top$ for each $t \in \{1, \ldots, n\}$. Then, for any $1 \leq m \leq n$,*

$$\sum_{t=m}^{n} \|x_t\|_{H_{t-1}^{-1}}^2 \leq 2 \log \left( \frac{\det(H_n)}{\det(H_{m-1})} \right) .$$

*Proof.* Since $H_{t-1}^{-1} \preceq \lambda^{-1} I$ for all $t$, it follows that $H_{t-1}^{-1} \preceq \lambda^{-1} I \preceq I$. Consequently, $\|x_t\|_{H_{t-1}^{-1}}^2 \leq \|x_t\|_2^2 \leq 1$ for each $t \in [m, n]$. Applying Lemma 5 over the interval $[m, n]$ yields the stated bound. $\square$

**Lemma 6.** *Let $H$ be a positive definite matrix. Suppose $x \in \mathbb{R}^d$ satisfies $\|x\|_{H^{-1}}^2 \leq c$. Then $cH \succeq xx^\top$.*

*Proof.* For any $z \in \mathbb{R}^d$, we have

$$z^\top (cH) z \geq \|x\|_{H^{-1}}^2 \cdot \|z\|_H^2 = \|H^{-\frac{1}{2}} x\|^2 \cdot \|H^{\frac{1}{2}} z\|^2 \geq \langle x, z \rangle^2 = z^\top (xx^\top) z .$$

The first inequality follows from the assumption $\|x\|_{H^{-1}}^2 \leq c$. The second inequality follows from the Cauchy–Schwarz inequality. Since this bound holds for all $z \in \mathbb{R}^d$, the matrix inequality $cH \succeq xx^\top$ follows. $\square$

**Lemma 7.** *(Ruan et al., 2021) Let $x_1, \ldots, x_n$ be independent and identically distributed (i.i.d.) random vectors drawn from a distribution $\mathcal{D}$ such that $\|x_i\|_2 \leq 1$ almost surely. For any cutoff level $\lambda > 0$, the following inequality holds with probability at least $1 - 2d \exp(-\frac{\lambda n}{8})$*

$$3\lambda I + \frac{1}{n} \sum_{i=1}^{n} x_i x_i^\top \succeq \frac{1}{8} \mathbb{E}_{x \sim \mathcal{D}} \left[ xx^\top \right]$$

**Corollary 3.** *Let $x_1, \ldots, x_n$ be independent and identically distributed (i.i.d.) random vectors drawn from a distribution $\mathcal{D}$ such that $\|x_i\|_2 \leq 1$ almost surely. Then, the following inequality holds with probability at least $1 - \frac{2}{T}$*

$$\frac{24 \log(dT)}{n} I + \frac{1}{n} \sum_{i=1}^{n} x_i x_i^\top \succeq \frac{1}{8} \mathbb{E}_{x \sim \mathcal{D}} \left[ xx^\top \right]$$

*Proof.* The proof is a direct application of Lemma 7. We achieve the desired inequality by setting the cutoff level $\lambda$ to $\lambda = \frac{8 \log(dT)}{n}$. $\square$

**Lemma 8.** *(Todd & Yıldırım, 2007) Let $X = \{x_1, \ldots, x_K\} \subset \mathbb{R}^d$ be a set of $K$ points that spans $\mathbb{R}^d$, and fix $\varepsilon \in (0, 1]$. Then Khachiyan's barycentric coordinate descent algorithm computes a $(1 + \varepsilon)$-approximation to the minimum-volume enclosing ellipsoid of $X$ in*

$$O\Big(Kd^2 \left([(1 + \varepsilon)^{2/(d+1)} - 1]^{-1} + \log d\right)\Big)$$

*arithmetic operations. In particular, since $[(1 + \varepsilon)^{2/(d+1)} - 1]^{-1} = \Theta(d/\varepsilon)$ for $\varepsilon \in (0, 1]$, the total time complexity simplifies to*

$$O\Big(\frac{Kd^3}{\varepsilon}\Big) .$$

**Corollary 4** (Near G-optimal design). *Let $X = \{x_1, \ldots, x_K\} \subset \mathbb{R}^d$ and let $r = \text{rank}(X) \leq d$. Let $U \in \mathbb{R}^{d \times r}$ have orthonormal columns spanning $\text{span}\{x_i\}$, and define $x_i' := U^\top x_i \in \mathbb{R}^r$ together with $\Sigma'(w) := \sum_{i=1}^{K} w_i x_i' x_i'^\top$ for $w \in \Delta_K$ (K-dimensional probability simplex). Applying the algorithm of Lemma 8 to $\{x_i'\}_{i=1}^{K}$ with accuracy parameter fixed to $\varepsilon = 1$ returns weights $w_\circ \in \Delta_K$ such that*

$$\max_{j \in [K]} x_j'^\top \Big(\Sigma'(w_\circ)\Big)^{-1} x_j' \leq 2r ,$$

*in*

$$O\Big(Kr^2 \big([(1 + 1)^{2/(r+1)} - 1]^{-1} + \log r\big)\Big) = O(Kr^3)$$

*arithmetic operations. Equivalently, in the original space we have*

$$\max_{j \in [K]} x_j^\top \big( \Pi\, \Sigma(w_\circ)^+ \Pi \big)\, x_j \;\le\; 2\,r \;, \qquad where\; \Sigma(w) := \sum_{i=1}^{K} w_i\, x_i x_i^\top, \; \Pi := U U^\top \;,$$

*with $^+$ denoting the Moore–Penrose pseudoinverse. Hence $w_\circ$ is a constant-factor near-optimal design in the effective dimension $r$, computed in polynomial time $O(Kr^3)$ by the method in Lemma 8.*

*Proof.* Consider the subspace spanned by $\{x_i\}$ of dimension $r$, let $U$ be an orthonormal basis, and project $x_i' := U^\top x_i \in \mathbb{R}^r$. The MVEE and $D/G$-optimal design problems are equivalently posed in $\mathbb{R}^r$, and guarantees for $x_i'$ and $\Sigma'(w) := \sum_i w_i x_i' x_i'^\top$ transfer back to the original space through the projector $\Pi = U U^\top$ and the Moore–Penrose pseudoinverse. By Lemma 8, Khachiyan's barycentric coordinate descent computes an $\varepsilon$-approximate MVEE of $\{x_i'\}$ in

$$O\Big( Kr^2 \big([(1+\varepsilon)^{2/(r+1)} - 1]^{-1} + \log r \big)\Big)$$

arithmetic operations; fixing $\varepsilon = 1$ gives $O(Kr^3)$.

Moreover, the standard MVEE $\leftrightarrow D/G$-optimal duality and the algorithm's stopping rule imply that the returned weights $w_\circ \in \Delta_K$ satisfy the *constraint-violation guarantee*

$$\max_{j \in [K]} x_j'^\top \Big( \Sigma'(w_\circ) \Big)^{-1} x_j' \;\le\; (1+\varepsilon)\,r \;.$$

With $\varepsilon = 1$ this yields

$$\max_{j \in [K]} x_j'^\top \Big( \Sigma'(w_\circ) \Big)^{-1} x_j' \;\le\; 2\,r \;.$$

Finally, lifting back to the original space uses the identity

$$x_j'^\top \Big( \Sigma'(w_\circ) \Big)^{-1} x_j' \;=\; x_j^\top \big( \Pi\, \Sigma(w_\circ)^+ \Pi \big)\, x_j \;, \qquad \Sigma(w) := \sum_{i=1}^{K} w_i\, x_i x_i^\top, \; \Pi := U U^\top,$$

so that

$$\max_{j \in [K]} x_j^\top \big( \Pi\, \Sigma(w_\circ)^+ \Pi \big)\, x_j \;\le\; 2\,r \;.$$

Hence $w_\circ$ is a constant-factor near-$G$-optimal design in effective dimension $r$, computed in polynomial time $O(Kr^3)$ by the method in Lemma 8. $\qquad\square$

*Proof of Theorem 1.* To establish the claimed batch complexity bound, we analyze the lower bound on the length of the $\ell$-th batch, where $\ell = \lceil \log_2 \log_2 T \rceil$. By definition of the schedule, the length of this batch is at least

$$\left\lceil \frac{T^{1-2^{-\ell}}}{\log_2 \log_2 T} \right\rceil + 2 \ge \frac{T^{1-2^{-\lceil \log_2 \log_2 T \rceil}}}{\log_2 \log_2 T} \ge \frac{T^{1-2^{-\log_2 \log_2 T}}}{\log_2 \log_2 T} = \frac{T^{1-\frac{1}{\log_2 T}}}{\log_2 \log_2 T} = \frac{T}{2 \log_2 \log_2 T} \;.$$

Since the batch length is non-decreasing in $\ell$, every batch with index $\ell \ge \lceil \log_2 \log_2 T \rceil$ has length at least $\frac{T}{2 \log_2 \log_2 T}$. Given the total time horizon $T$, the number of such batches is therefore at most $\lceil 2 \log_2 \log_2 T \rceil$. Including the initial $\lceil \log_2 \log_2 T \rceil - 1$ batches, the total number of batches $B$ is bounded by

$$B \le \lceil 2 \log_2 \log_2 T \rceil + \lceil \log_2 \log_2 T \rceil - 1 \le 3 \log_2 \log_2 T + 1 \;,$$

which implies that the batch complexity is $\mathcal{O}(\log \log T)$.

Next, we begin by decomposing the cumulative expected regret based on the good event $E$. The regret can be written as

$$\mathcal{R}(T) = \sum_{k=1}^{B} \sum_{t=\mathcal{T}_{k-1}+1}^{\mathcal{T}_k} \mathbb{E}\left[ \max_{x \in \mathcal{A}_t} \langle x, \theta^* \rangle - \langle x_{t,a_t}, \theta^* \rangle \right]$$

$$= \sum_{k=1}^{B} \sum_{t=\mathcal{T}_{k-1}+1}^{\mathcal{T}_k} \mathbb{E}\left[\langle x_t^* - x_{t,a_t}, \theta^* \rangle \Big| E^c \right] \cdot \mathbb{P}(E^c) + \mathbb{E}\left[\langle x_t^* - x_{t,a_t}, \theta^* \rangle \Big| E \right] \cdot \mathbb{P}(E) .$$

Using the triangle inequality followed by the Cauchy-Schwarz inequality, we have

$$\begin{aligned} \langle x_t^* - x_{t,a_t}, \theta^* \rangle &\leq |\langle x_t^* - x_{t,a_t}, \theta^* \rangle| \\ &\leq |\langle x_t^*, \theta^* \rangle| + |\langle x_{t,a_t}, \theta^* \rangle| \\ &\leq \|x_t^*\| \cdot \|\theta^*\| + \|x_{t,a_t}\| \cdot \|\theta^*\| \\ &\leq 2 , \end{aligned}$$

where the last inequality follows from Assumption 1. Hence, the cumulative expected regret can be bounded as

$$\mathcal{R}(T) \leq 2T \cdot \mathbb{P}(E^c) + \sum_{k=1}^{B} \sum_{t=\mathcal{T}_{k-1}+1}^{\mathcal{T}_k} \mathbb{E}\left[\langle x_t^* - x_{t,a_t}, \theta^* \rangle \Big| E \right] \cdot \mathbb{P}(E)$$

$$\leq \mathcal{O}(1) + \sum_{k=1}^{B} \sum_{t=\mathcal{T}_{k-1}+1}^{\mathcal{T}_k} \mathbb{E}\left[\langle x_t^* - x_{t,a_t}, \theta^* \rangle \Big| E \right] ,$$

where the $\mathcal{O}(1)$ term follows from the high-probability guarantee $\mathbb{P}(E^c) = \mathcal{O}(1/T)$. Throughout the remainder of the analysis, we therefore condition on the good event $E$. Let $\text{Regret}_\ell$ denote the cumulative expected regret incurred during batch $\ell$. We analyze the regret separately for the case $\ell = 1$ and for all subsequent batches $\ell \geq 2$.

**Case 1: $\ell = 1$.** In the first batch, the number of rounds is $\mathcal{T}_1 = \left\lceil \frac{\sqrt{T}}{\log_2 \log_2 T} \right\rceil + 1$, and the instantaneous regret is bounded by 2. Therefore,

$$\text{Regret}_1 = \sum_{t=1}^{\mathcal{T}_1} \mathbb{E}[\langle x_t^* - x_{t,a_t}, \theta^* \rangle] \leq 2\mathcal{T}_1 \leq 2\left( \frac{\sqrt{T}}{\log_2 \log_2 T} + 2 \right) = \mathcal{O}(\sqrt{T}) .$$

**Case 2: $\ell \geq 2$.** For each batch $\ell \geq 2$, the rounds following the arm elimination steps are divided into three phases:

**Phases 1 & 2.** In the first two phases of length $\left\lceil \frac{c^2 T^{1-2^{-\ell}}}{\log_2 \log_2 T} \right\rceil + \left\lceil \frac{c(1-c)T^{1-2^{-\ell}}}{\log_2 \log_2 T} \right\rceil$, the algorithm selects arm according to a near G-optimal design and the most informative direction with respect to the current Gram matrix. For any round $t$ in these phases, the instantaneous regret satisfies

$$\begin{aligned} \langle x_t^* - x_{t,a_t}, \theta^* \rangle &= \langle x_t^* - x_{t,a_t}, \theta^* - \hat{\theta}_{\ell-1} \rangle + \langle x_t^* - x_{t,a_t}, \hat{\theta}_{\ell-1} \rangle \\ &\leq \langle x_t^* - x_{t,a_t}, \theta^* - \hat{\theta}_{\ell-1} \rangle + \langle x_t^{(\ell-1)} - x_{t,a_t}, \hat{\theta}_{\ell-1} \rangle \\ &\leq |\langle x_t^*, \hat{\theta}_{\ell-1} - \theta^* \rangle| + |\langle x_{t,a_t}, \hat{\theta}_{\ell-1} - \theta^* \rangle| + 2\varepsilon_{t,\ell-1} \\ &\leq 4\varepsilon_{t,\ell-1} . \end{aligned} \tag{3}$$

The first inequality uses the fact that $x_t^{(\ell-1)}$ is optimal with respect to $\hat{\theta}_{\ell-1}$ and that $x_t^*$ belongs to $\mathcal{A}_t^{(\ell-2)}$ by Lemma 4. The second inequality follows since $x_{t,a_t} \in \mathcal{A}_t^{(\ell-1)}$ by construction of the arm elimination step. The final inequality follows from the definition of the good event $E$ and again from Lemma 4, which guarantees $x_t^* \in \mathcal{A}_t^{(\ell-2)}$.

**Phase 3.** In the last phase, the algorithm selects arms greedily with respect to the estimated parameter, i.e., $x_{t,a_t} \in \arg\max_{x \in \mathcal{A}_t^{(\ell-1)}} \langle x, \hat{\theta}_{\ell-1} \rangle$. For any round $t$ in this phase, the instantaneous regret satisfies

$$\begin{aligned} \langle x_t^* - x_{t,a_t}, \theta^* \rangle &= \langle x_t^* - x_{t,a_t}, \theta^* - \hat{\theta}_{\ell-1} \rangle + \langle x_t^* - x_{t,a_t}, \hat{\theta}_{\ell-1} \rangle \\ &\leq \langle x_t^* - x_{t,a_t}, \theta^* - \hat{\theta}_{\ell-1} \rangle \\ &\leq |\langle x_t^*, \hat{\theta}_{\ell-1} - \theta^* \rangle| + |\langle x_{t,a_t}, \hat{\theta}_{\ell-1} - \theta^* \rangle| \end{aligned}$$

$$\leq 2\varepsilon_{t,\ell-1} \ . \tag{4}$$

The first inequality follows from the fact that $x_{t,a_t}$ is optimal with respect to $\hat{\theta}_{\ell-1}$ and that $x_t^*$ belongs to $\mathcal{A}_t^{(\ell-1)}$ by Lemma 4. The final inequality follows directly from the definition of the good event $E$ and from Lemma 4, as in Phases 1 & 2.

Therefore, it suffices to upper bound the quantity $\sum_{t=\mathcal{T}_\ell+1}^{\mathcal{T}_{\ell+1}} \varepsilon_{t,\ell}$ for each $\ell \in [B-1]$, which reduces to bounding the sum $\sum_{t=\mathcal{T}_\ell+1}^{\mathcal{T}_{\ell+1}} \max_{y \in \mathcal{A}_t^{(\ell-1)}} \|y\|_{V_\ell^{-1}}$. Since both $V_\ell$ and the arm elimination rule—determined by $\hat{\theta}_1, \ldots, \hat{\theta}_{\ell-1}$—are measurable with respect to $\mathcal{F}_{\mathcal{T}_\ell}$, they can be treated as fixed quantities conditional on this filtration. Given that the contexts are drawn independently and identically, it follows by the tower property that for any $t, v \in [\mathcal{T}_\ell+1, \mathcal{T}_{\ell+1}]$, we have

$$\mathbb{E}\left[\max_{y \in \mathcal{A}_t^{(\ell-1)}} y^\top V_\ell^{-1} y\right] = \mathbb{E}\left[\mathbb{E}\left[\max_{y \in \mathcal{A}_t^{(\ell-1)}} y^\top V_\ell^{-1} y \,\Big|\, \mathcal{F}_{\mathcal{T}_\ell}\right]\right]$$

$$= \mathbb{E}\left[\mathbb{E}\left[\max_{y \in \mathcal{A}_v^{(\ell-1)}} y^\top V_\ell^{-1} y \,\Big|\, \mathcal{F}_{\mathcal{T}_\ell}\right]\right]$$

$$= \mathbb{E}\left[\max_{y \in \mathcal{A}_v^{(\ell-1)}} y^\top V_\ell^{-1} y\right] . \tag{5}$$

Define $\mathcal{T}'_{\ell-1} := \mathcal{T}_{\ell-1} + \left\lceil \frac{c^2 T^{1-2^{-\ell}}}{\log_2 \log_2 T} \right\rceil$ and $\mathcal{T}''_{\ell-1} := \mathcal{T}_{\ell-1} + \left\lceil \frac{c(1-c)T^{1-2^{-\ell}}}{\log_2 \log_2 T} \right\rceil$ for all $\ell \geq 2$, whereas for the first batch ($\ell = 1$), we set $\mathcal{T}'_0 := \mathcal{T}_0 + \left\lceil \frac{c\sqrt{T}}{\log_2 \log_2 T} \right\rceil$ and $\mathcal{T}''_0 := \mathcal{T}_1$. Now, fix any $t \in [\mathcal{T}_\ell+1, \mathcal{T}_{\ell+1}]$ and consider the interval $s \in [\mathcal{T}'_{\ell-1}+1, \mathcal{T}''_{\ell-1}]$. Then, we have the following result

$$\sum_{s=\mathcal{T}'_{\ell-1}+1}^{\mathcal{T}''_{\ell-1}} \mathbb{E}\left[\max_{y \in \mathcal{A}_t^{(\ell-1)}} y^\top V_\ell^{-1} y\right] \leq \sum_{s=\mathcal{T}'_{\ell-1}+1}^{\mathcal{T}''_{\ell-1}} \mathbb{E}\left[\max_{y \in \mathcal{A}_t^{(\ell-1)}} y^\top H_{s-1}^{-1} y\right]$$

$$= \sum_{s=\mathcal{T}'_{\ell-1}+1}^{\mathcal{T}''_{\ell-1}} \mathbb{E}\left[\mathbb{E}\left[\max_{y \in \mathcal{A}_t^{(\ell-1)}} y^\top H_{s-1}^{-1} y \,\Big|\, \mathcal{F}_{s-1}\right]\right]$$

$$= \sum_{s=\mathcal{T}'_{\ell-1}+1}^{\mathcal{T}''_{\ell-1}} \mathbb{E}\left[\mathbb{E}\left[\max_{y \in \mathcal{A}_s^{(\ell-1)}} y^\top H_{s-1}^{-1} y \,\Big|\, \mathcal{F}_{s-1}\right]\right]$$

$$= \sum_{s=\mathcal{T}'_{\ell-1}+1}^{\mathcal{T}''_{\ell-1}} \mathbb{E}\left[\max_{y \in \mathcal{A}_s^{(\ell-1)}} y^\top H_{s-1}^{-1} y\right] . \tag{6}$$

The first inequality follows from the monotonicity of the matrices, as $H_{s-1} \preceq V_\ell$ for all $s$ in the interval. The first and last equalities follow from the tower property. The second equality uses the fact that, conditional on $\mathcal{F}_{s-1}$, both $H_{s-1}$ and the arm elimination rule (determined by $\hat{\theta}_1, \ldots, \hat{\theta}_{\ell-1}$) are fixed, and the distribution of the contexts remains unchanged due to their i.i.d. nature.

In Phase 2, the algorithm proceeds by selecting the most informative direction at each step with respect to the current Gram matrix. For all $\ell \geq 1$, we obtain

$$\sum_{s=\mathcal{T}'_{\ell-1}+1}^{\mathcal{T}''_{\ell-1}} \mathbb{E}\left[\max_{y \in \mathcal{A}_s^{(\ell-1)}} y^\top H_{s-1}^{-1} y\right] = \mathbb{E}\left[\sum_{s=\mathcal{T}'_{\ell-1}+1}^{\mathcal{T}''_{\ell-1}} \max_{y \in \mathcal{A}_s^{(\ell-1)}} y^\top H_{s-1}^{-1} y\right]$$

$$= \mathbb{E}\left[\sum_{s=\mathcal{T}'_{\ell-1}+1}^{\mathcal{T}''_{\ell-1}} x_{s,a_s}^\top H_{s-1}^{-1} x_{s,a_s}\right]$$

$$\leq \mathbb{E}\left[2\log\left(\frac{\det(H_{\mathcal{T}''_{\ell-1}})}{\det(H_{\mathcal{T}'_{\ell-1}})}\right)\right] , \tag{7}$$

where the second equality follows from the arm-selection strategy of Phase 2, and the first inequality follows from Corollary 2.

Next, we establish a probabilistic bound on the Gram matrix. With probability at least $1 - \frac{2}{T}$, the following holds for all $\ell \geq 1$

$$
H_{\mathcal{T}''_{\ell-1}} = H_{\mathcal{T}'_{\ell-1}} + \sum_{s=\mathcal{T}'_{\ell-1}+1}^{\mathcal{T}''_{\ell-1}} x_{s,a_s} x_{s,a_s}^\top
$$

$$
\preceq H_{\mathcal{T}'_{\ell-1}} + \sum_{s=\mathcal{T}'_{\ell-1}+1}^{\mathcal{T}''_{\ell-1}} 2d \cdot \mathbb{E}_{z \sim \pi_{G'}(\mathcal{A}_s^{(\ell-1)})} \left[ zz^\top \right]
$$

$$
= H_{\mathcal{T}'_{\ell-1}} + \frac{2d(\mathcal{T}''_{\ell-1} - \mathcal{T}'_{\ell-1})}{\mathcal{T}'_{\ell-1} - \mathcal{T}_{\ell-1}} \cdot \sum_{s=\mathcal{T}_{\ell-1}+1}^{\mathcal{T}'_{\ell-1}} \mathbb{E}_{z \sim \pi_{G'}(\mathcal{A}_s^{(\ell-1)})} \left[ zz^\top \right]
$$

$$
\preceq H_{\mathcal{T}'_{\ell-1}} + \frac{16d \left( \frac{cT^{1-2^{-\ell}}}{\log_2 \log_2 T} + 1 \right)}{\frac{c^2 T^{1-2^{-\ell}}}{\log_2 \log_2 T}} \cdot \left( 24 \log(dT) I + \sum_{s=\mathcal{T}_{\ell-1}+1}^{\mathcal{T}'_{\ell-1}} x_{s,a_s} x_{s,a_s}^\top \right)
$$

$$
\preceq \left( 1 + \frac{384d}{c} + \frac{384d \log_2 \log_2 T}{c^2 T^{1-2^{-\ell}}} \right) \cdot H_{\mathcal{T}'_{\ell-1}}
$$

$$
\preceq \frac{(384c + 385)d}{c^2} H_{\mathcal{T}'_{\ell-1}} .
$$

The first inequality follows from the application of Corollary 4 together with Lemma 6. The second equality holds because the contexts are drawn i.i.d., which allows the sum over the current phase to be related to the sum over the previous phase by a scaling factor. The second inequality follows directly from Lemma 7 since in Phase 1 the algorithm selects arm according to a near G-optimal design. The third inequality follows from the bound $24 \log(dT) I + \sum_{s=\mathcal{T}_{\ell-1}+1}^{\mathcal{T}'_{\ell-1}} x_{s,a_s} x_{s,a_s}^\top \preceq 24 H_{\mathcal{T}'_{\ell-1}}$. Finally, the last inequality is obtained by simplifying constants, using the facts that $1 \leq \frac{d}{c^2}$ and $\log_2 \log_2 T \leq T^{1-2^{-\ell}}$ for all $T \geq 1$.

We define the event $E_\ell$, which occurs with probability at least $1 - \frac{2}{T}$, as

$$
E_\ell := \left\{ H_{\mathcal{T}''_{\ell-1}} \preceq \frac{(384c + 385)d}{c^2} H_{\mathcal{T}'_{\ell-1}} \right\} .
$$

This event ensures that the Gram matrix does not grow excessively during Phase 2 of the algorithm. Conditioning on $E_\ell$ and $E_\ell^c$, we obtain the following upper bound for (7), valid for all $\ell \geq 1$

$$
\sum_{s=\mathcal{T}'_{\ell-1}+1}^{\mathcal{T}''_{\ell-1}} \mathbb{E} \left[ \max_{y \in \mathcal{A}_s^{(\ell-1)}} y^\top H_{s-1}^{-1} y \right] \leq \mathbb{E} \left[ 2 \log \left( \frac{\det(H_{\mathcal{T}''_{\ell-1}})}{\det(H_{\mathcal{T}'_{\ell-1}})} \right) \right]
$$

$$
= \mathbb{E} \left[ 2 \log \left( \frac{\det(H_{\mathcal{T}''_{\ell-1}})}{\det(H_{\mathcal{T}'_{\ell-1}})} \right) \middle| E_\ell \right] \cdot \mathbb{P}(E_\ell) + \mathbb{E} \left[ 2 \log \left( \frac{\det(H_{\mathcal{T}''_{\ell-1}})}{\det(H_{\mathcal{T}'_{\ell-1}})} \right) \middle| E_\ell^c \right] \cdot \mathbb{P}(E_\ell^c)
$$

$$
\leq 2d \log \left( \frac{(384c + 385)d}{c^2} \right) + \frac{2}{T} \cdot \mathbb{E} \left[ 2 \log \left( \frac{\det(H_{\mathcal{T}''_{\ell-1}})}{\det(\lambda I)} \right) \middle| E_\ell^c \right]
$$

$$
\leq 2d \log \left( \frac{769d}{c^2} \right) + \frac{4}{T} \cdot \mathbb{E} \left[ \log \left( \frac{(\mathrm{tr}(H_{\mathcal{T}''_{\ell-1}})/d)^d}{\lambda^d} \right) \middle| E_\ell^c \right]
$$

$$
\leq 2d \log \left( \frac{769d}{c^2} \right) + \frac{4d}{T} \cdot \log \left( 1 + \frac{T}{d\lambda} \right)
$$

$$
\leq 2d \log \left( \frac{769d}{c^2} \right) + \frac{4d \log(2T)}{T} . \tag{8}
$$

The second inequality follows from the definition of $E_\ell$ and its high-probability guarantee. The third inequality uses $c \leq 1$ and replaces the determinant by the bound $\det(H) \leq (\operatorname{tr}(H)/d)^d$, which is a consequence of the AM–GM inequality. The fourth inequality applies the fact that $\|x\|_2 \leq 1$ for all contexts, which implies $\operatorname{tr}(H_{\mathcal{T}''_{\ell-1}}) \leq d\lambda + T$.

Now, we establish an upper bound on the cumulative expected regret. By combining the bounds derived in (3) and (4), we can bound the cumulative expected regret for batch $\ell$, denoted by $\text{Regret}_\ell$, for any $\ell \geq 2$ as follows

$$
\text{Regret}_\ell = \sum_{t=\mathcal{T}_{\ell-1}+1}^{\mathcal{T}_\ell} \mathbb{E}[\langle x_t^* - x_{t,a_t}, \theta^* \rangle]
$$

$$
\leq 4 \sum_{t=\mathcal{T}_{\ell-1}+1}^{\mathcal{T}_\ell} \mathbb{E}[\varepsilon_{t,\ell-1}]
$$

$$
= 4 \sum_{t=\mathcal{T}_{\ell-1}+1}^{\mathcal{T}_\ell} \mathbb{E}\left[ \max_{y \in \mathcal{A}_t^{(\ell-2)}} \|y\|_{V_{\ell-1}^{-1}} \cdot \left( \beta_{t,\ell-1}^{(1)}\left(\frac{2}{T}\right) \wedge \beta_{t,\ell-1}^{(2)}\left(\frac{2}{T}\right) \right) \right]
$$

$$
\leq 4 \sum_{t=\mathcal{T}_{\ell-1}+1}^{\mathcal{T}_\ell} \mathbb{E}\left[ \max_{y \in \mathcal{A}_t^{(\ell-2)}} \|y\|_{V_{\ell-1}^{-1}} \right] \cdot \left( \sqrt{2\log\left(K(B-1)T^2\right)} + \sqrt{\lambda} \bigwedge 2\sqrt{\log\left(\frac{2^{6d-5}\pi d(B-1)^2 T^2}{15^{d-1}}\right)} + 2\sqrt{\lambda} \right) .
$$

The last inequality follows from the fact that the size of the candidate arm set satisfies $|\mathcal{A}_t^{(\ell-2)}| \leq |\mathcal{A}_t| = K$ for all $t \in [\mathcal{T}_{\ell-1} + 1, \mathcal{T}_\ell]$, allowing us to upper bound the confidence parameter $\beta_{t,\ell-1}^{(1)}$ uniformly over the action set.

Using (5), (6), and (8), we can bound the summation $\sum_{t=\mathcal{T}_{\ell-1}+1}^{\mathcal{T}_\ell} \mathbb{E}\left[ \max_{y \in \mathcal{A}_t^{(\ell-2)}} \|y\|_{V_{\ell-1}^{-1}} \right]$ for any $\ell \geq 2$ as

$$
\sum_{t=\mathcal{T}_{\ell-1}+1}^{\mathcal{T}_\ell} \mathbb{E}\left[ \max_{y \in \mathcal{A}_t^{(\ell-2)}} \|y\|_{V_{\ell-1}^{-1}} \right] \leq \sum_{t=\mathcal{T}_{\ell-1}+1}^{\mathcal{T}_\ell} \sqrt{\mathbb{E}\left[ \max_{y \in \mathcal{A}_t^{(\ell-2)}} y^\top V_{\ell-1}^{-1} y \right]}
$$

$$
= (\mathcal{T}_\ell - \mathcal{T}_{\ell-1}) \cdot \sqrt{\mathbb{E}\left[ \max_{y \in \mathcal{A}_t^{(\ell-2)}} y^\top V_{\ell-1}^{-1} y \right]}
$$

$$
\leq \frac{\mathcal{T}_\ell - \mathcal{T}_{\ell-1}}{\sqrt{\mathcal{T}''_{\ell-2} - \mathcal{T}'_{\ell-2}}} \cdot \sqrt{\sum_{s=\mathcal{T}'_{\ell-2}+1}^{\mathcal{T}''_{\ell-2}} \mathbb{E}\left[ \max_{y \in \mathcal{A}_s^{(\ell-2)}} y^\top H_{s-1}^{-1} y \right]}
$$

$$
\leq \frac{\mathcal{T}_\ell - \mathcal{T}_{\ell-1}}{\sqrt{\mathcal{T}''_{\ell-2} - \mathcal{T}'_{\ell-2}}} \cdot \sqrt{2d\log\left(\frac{769d}{c^2}\right) + \frac{4d\log(2T)}{T}}
$$

$$
\leq \frac{\frac{T^{1-2^{-\ell}}}{\log_2\log_2 T} + 3}{\sqrt{\frac{c(1-c)T^{1-2^{1-\ell}}}{\log_2\log_2 T}}} \cdot \sqrt{2d\log\left(\frac{769d}{c^2}\right) + \frac{4d\log(2T)}{T}}
$$

$$
= \left( \sqrt{\frac{T}{c(1-c)\log_2\log_2 T}} + 3\sqrt{\frac{\log_2\log_2 T}{c(1-c)T^{1-2^{1-\ell}}}} \right) \cdot \sqrt{2d\log\left(\frac{769d}{c^2}\right) + \frac{4d\log(2T)}{T}}
$$

$$
\leq \left( \sqrt{\frac{T}{c(1-c)\log_2\log_2 T}} + \frac{3}{\sqrt{c(1-c)}} \right) \cdot \sqrt{2d\log d + 2\left(\log\left(\frac{769}{c^2}\right) + 2\right)d}
$$

$$
\leq c' \cdot \sqrt{\frac{dT\log d}{\log_2\log_2 T}} ,
$$

where $t$ is arbitrary in the interval $[\mathcal{T}_{\ell-1} + 1, \mathcal{T}_\ell]$, and we define $c' := 4\sqrt{\frac{2(\log(769/c^2)+3)}{c(1-c)}}$. The first inequality applies Jensen's inequality to move the square root outside the expectation. The first equality follows from (5), while the second inequality uses the bound in (6). The third inequality follows from (8). The fifth inequality follows from the facts that $\log_2 \log_2 T \leq T^{1-2^{1-\ell}}$ and $\log(2T) \leq T$ for all $T \geq 1$. The final inequality uses the bounds $1 \leq \frac{T}{\log_2 \log_2 T}$ and $d \leq d \log d$ for all $T \geq 1$ and $d \geq 3$.

We now derive the cumulative expected regret after the first batch. Based on the previously derived results, the total regret from batches $\ell = 2$ to $B$ can be bounded as follows

$$
\sum_{\ell=2}^{B} \text{Regret}_\ell \leq \sum_{\ell=2}^{B} 4c' \sqrt{\frac{dT \log d}{\log_2 \log_2 T}} \cdot \left( \sqrt{2 \log\left(K(B-1)T^2\right)} + \sqrt{\lambda} \bigwedge 2\sqrt{\log\left(\frac{2^{6d-5}\pi d(B-1)^2 T^2}{15^{d-1}}\right)} + 2\sqrt{\lambda} \right)
$$

$$
= \mathcal{O}\left( \sqrt{dT \log d \log \log T} \cdot \left( \sqrt{\log(KT)} + \sqrt{\log(dT)} \wedge \sqrt{d \log\left(\frac{64}{15}\right) + \log(dT) + \sqrt{\log(dT)}} \right) \right)
$$

$$
= \mathcal{O}\left( \sqrt{dT \log d \log \log T} \cdot \left( \sqrt{\log(KT)} + \sqrt{\log T} \wedge \sqrt{d + \log T} + \sqrt{\log T} \right) \right)
$$

$$
= \mathcal{O}\left( \sqrt{dT \log d \log \log T} \cdot \left( \sqrt{\log(KT)} \wedge \sqrt{d + \log T} \right) \right).
$$

The first equality follows from substituting $B = \mathcal{O}(\log \log T)$ and $\lambda = \log(dT)$. The second equality holds in the regime where $T \geq d$. This is a safe assumption, as for $T \leq d$, the total regret is trivially bounded by $\mathcal{R}(T) = \sum_{t=1}^{T} \mathbb{E}[\langle x_t^* - x_{t,a_t}, \theta^* \rangle] \leq 2T \leq 2\sqrt{dT} = \mathcal{O}(\sqrt{dT})$, which is a much smaller bound.

Thus, the total worst-case regret is bounded as

$$
\mathcal{R}(T) \leq \sum_{k=1}^{B} \sum_{t=\mathcal{T}_{k-1}+1}^{\mathcal{T}_k} \mathbb{E}\left[ \langle x_t^* - x_{t,a_t}, \theta^* \rangle \Big| E \right]
$$

$$
= \text{Regret}_1 + \sum_{k=2}^{B} \text{Regret}_k
$$

$$
= \mathcal{O}(\sqrt{T}) + \mathcal{O}\left( \sqrt{dT \log d \log \log T} \cdot \left( \sqrt{\log(KT)} \wedge \sqrt{d + \log T} \right) \right)
$$

$$
= \mathcal{O}\left( \sqrt{dT} \left( \sqrt{\log(KT)} \wedge \sqrt{d + \log T} \right) \sqrt{\log d \log \log T} \right).
$$

Therefore, the worst-case regret for the algorithm is given by

$$
\mathcal{R}(T) = \mathcal{O}\left( \sqrt{dT} \left( \sqrt{\log(KT)} \wedge \sqrt{d + \log T} \right) \sqrt{\log d \log \log T} \right) = \tilde{\mathcal{O}}\left( \sqrt{dT \log K} \wedge d\sqrt{T} \right).
$$

$\square$

# B    PROOF OF THEOREM 2

*Proof of Theorem 2.* As Remark 1 encompasses the standard i.i.d. context assumption, we establish Theorem 2 under the more general conditions specified therein.

To establish the claimed batch complexity bound, we analyze the lower bound on the length of the $\ell$-th batch, where $\ell = \lceil \log_2 \log_2 T \rceil$. By definition of the schedule, the length of this batch is at least

$$\left\lceil \frac{T^{1-2^{-\ell}}}{\log_2 \log_2 T} \right\rceil + 1 \geq \frac{T^{1-2^{-\lceil \log_2 \log_2 T \rceil}}}{\log_2 \log_2 T} \geq \frac{T^{1-2^{-\log_2 \log_2 T}}}{\log_2 \log_2 T} = \frac{T^{1-\frac{1}{\log_2 T}}}{\log_2 \log_2 T} = \frac{T}{2 \log_2 \log_2 T} \ .$$

Since the batch length is non-decreasing in $\ell$, every batch with index $\ell \geq \lceil \log_2 \log_2 T \rceil$ has length at least $\frac{T}{2 \log_2 \log_2 T}$. Given the total time horizon $T$, the number of such batches is therefore at most $\lceil 2 \log_2 \log_2 T \rceil$. Including the initial $\lceil \log_2 \log_2 T \rceil - 1$ batches, the total number of batches $B$ is bounded by

$$B \leq \lceil 2 \log_2 \log_2 T \rceil + \lceil \log_2 \log_2 T \rceil - 1 \leq 3 \log_2 \log_2 T + 1 \ ,$$

which implies that the batch complexity is $\mathcal{O}(\log \log T)$.

Next, we begin by decomposing the cumulative expected regret based on the good event $E$. The regret can be written as

$$\mathcal{R}(T) = \sum_{k=1}^{B} \sum_{t=\mathcal{T}_{k-1}+1}^{\mathcal{T}_k} \mathbb{E}\left[ \max_{x \in \mathcal{A}_t} \langle x, \theta^* \rangle - \langle x_{t,a_t}, \theta^* \rangle \right]$$

$$= \sum_{k=1}^{B} \sum_{t=\mathcal{T}_{k-1}+1}^{\mathcal{T}_k} \mathbb{E}\left[ \langle x_t^* - x_{t,a_t}, \theta^* \rangle \Big| E^c \right] \cdot \mathbb{P}(E^c) + \mathbb{E}\left[ \langle x_t^* - x_{t,a_t}, \theta^* \rangle \Big| E \right] \cdot \mathbb{P}(E) \ .$$

Using the triangle inequality followed by the Cauchy-Schwarz inequality, we have

$$\begin{aligned} \langle x_t^* - x_{t,a_t}, \theta^* \rangle &\leq |\langle x_t^* - x_{t,a_t}, \theta^* \rangle| \\ &\leq |\langle x_t^*, \theta^* \rangle| + |\langle x_{t,a_t}, \theta^* \rangle| \\ &\leq \|x_t^*\| \cdot \|\theta^*\| + \|x_{t,a_t}\| \cdot \|\theta^*\| \\ &\leq 2 \ , \end{aligned}$$

where the last inequality follows from Assumption 1. Hence, the cumulative expected regret can be bounded as

$$\mathcal{R}(T) \leq 2T \cdot \mathbb{P}(E^c) + \sum_{k=1}^{B} \sum_{t=\mathcal{T}_{k-1}+1}^{\mathcal{T}_k} \mathbb{E}\left[ \langle x_t^* - x_{t,a_t}, \theta^* \rangle \Big| E \right] \cdot \mathbb{P}(E)$$

$$\leq \mathcal{O}(1) + \sum_{k=1}^{B} \sum_{t=\mathcal{T}_{k-1}+1}^{\mathcal{T}_k} \mathbb{E}\left[ \langle x_t^* - x_{t,a_t}, \theta^* \rangle \Big| E \right] \ ,$$

where the $\mathcal{O}(1)$ term follows from the high-probability guarantee $\mathbb{P}(E^c) = \mathcal{O}(1/T)$. Throughout the remainder of the analysis, we therefore condition on the good event $E$. Let $\text{Regret}_\ell$ denote the cumulative expected regret incurred during batch $\ell$. We analyze the regret separately for the case $\ell = 1$ and for all subsequent batches $\ell \geq 2$.

**Case 1: $\ell = 1$.**    In the first batch, the number of rounds is $\mathcal{T}_1 = \left\lceil \frac{\sqrt{T}}{\log_2 \log_2 T} \right\rceil$, and the instantaneous regret is bounded by 2. Therefore,

$$\text{Regret}_1 = \sum_{t=1}^{\mathcal{T}_1} \mathbb{E}[\langle x_t^* - x_{t,a_t}, \theta^* \rangle] \leq 2\mathcal{T}_1 \leq 2\left( \frac{\sqrt{T}}{\log_2 \log_2 T} + 1 \right) = \mathcal{O}(\sqrt{T}) \ .$$

**Case 2: $\ell \geq 2$.** For each batch $\ell \geq 2$, the rounds following the arm elimination steps are divided into two phases:

**Phase 1.** In the first phase of length $\left\lceil \frac{cT^{1-2^{-\ell}}}{\log_2 \log_2 T} \right\rceil$, the algorithm selects the most informative direction with respect to the current Gram matrix. For any round $t$ in these phases, the instantaneous regret satisfies

$$
\begin{aligned}
\langle x_t^* - x_{t,a_t}, \theta^* \rangle &= \langle x_t^* - x_{t,a_t}, \theta^* - \hat{\theta}_{\ell-1} \rangle + \langle x_t^* - x_{t,a_t}, \hat{\theta}_{\ell-1} \rangle \\
&\leq \langle x_t^* - x_{t,a_t}, \theta^* - \hat{\theta}_{\ell-1} \rangle + \langle x_t^{(\ell-1)} - x_{t,a_t}, \hat{\theta}_{\ell-1} \rangle \\
&\leq |\langle x_t^*, \hat{\theta}_{\ell-1} - \theta^* \rangle| + |\langle x_{t,a_t}, \hat{\theta}_{\ell-1} - \theta^* \rangle| + 2\varepsilon_{t,\ell-1} \\
&\leq 4\varepsilon_{t,\ell-1} .
\end{aligned}
\tag{9}
$$

The first inequality uses the fact that $x_t^{(\ell-1)}$ is optimal with respect to $\hat{\theta}_{\ell-1}$ and that $x_t^*$ belongs to $\mathcal{A}_t^{(\ell-2)}$ by Lemma 4. The second inequality follows since $x_{t,a_t} \in \mathcal{A}_t^{(\ell-1)}$ by construction of the arm elimination step. The final inequality follows from the definition of the good event $E$ and again from Lemma 4, which guarantees $x_t^* \in \mathcal{A}_t^{(\ell-2)}$.

**Phase 2.** In the last phase, the algorithm selects arms greedily with respect to the estimated parameter, i.e., $x_{t,a_t} \in \arg\max_{x \in \mathcal{A}_t^{(\ell-1)}} \langle x, \hat{\theta}_{\ell-1} \rangle$. For any round $t$ in this phase, the instantaneous regret satisfies

$$
\begin{aligned}
\langle x_t^* - x_{t,a_t}, \theta^* \rangle &= \langle x_t^* - x_{t,a_t}, \theta^* - \hat{\theta}_{\ell-1} \rangle + \langle x_t^* - x_{t,a_t}, \hat{\theta}_{\ell-1} \rangle \\
&\leq \langle x_t^* - x_{t,a_t}, \theta^* - \hat{\theta}_{\ell-1} \rangle \\
&\leq |\langle x_t^*, \hat{\theta}_{\ell-1} - \theta^* \rangle| + |\langle x_{t,a_t}, \hat{\theta}_{\ell-1} - \theta^* \rangle| \\
&\leq 2\varepsilon_{t,\ell-1} .
\end{aligned}
\tag{10}
$$

The first inequality follows from the fact that $x_{t,a_t}$ is optimal with respect to $\hat{\theta}_{\ell-1}$ and that $x_t^*$ belongs to $\mathcal{A}_t^{(\ell-1)}$ by Lemma 4. The final inequality follows directly from the definition of the good event $E$ and from Lemma 4, as in Phase 1.

Therefore, it suffices to upper bound the quantity $\sum_{t=\mathcal{T}_\ell+1}^{\mathcal{T}_{\ell+1}} \varepsilon_{t,\ell}$ for each $\ell \in [B-1]$, which reduces to bounding the sum $\sum_{t=\mathcal{T}_\ell+1}^{\mathcal{T}_{\ell+1}} \max_{y \in \mathcal{A}_t^{(\ell-1)}} \|y\|_{V_\ell^{-1}}$. Since both $V_\ell$ and the arm elimination rule—determined by $\hat{\theta}_1, \ldots, \hat{\theta}_{\ell-1}$—are measurable with respect to $\mathcal{F}_{\mathcal{T}_\ell}$, they can be treated as fixed quantities conditional on this filtration. Moreover, conditional on $\mathcal{F}_{\mathcal{T}_\ell}$ the action sets $\{\mathcal{A}_s\}_{s=\mathcal{T}_\ell+1}^{\mathcal{T}_{\ell+2}}$ are *identically distributed* with common law $\mathcal{D}_\ell$. Hence, by the tower property, for any $t, v \in [\mathcal{T}_\ell + 1, \mathcal{T}_{\ell+1}]$ we have

$$
\begin{aligned}
\mathbb{E}\left[ \max_{y \in \mathcal{A}_t^{(\ell-1)}} y^\top V_\ell^{-1} y \right] &= \mathbb{E}\left[ \mathbb{E}\left[ \max_{y \in \mathcal{A}_t^{(\ell-1)}} y^\top V_\ell^{-1} y \,\Big|\, \mathcal{F}_{\mathcal{T}_\ell} \right] \right] \\
&= \mathbb{E}\left[ \mathbb{E}\left[ \max_{y \in \mathcal{A}_v^{(\ell-1)}} y^\top V_\ell^{-1} y \,\Big|\, \mathcal{F}_{\mathcal{T}_\ell} \right] \right] \\
&= \mathbb{E}\left[ \max_{y \in \mathcal{A}_v^{(\ell-1)}} y^\top V_\ell^{-1} y \right] .
\end{aligned}
\tag{11}
$$

Define $\mathcal{T}'_{\ell-1} := \mathcal{T}_{\ell-1} + \left\lceil \frac{cT^{1-2^{-\ell}}}{\log_2 \log_2 T} \right\rceil$ for all $\ell \geq 2$, while for the first batch we set $\mathcal{T}'_0 := \mathcal{T}_1$. Now fix any $t \in [\mathcal{T}_\ell + 1, \mathcal{T}_{\ell+1}]$ and consider the interval $s \in [\mathcal{T}_{\ell-1} + 1, \mathcal{T}'_{\ell-1}]$. Then we obtain

$$
\begin{aligned}
\sum_{s=\mathcal{T}_{\ell-1}+1}^{\mathcal{T}'_{\ell-1}} \mathbb{E}\left[ \max_{y \in \mathcal{A}_t^{(\ell-1)}} y^\top V_\ell^{-1} y \right] &\leq \sum_{s=\mathcal{T}_{\ell-1}+1}^{\mathcal{T}'_{\ell-1}} \mathbb{E}\left[ \max_{y \in \mathcal{A}_t^{(\ell-1)}} y^\top H_{s-1}^{-1} y \right] \\
&= \sum_{s=\mathcal{T}_{\ell-1}+1}^{\mathcal{T}'_{\ell-1}} \mathbb{E}\left[ \mathbb{E}\left[ \max_{y \in \mathcal{A}_t^{(\ell-1)}} y^\top H_{s-1}^{-1} y \,\Big|\, \mathcal{F}_{\mathcal{T}_{\ell-1}} \vee \sigma(H_{s-1}) \right] \right]
\end{aligned}
$$

$$= \sum_{s=\mathcal{T}_{\ell-1}+1}^{\mathcal{T}'_{\ell-1}} \mathbb{E}\left[\mathbb{E}\left[\max_{y\in\mathcal{A}_s^{(\ell-1)}} y^\top H_{s-1}^{-1}y \,\Big|\, \mathcal{F}_{\mathcal{T}_{\ell-1}} \vee \sigma(H_{s-1})\right]\right]$$

$$= \sum_{s=\mathcal{T}_{\ell-1}+1}^{\mathcal{T}'_{\ell-1}} \mathbb{E}\left[\max_{y\in\mathcal{A}_s^{(\ell-1)}} y^\top H_{s-1}^{-1}y\right] . \tag{12}$$

The first inequality follows from monotonicity of the matrices, since $H_{s-1} \preceq V_\ell$ for all $s$ in the interval. The first and last equalities follow from the tower property. The second equality relies on the fact that, conditional on $\mathcal{F}_{\mathcal{T}_{\ell-1}} \vee \sigma(H_{s-1})$, both $H_{s-1}$ and the arm elimination rule (determined by $\hat\theta_1,\ldots,\hat\theta_{\ell-1}$) are fixed. Moreover, conditional on $\mathcal{F}_{\mathcal{T}_{\ell-1}}$, the action set $\mathcal{A}_t$ is independent of $\{\mathcal{A}_v, x_{v,a_v}, r_v\}_{v=\mathcal{T}_{\ell-1}+1}^{\mathcal{T}_\ell}$ for $t \in [\mathcal{T}_\ell+1, \mathcal{T}_{\ell+1}]$, and $\mathcal{A}_s$ is independent of $\{\mathcal{A}_u, x_{u,a_u}, r_u\}_{u=\mathcal{T}_{\ell-1}+1}^{s-1}$ for $s \in [\mathcal{T}_{\ell-1}+1, \mathcal{T}_\ell]$. Hence, conditional on $\mathcal{F}_{\mathcal{T}_{\ell-1}} \vee \sigma(H_{s-1})$, both $\mathcal{A}_t$ and $\mathcal{A}_s$ share the same conditional law $\mathcal{D}_{\ell-1}$, which justifies replacing $\mathcal{A}_t$ with $\mathcal{A}_s$ in the inner expectation.

In Phase 1, the algorithm proceeds by selecting the most informative direction at each step with respect to the current Gram matrix. For all $\ell \geq 1$, we obtain

$$\sum_{s=\mathcal{T}_{\ell-1}+1}^{\mathcal{T}'_{\ell-1}} \mathbb{E}\left[\max_{y\in\mathcal{A}_s^{(\ell-1)}} y^\top H_{s-1}^{-1}y\right] = \mathbb{E}\left[\sum_{s=\mathcal{T}_{\ell-1}+1}^{\mathcal{T}'_{\ell-1}} \max_{y\in\mathcal{A}_s^{(\ell-1)}} y^\top H_{s-1}^{-1}y\right]$$

$$= \mathbb{E}\left[\sum_{s=\mathcal{T}_{\ell-1}+1}^{\mathcal{T}'_{\ell-1}} x_{s,a_s}^\top H_{s-1}^{-1} x_{s,a_s}\right]$$

$$\leq \mathbb{E}\left[2\log\left(\frac{\det(H_{\mathcal{T}'_{\ell-1}})}{\det(\lambda I)}\right)\right]$$

$$\leq \mathbb{E}\left[2\log\left(\frac{(\mathrm{tr}(H_{\mathcal{T}'_{\ell-1}})/d)^d}{\lambda^d}\right)\right]$$

$$\leq 2d\log\left(1 + \frac{T}{d\lambda}\right)$$

$$\leq 2d\log(2T) , \tag{13}$$

where the second equality follows from the arm-selection strategy of Phase 1, and the first inequality follows from Corollary 2. The second inequality replaces the determinant by the bound $\det(H) \leq (\mathrm{tr}(H)/d)^d$, which is a consequence of the AM–GM inequality. The third inequality applies the fact that $\|x\|_2 \leq 1$ for all contexts, which implies $\mathrm{tr}(H_{\mathcal{T}'_{\ell-1}}) \leq d\lambda + T$.

Now, we establish an upper bound on the cumulative expected regret. By combining the bounds derived in (9) and (10), we can bound the cumulative expected regret for batch $\ell$, denoted by $\mathrm{Regret}_\ell$, for any $\ell \geq 2$ as follows

$$\mathrm{Regret}_\ell = \sum_{t=\mathcal{T}_{\ell-1}+1}^{\mathcal{T}_\ell} \mathbb{E}[\langle x_t^* - x_{t,a_t}, \theta^*\rangle]$$

$$\leq 4 \sum_{t=\mathcal{T}_{\ell-1}+1}^{\mathcal{T}_\ell} \mathbb{E}[\varepsilon_{t,\ell-1}]$$

$$= 4 \sum_{t=\mathcal{T}_{\ell-1}+1}^{\mathcal{T}_\ell} \mathbb{E}\left[\max_{y\in\mathcal{A}_t^{(\ell-2)}} \|y\|_{V_{\ell-1}^{-1}} \cdot \left(\beta_{t,\ell-1}^{(1)}\left(\frac{2}{T}\right) \wedge \beta_{t,\ell-1}^{(2)}\left(\frac{2}{T}\right)\right)\right]$$

$$\leq 4 \sum_{t=\mathcal{T}_{\ell-1}+1}^{\mathcal{T}_\ell} \mathbb{E}\left[\max_{y\in\mathcal{A}_t^{(\ell-2)}} \|y\|_{V_{\ell-1}^{-1}}\right] \cdot \left(\sqrt{2\log\left(K(B-1)T^2\right)} + \sqrt{\lambda} \bigwedge 2\sqrt{\log\left(\frac{2^{6d-5}\pi d(B-1)^2 T^2}{15^{d-1}}\right)} + 2\sqrt{\lambda}\right) .$$

The last inequality follows from the fact that the size of the candidate arm set satisfies $|\mathcal{A}_t^{(\ell-2)}| \leq |\mathcal{A}_t| = K$ for all $t \in [\mathcal{T}_{\ell-1}+1, \mathcal{T}_\ell]$, allowing us to upper bound the confidence

parameter $\beta_{t,\ell-1}^{(1)}$ uniformly over the action set.

Using (11), (12), and (13), we can bound the summation $\sum_{t=\mathcal{T}_{\ell-1}+1}^{\mathcal{T}_\ell} \mathbb{E}\left[\max_{y \in \mathcal{A}_t^{(\ell-2)}} \|y\|_{V_{\ell-1}^{-1}}\right]$ for any $\ell \geq 2$ as

$$
\sum_{t=\mathcal{T}_{\ell-1}+1}^{\mathcal{T}_\ell} \mathbb{E}\left[\max_{y \in \mathcal{A}_t^{(\ell-2)}} \|y\|_{V_{\ell-1}^{-1}}\right] \leq \sum_{t=\mathcal{T}_{\ell-1}+1}^{\mathcal{T}_\ell} \sqrt{\mathbb{E}\left[\max_{y \in \mathcal{A}_t^{(\ell-2)}} y^\top V_{\ell-1}^{-1} y\right]}
$$

$$
= (\mathcal{T}_\ell - \mathcal{T}_{\ell-1}) \cdot \sqrt{\mathbb{E}\left[\max_{y \in \mathcal{A}_t^{(\ell-2)}} y^\top V_{\ell-1}^{-1} y\right]}
$$

$$
\leq \frac{\mathcal{T}_\ell - \mathcal{T}_{\ell-1}}{\sqrt{\mathcal{T}_{\ell-2}' - \mathcal{T}_{\ell-2}}} \cdot \sqrt{\sum_{s=\mathcal{T}_{\ell-2}+1}^{\mathcal{T}_{\ell-2}'} \mathbb{E}\left[\max_{y \in \mathcal{A}_s^{(\ell-2)}} y^\top H_{s-1}^{-1} y\right]}
$$

$$
\leq \frac{\mathcal{T}_\ell - \mathcal{T}_{\ell-1}}{\sqrt{\mathcal{T}_{\ell-2}' - \mathcal{T}_{\ell-2}}} \cdot \sqrt{2d \log(2T)}
$$

$$
\leq \frac{\frac{T^{1-2^{-\ell}}}{\log_2 \log_2 T} + 2}{\sqrt{\frac{cT^{1-2^{1-\ell}}}{\log_2 \log_2 T}}} \cdot \sqrt{2d \log(2T)}
$$

$$
= \left(\sqrt{\frac{T}{c \log_2 \log_2 T}} + 2\sqrt{\frac{\log_2 \log_2 T}{cT^{1-2^{1-\ell}}}}\right) \cdot \sqrt{2d \log(2T)}
$$

$$
\leq \left(\sqrt{\frac{T}{c \log_2 \log_2 T}} + \frac{2}{\sqrt{c}}\right) \cdot 2\sqrt{d \log T}
$$

$$
\leq c' \cdot \sqrt{\frac{dT \log T}{\log_2 \log_2 T}},
$$

where $t$ is arbitrary in the interval $[\mathcal{T}_{\ell-1}+1, \mathcal{T}_\ell]$, and we define $c' := \frac{6}{\sqrt{c}}$. The first inequality applies Jensen's inequality to move the square root outside the expectation. The first equality follows from (11), while the second inequality uses the bound in (12). The third inequality follows from (13). The fifth inequality follows from the facts that $\log_2 \log_2 T \leq T^{1-2^{1-\ell}}$ and $\log(2T) \leq 2 \log T$ for all $T \geq 2$. The final inequality uses the bounds $1 \leq \frac{T}{\log_2 \log_2 T}$ for all $T \geq 1$.

We now derive the cumulative expected regret after the first batch. Building on the previously established results, the total regret incurred from batches $\ell = 2$ to $B$ can be bounded as

$$
\sum_{\ell=2}^{B} \text{Regret}_\ell \leq \sum_{\ell=2}^{B} 4c' \sqrt{\frac{dT \log T}{\log_2 \log_2 T}} \cdot \left(\sqrt{2 \log(K(B-1)T^2)} + \sqrt{\lambda} \bigwedge 2\sqrt{\log\left(\frac{2^{6d-5}\pi d(B-1)^2 T^2}{15^{d-1}}\right)} + 2\sqrt{\lambda}\right)
$$

$$
= \mathcal{O}\left(\sqrt{dT \log T \log \log T} \cdot \left(\sqrt{\log(KT)} \wedge \sqrt{d \log\left(\frac{64}{15}\right) + \log(dT)}\right)\right)
$$

$$
= \mathcal{O}\left(\sqrt{dT \log T \log \log T} \cdot \left(\sqrt{\log(KT)} \wedge \sqrt{d + \log T}\right)\right).
$$

Here, the first equality follows from substituting $B = \mathcal{O}(\log \log T)$ and $\lambda = \mathcal{O}(1)$, while the second equality holds because the $\log(d)$ term is dominated by $d$.

Thus, the total worst-case regret is bounded as

$$
\mathcal{R}(T) \leq \sum_{k=1}^{B} \sum_{t=\mathcal{T}_{k-1}+1}^{\mathcal{T}_k} \mathbb{E}\left[\langle x_t^* - x_{t,a_t}, \theta^*\rangle \Big| E\right]
$$

$$= \text{Regret}_1 + \sum_{k=2}^{B} \text{Regret}_k$$

$$= \mathcal{O}(\sqrt{T}) + \mathcal{O}\left(\sqrt{dT \log T \log \log T} \cdot \left(\sqrt{\log(KT)} \wedge \sqrt{d + \log T}\right)\right)$$

$$= \mathcal{O}\left(\sqrt{dT}\left(\sqrt{\log(KT)} \wedge \sqrt{d + \log T}\right)\sqrt{\log T \log \log T}\right) .$$

Therefore, the worst-case regret for the algorithm is given by

$$\mathcal{R}(T) = \mathcal{O}\left(\sqrt{dT}\left(\sqrt{\log(KT)} \wedge \sqrt{d + \log T}\right)\sqrt{\log T \log \log T}\right) = \tilde{\mathcal{O}}\left(\sqrt{dT \log K} \wedge d\sqrt{T}\right) .$$

$\square$

---

**Algorithm 3** BGLE

---

1: **Input:** Horizon $T$; batch end times $\mathcal{T}_\ell = \ell \left\lceil \frac{\sqrt[3]{T}}{\log_2 \log_2 T} \right\rceil$ for $\ell \leq 3$, $\mathcal{T}_\ell = \left( \mathcal{T}_{\ell-1} + \left\lceil \frac{T^{1-\frac{1}{3 \cdot 2^{\ell-4}}}}{\log_2 \log_2 T} \right\rceil \right) \wedge$
$\quad T$ for $\ell \geq 4$; number of batches $B$, with $\mathcal{T}_B = T$; within-batch allocation rate $c$;

---

2: **Initialize:** $\lambda \leftarrow R^2(d + \log T)$, $H_0 \leftarrow \lambda I$;
3: **for** $t \leftarrow 1, 2, \ldots, \mathcal{T}_1$ **do**
4: $\quad$ Pull arm $x_{t,a_t} \in \arg\max_{x \in \mathcal{A}_t} \|x\|_{H_{t-1}^{-1}}$, and receive reward $r_t$;
5: $\quad$ $H_t \leftarrow H_{t-1} + x_{t,a_t} x_{t,a_t}^\top$;
6: $V_1 \leftarrow H_{\mathcal{T}_1}, \hat{\theta}_1 \leftarrow \arg\min_\theta \sum_{t=1}^{\mathcal{T}_1} \ell_t(\theta), H_{\mathcal{T}_1} \leftarrow \lambda I$;
7: **for** $\ell \leftarrow 2, \ldots, B$ **do**
8: $\quad$ **for** $t \leftarrow \mathcal{T}_{\ell-1} + 1, \ldots, \mathcal{T}_\ell$ **do**
9: $\quad\quad$ **if** $\ell \geq 3$ **then**
10: $\quad\quad\quad$ **for** $k \leftarrow 2, \ldots, \ell - 1$ **do**
11: $\quad\quad\quad\quad$ $x_t^{(k)} \leftarrow \arg\max_{x \in \mathcal{A}_t^{(k-1)}} \langle x, \hat{\theta}_k \rangle$;
12: $\quad\quad\quad\quad$ $\mathcal{A}_t^{(k)} \leftarrow \left\{ x \in \mathcal{A}_t^{(k-1)} \,\middle|\, \langle \hat{\theta}_k, x_t^{(k)} - x \rangle \leq 2\varepsilon'_{t,k}(\lambda) \right\}$;
13: $\quad\quad$ **if** $t \leq \mathcal{T}_{\ell-1} + \left\lceil cT^{1-2^{((4-\ell)\wedge 1)}/3} / \log_2 \log_2 T \right\rceil$ **then**
14: $\quad\quad\quad$ Pull arm $x_{t,a_t} \in \arg\max_{x \in \mathcal{A}_t^{(\ell-1)}} \|x\|_{H_{t-1}^{-1}}$, and receive reward $r_t$;
15: $\quad\quad$ **else**
16: $\quad\quad\quad$ Pull arm $x_{t,a_t} \in \arg\max_{x \in \mathcal{A}_t^{(\ell-1)}} \langle x, \hat{\theta}_{\ell-1} \rangle$, and receive reward $r_t$;
17: $\quad\quad$ $H_t \leftarrow H_{t-1} + \alpha_{t,\ell-1}(\lambda) \dot{\mu}(\langle x_{t,a_t}, \hat{\theta}_{\ell-1} \rangle) x_{t,a_t} x_{t,a_t}^\top$;
18: $\quad$ $V_\ell \leftarrow H_{\mathcal{T}_\ell}, \hat{\theta}_\ell \leftarrow \arg\min_\theta \sum_{t=\mathcal{T}_{\ell-1}+1}^{\mathcal{T}_\ell} \ell_t(\theta), H_{\mathcal{T}_\ell} \leftarrow \lambda I$;

## C    PROOF OF THEOREM 3

We begin by assuming that the MLE estimator $\hat{\theta}$, obtained by minimizing the log-loss objective, always satisfies the boundedness condition $\|\hat{\theta}\|_2 \leq S$. If this condition does not hold, one may instead apply the non-convex projection technique of Sawarni et al. (2024). The projected estimator preserves the same guarantees established in Sawarni et al. (2024), up to a multiplicative factor of 2. Therefore, the assumption $\|\hat{\theta}\|_2 \leq S$ can be made without loss of generality.

**Lemma 9.** *For any $x \in [0, C]$, the following inequality holds*

$$e^x \leq \frac{x(e^C - 1)}{C} + 1 \ .$$

*Proof.* Apply the definition of convexity, $f\big((1-\alpha)a + \alpha b\big) \leq (1-\alpha)f(a) + \alpha f(b)$, to $f(t) = e^t$ with $a = 0$, $b = C$, and $\alpha = x/C \in [0, 1]$. This gives

$$e^{(1-\alpha)0 + \alpha C} \leq (1-\alpha)e^0 + \alpha e^C \quad \Rightarrow \quad e^x \leq 1 + \tfrac{x}{C}(e^C - 1) \ ,$$

which is the claim. $\qquad\square$

**Lemma 10.** *For an exponential family distribution with log-partition function $m(\cdot)$, let $\mu(z) := m'(z)$. Then, for all $x_1, x_2 \in \mathbb{R}$, we have*

$$e^{-R|x_2 - x_1|} \dot{\mu}(x_2) \leq \dot{\mu}(x_1) \leq e^{R|x_2 - x_1|} \dot{\mu}(x_2) \ .$$

*Proof.* Without loss of generality, assume that $x_2 \geq x_1$. Define $h_1(x) := \dot{\mu}(x)e^{Rx}$ and $h_2(x) := \dot{\mu}(x)e^{-Rx}$. Differentiating these functions yields $h_1'(x) = (\ddot{\mu}(x) + R\dot{\mu}(x))e^{Rx}$ and $h_2'(x) = (\ddot{\mu}(x) - R\dot{\mu}(x))e^{-Rx}$. By the self-concordance property, we have $h_1'(x) \geq 0$ and $h_2'(x) \leq 0$, which implies that $h_1(x)$ is non-decreasing and $h_2(x)$ is non-increasing. Consequently, $h_1(x_2) \geq h_1(x_1)$ and $h_2(x_2) \leq h_2(x_1)$, which together establish the desired inequality. $\qquad\square$

**Lemma 11.** *(Sawarni et al., 2024) For each batch $\ell \geq 1$, let $r_{\mathcal{T}_{\ell-1}+1}, \ldots, r_{\mathcal{T}_\ell}$ denote independent random variables drawn from the canonical exponential family such that $\mathbb{E}[r_s] = \mu(\langle x_{s,a_s}, \theta^* \rangle)$ for some $\theta^* \in \mathbb{R}^d$. Define the maximum likelihood estimator by $\hat{\theta}_\ell = \arg\min_\theta \sum_{t=\mathcal{T}_{\ell-1}+1}^{\mathcal{T}_\ell} \ell_t(\theta)$, and let $V_\ell^* := \lambda I + \sum_{s=\mathcal{T}_{\ell-1}+1}^{\mathcal{T}_\ell} \dot{\mu}(\langle x_{s,a_s}, \theta^* \rangle) x_{s,a_s} x_{s,a_s}^\top$. Then, with probability at least $1 - \frac{1}{T^2}$, the following inequality holds*

$$\|\hat{\theta}_\ell - \theta^*\|_{V_\ell^*} \leq 24RS\left(\sqrt{d + \log T} + \frac{R(d + \log T)}{\sqrt{\lambda}}\right) + 2S\sqrt{\lambda} \triangleq \beta(\lambda) .$$

**Lemma 12.** *For any batch $\ell \geq 2$, define $V_\ell^* := \lambda I + \sum_{s=\mathcal{T}_{\ell-1}+1}^{\mathcal{T}_\ell} \dot{\mu}(\langle x_{s,a_s}, \theta^* \rangle) x_{s,a_s} x_{s,a_s}^\top$ and $V_\ell := \lambda I + \sum_{s=\mathcal{T}_{\ell-1}+1}^{\mathcal{T}_\ell} \alpha_{s,\ell-1}(\lambda) \dot{\mu}(\langle x_{s,a_s}, \hat{\theta}_{\ell-1} \rangle) x_{s,a_s} x_{s,a_s}^\top$. Then, for every $\ell \geq 2$, with probability at least $1 - \frac{1}{T^2}$, the following matrix inequality holds*

$$V_\ell \preceq V_\ell^* .$$

*Proof.* We first consider the case $\ell = 2$, applying Lemma 10 yields

$$e^{-R|\langle x_{s,a_s}, \hat{\theta}_1 - \theta^* \rangle|} \dot{\mu}(\langle x_{s,a_s}, \hat{\theta}_1 \rangle) \leq \dot{\mu}(\langle x_{s,a_s}, \theta^* \rangle) .$$

By the assumptions $\|x_{s,a_s}\|_2 \leq 1$, $\|\theta^*\|_2 \leq S$, and $\|\hat{\theta}_1\|_2 \leq S$, we further obtain

$$|\langle x_{s,a_s}, \hat{\theta}_1 - \theta^* \rangle| \leq \|x_{s,a_s}\|_2 \cdot \|\hat{\theta}_1 - \theta^*\|_2 \leq \|\hat{\theta}_1\|_2 + \|\theta^*\|_2 \leq 2S .$$

Consequently,

$$\begin{aligned}
\alpha_{s,1}(\lambda) \dot{\mu}(\langle x_{s,a_s}, \hat{\theta}_1 \rangle) &= e^{-2RS} \dot{\mu}(\langle x_{s,a_s}, \hat{\theta}_1 \rangle) \\
&\leq e^{-R|\langle x_{s,a_s}, \hat{\theta}_1 - \theta^* \rangle|} \dot{\mu}(\langle x_{s,a_s}, \hat{\theta}_1 \rangle) \\
&\leq \dot{\mu}(\langle x_{s,a_s}, \theta^* \rangle) ,
\end{aligned}$$

which establishes $V_2 \preceq V_2^*$.

For the general case $\ell \geq 3$, Lemma 10 gives

$$e^{-R|\langle x_{s,a_s}, \hat{\theta}_{\ell-1} - \theta^* \rangle|} \dot{\mu}(\langle x_{s,a_s}, \hat{\theta}_{\ell-1} \rangle) \leq \dot{\mu}(\langle x_{s,a_s}, \theta^* \rangle) .$$

Using Lemma 11 together with the assumptions $\|x_{s,a_s}\|_2 \leq 1$, $\|\theta^*\|_2 \leq S$ and $\|\hat{\theta}_{\ell-1}\|_2 \leq S$, we obtain

$$|\langle x_{s,a_s}, \hat{\theta}_{\ell-1} - \theta^* \rangle| \leq (2S \wedge \|x_{s,a_s}\|_{V_{\ell-1}^{*-1}} \beta(\lambda)) \leq (2S \wedge \|x_{s,a_s}\|_{V_{\ell-1}^{-1}} \beta(\lambda)) ,$$

where the second inequality follows inductively from $V_{\ell-1} \preceq V_{\ell-1}^*$. Therefore,

$$\begin{aligned}
\alpha_{s,\ell-1}(\lambda) \dot{\mu}(\langle x_{s,a_s}, \hat{\theta}_{\ell-1} \rangle) &= e^{-R(2S \wedge \|x_{s,a_s}\|_{V_{\ell-1}^{-1}} \beta(\lambda))} \dot{\mu}(\langle x_{s,a_s}, \hat{\theta}_{\ell-1} \rangle) \\
&\leq e^{-R|\langle x_{s,a_s}, \hat{\theta}_{\ell-1} - \theta^* \rangle|} \dot{\mu}(\langle x_{s,a_s}, \hat{\theta}_{\ell-1} \rangle) \\
&\leq \dot{\mu}(\langle x_{s,a_s}, \theta^* \rangle) ,
\end{aligned}$$

which completes the proof that $V_\ell \preceq V_\ell^*$ for all $\ell \geq 2$. $\square$

**Lemma 13** (Good event). *Define the following quantities:*

$$\beta(\lambda) := 24RS\left(\sqrt{d + \log T} + \frac{R(d + \log T)}{\sqrt{\lambda}}\right) + 2S\sqrt{\lambda},$$

$$\varepsilon'_{t,\ell}(\lambda) := \max_{y \in \mathcal{A}_t^{(\ell-1)}} \|y\|_{V_\ell^{-1}} \cdot \beta(\lambda) \text{ for } \ell \geq 2 .$$

*Then, with probability at least $1 - \frac{2(B-2)}{T^2}$, the following event holds*

$$E' := \bigcap_{\ell=2}^{B-1} \bigcap_{t=\mathcal{T}_\ell+1}^{T} \left\{ |\langle x, \hat{\theta}_\ell - \theta^* \rangle| \leq \varepsilon'_{t,\ell}(\lambda), \ \forall x \in \mathcal{A}_t^{(\ell-1)} \right\} .$$

*Proof.* For any batch $\ell \geq 2$, by the Cauchy-Schwarz inequality together with Lemma 11 and Lemma 12, it follows that for every round $t$ and for all $x \in \mathcal{A}_t^{(\ell-1)}$, with probability at least $1 - \frac{2}{T^2}$ we have

$$|\langle x, \hat{\theta}_\ell - \theta^* \rangle| \leq \|x\|_{V_\ell^{*-1}} \|\hat{\theta}_\ell - \theta^*\|_{V_\ell^*} \leq \|x\|_{V_\ell^{-1}} \cdot \beta(\lambda) \leq \max_{y \in \mathcal{A}_t^{(\ell-1)}} \|y\|_{V_\ell^{-1}} \cdot \beta(\lambda) .$$

Applying a union bound over all batches $\ell$ then guarantees that the event $E'$ holds with probability at least $1 - \frac{2(B-2)}{T^2}$, which completes the proof. $\qquad\square$

**Lemma 14.** *Let $E'$ be the good event defined in Lemma 13. Conditioned on $E'$, the optimal arm $x_t^* \in \arg\max_{x \in \mathcal{A}_t} \langle x, \theta^* \rangle$ is never eliminated at any round $t$. In particular,*

$$x_t^* \in \mathcal{A}_t^{(\ell)}, \quad \text{for all } 1 \leq \ell \leq B-1 \text{ and } \mathcal{T}_\ell + 1 \leq t \leq T .$$

*Proof.* Fix $t \in [\mathcal{T}_s + 1, \mathcal{T}_{s+1}]$ for some $s \in [B-1]$. We show by induction on $\ell$ that $x_t^* \in \mathcal{A}_t^{(\ell)}$ for all $\ell \in [s]$.

**Base case ($\ell = 1$).** By definition we have $\mathcal{A}_t = \mathcal{A}_t^{(0)} = \mathcal{A}_t^{(1)}$, which immediately implies that $x_t^* \in \mathcal{A}_t^{(1)}$ holds trivially.

**Inductive step.** Assume $x_t^* \in \mathcal{A}_t^{(\ell-1)}$ for some $\ell \in \{2, \ldots, s\}$. Since $x_t^{(\ell)} \in \mathcal{A}_t^{(\ell-1)}$, we similarly obtain

$$\langle \hat{\theta}_\ell, x_t^{(\ell)} - x_t^* \rangle = \langle \hat{\theta}_\ell - \theta^*, x_t^{(\ell)} - x_t^* \rangle + \langle \theta^*, x_t^{(\ell)} - x_t^* \rangle$$
$$\leq \langle \hat{\theta}_\ell - \theta^*, x_t^{(\ell)} - x_t^* \rangle$$
$$\leq |\langle \hat{\theta}_\ell - \theta^*, x_t^{(\ell)} \rangle| + |\langle \hat{\theta}_\ell - \theta^*, x_t^* \rangle|$$
$$\leq 2\varepsilon'_{t,\ell}(\lambda) ,$$

which shows that $x_t^* \in \mathcal{A}_t^{(\ell)}$. By induction, the claim holds for all $\ell \in [s]$, completing the proof. $\quad\square$

*Proof of Theorem 3.* To establish the claimed batch complexity bound, we analyze the lower bound on the length of the $\ell$-th batch, where $\ell = \lceil \log_2 \log_2 T \rceil + 4$. By definition of the schedule, the length of this batch is at least

$$\left\lceil \frac{T^{1 - \frac{1}{3 \cdot 2^{\ell-4}}}}{\log_2 \log_2 T} \right\rceil \geq \frac{T^{1 - 2^{-\lceil \log_2 \log_2 T \rceil}}}{\log_2 \log_2 T} \geq \frac{T^{1 - 2^{-\log_2 \log_2 T}}}{\log_2 \log_2 T} = \frac{T^{1 - \frac{1}{\log_2 T}}}{\log_2 \log_2 T} = \frac{T}{2 \log_2 \log_2 T} .$$

Since the batch length is non-decreasing in $\ell$, every batch with index $\ell \geq \lceil \log_2 \log_2 T \rceil + 4$ has length at least $\frac{T}{2 \log_2 \log_2 T}$. Given the total time horizon $T$, the number of such batches is therefore at most $\lceil 2 \log_2 \log_2 T \rceil$. Including the initial $\lceil \log_2 \log_2 T \rceil + 3$ batches, the total number of batches $B$ is bounded by

$$B \leq \lceil 2 \log_2 \log_2 T \rceil + \lceil \log_2 \log_2 T \rceil + 3 \leq 3 \log_2 \log_2 T + 5 ,$$

which implies that the batch complexity is $\mathcal{O}(\log \log T)$.

Next, we decompose the cumulative expected regret with respect to the good event $E'$. The regret can be expressed as

$$\mathcal{R}(T) = \sum_{k=1}^{B} \sum_{t=\mathcal{T}_{k-1}+1}^{\mathcal{T}_k} \mathbb{E}\left[ \max_{x \in \mathcal{A}_t} \mu(\langle x, \theta^* \rangle) - \mu(\langle x_{t,a_t}, \theta^* \rangle) \right]$$

$$= \sum_{k=1}^{B} \sum_{t=\mathcal{T}_{k-1}+1}^{\mathcal{T}_k} \mathbb{E}\left[ \mu(\langle x_t^*, \theta^* \rangle) - \mu(\langle x_{t,a_t}, \theta^* \rangle) \Big| E'^c \right] \cdot \mathbb{P}(E'^c) + \mathbb{E}\left[ \mu(\langle x_t^*, \theta^* \rangle) - \mu(\langle x_{t,a_t}, \theta^* \rangle) \Big| E' \right] \cdot \mathbb{P}(E') .$$

Since the rewards are supported on $[0, R]$ and $\mathbb{E}[r \mid x; \theta^*] = \mu(\langle x, \theta^* \rangle)$ holds, the instantaneous regret is bounded by the support width, i.e.,

$$\mu(\langle x_t^*, \theta^* \rangle) - \mu(\langle x_{t,a_t}, \theta^* \rangle) \leq R . \tag{14}$$

Hence, the cumulative expected regret can be bounded as

$$\mathcal{R}(T) \le RT \cdot \frac{2(B-2)}{T^2} + \sum_{k=1}^{B} \sum_{t=\mathcal{T}_{k-1}+1}^{\mathcal{T}_k} \mathbb{E}\left[\mu(\langle x_t^*, \theta^* \rangle) - \mu(\langle x_{t,a_t}, \theta^* \rangle)\Big| E'\right] \cdot \mathbb{P}(E')$$

$$\le \frac{2R(B-2)}{T} + \sum_{k=1}^{B} \sum_{t=\mathcal{T}_{k-1}+1}^{\mathcal{T}_k} \mathbb{E}\left[\mu(\langle x_t^*, \theta^* \rangle) - \mu(\langle x_{t,a_t}, \theta^* \rangle)\Big| E'\right]$$

$$= \mathcal{O}\left(\frac{R \log \log T}{T}\right) + \sum_{k=1}^{B} \sum_{t=\mathcal{T}_{k-1}+1}^{\mathcal{T}_k} \mathbb{E}\left[\mu(\langle x_t^*, \theta^* \rangle) - \mu(\langle x_{t,a_t}, \theta^* \rangle)\Big| E'\right].$$

Throughout the remainder of the analysis, we condition on the good event $E'$. Let $\text{Regret}_\ell$ denote the cumulative expected regret incurred during batch $\ell$. We analyze the regret by separating the discussion into four cases, namely $\ell = 1$, $\ell = 2$, $\ell = 3$, and the subsequent batches with $\ell \ge 4$.

**Case 1: $\ell \in \{1, 2, 3\}$.**  In the first three batches, the number of rounds is given by $\left\lceil \frac{\sqrt[3]{T}}{\log_2 \log_2 T} \right\rceil$. By (14), each round incurs an instantaneous regret of at most $R$. Consequently, for the first batch we obtain

$$\text{Regret}_1 = \sum_{t=1}^{\mathcal{T}_1} \mathbb{E}[\mu(\langle x_t^*, \theta^* \rangle) - \mu(\langle x_{t,a_t}, \theta^* \rangle)] \le R\mathcal{T}_1 \le R\left(\frac{\sqrt[3]{T}}{\log_2 \log_2 T} + 1\right) = \mathcal{O}\left(\frac{R\sqrt[3]{T}}{\log \log T}\right).$$

The same reasoning applies to the second and third batches, which yields the same order of regret,

$$\text{Regret}_2 = \text{Regret}_3 = \mathcal{O}\left(\frac{R\sqrt[3]{T}}{\log \log T}\right).$$

**Case 2: $\ell \ge 4$.**  For each batch $\ell \ge 4$, the instantaneous regret can be controlled using the Mean Value Theorem. For some $z$ lying between $\langle x_{t,a_t}, \theta^* \rangle$ and $\langle x_t^*, \theta^* \rangle$, we have

$$\mu(\langle x_t^*, \theta^* \rangle) - \mu(\langle x_{t,a_t}, \theta^* \rangle) = \dot{\mu}(z)\langle x_t^* - x_{t,a_t}, \theta^* \rangle$$

$$= \dot{\mu}(z)(\langle x_t^* - x_{t,a_t}, \theta^* - \hat{\theta}_{\ell-1} \rangle + \langle x_t^* - x_{t,a_t}, \hat{\theta}_{\ell-1} \rangle)$$

$$\le \dot{\mu}(z)(|\langle x_t^*, \hat{\theta}_{\ell-1} - \theta^* \rangle| + |\langle x_{t,a_t}, \hat{\theta}_{\ell-1} - \theta^* \rangle| + \langle x_t^{(\ell-1)} - x_{t,a_t}, \hat{\theta}_{\ell-1} \rangle)$$

$$\le \underbrace{4\dot{\mu}(z)\varepsilon_{t,\ell-1}'(\lambda)}_{\triangleq A_t}. \tag{15}$$

The first inequality uses the fact that $x_t^{(\ell-1)}$ is optimal with respect to $\hat{\theta}_{\ell-1}$, together with the guarantee from Lemma 14 that $x_t^*$ belongs to $\mathcal{A}_t^{(\ell-2)}$. The final bound follows because $x_{t,a_t} \in \mathcal{A}_t^{(\ell-1)}$ by the arm elimination rule, combined with the definition of the good event $E'$ and again Lemma 14, which ensures that $x_t^* \in \mathcal{A}_t^{(\ell-2)}$. As in the analysis of Algorithm 2, during the greedy selection step the regret can be bounded more tightly as $\mu(\langle x_t^*, \theta^* \rangle) - \mu(\langle x_{t,a_t}, \theta^* \rangle)$ by $2\dot{\mu}(z)\varepsilon_{t,\ell-1}'(\lambda)$, analogous to Equation (10). For simplicity in the subsequent analysis, we substitute the greedy selection step with uncertainty-driven exploration, so that in all batches the algorithm may be analyzed under uncertainty-driven exploration alone. For rigor, batch end time $\mathcal{T}_\ell$ must be replaced by $\mathcal{T}_{\ell-1}' := \mathcal{T}_{\ell-1} + \left\lceil \frac{cT^{1-2^{(4-\ell)/3}}}{\log_2 \log_2 T} \right\rceil$, as in the proof of Theorem 2, but this modification affects the regret bound only by a constant factor.

**Bounding $\sum_{t=\mathcal{T}_{\ell-1}+1}^{\mathcal{T}_\ell} \mathbb{E}[A_t]$.**

$$\sum_{t=\mathcal{T}_{\ell-1}+1}^{\mathcal{T}_\ell} \mathbb{E}[A_t] = 4 \sum_{t=\mathcal{T}_{\ell-1}+1}^{\mathcal{T}_\ell} \mathbb{E}[\dot{\mu}(z)\varepsilon_{t,\ell-1}'(\lambda)]$$

$$\le 4 \sum_{t=\mathcal{T}_{\ell-1}+1}^{\mathcal{T}_\ell} \mathbb{E}\left[e^{R(\langle x_t^*, \theta^* \rangle - z)}\dot{\mu}(\langle x_t^*, \theta^* \rangle)\varepsilon_{t,\ell-1}'(\lambda)\right]$$

$$\leq 4 \sum_{t=\mathcal{T}_{\ell-1}+1}^{\mathcal{T}_\ell} \mathbb{E}\left[ e^{R\langle x_t^* - x_{t,a_t}, \theta^*\rangle} \dot{\mu}(\langle x_t^*, \theta^*\rangle)\, \varepsilon_{t,\ell-1}'(\lambda) \right]$$

$$\leq 4 \sum_{t=\mathcal{T}_{\ell-1}+1}^{\mathcal{T}_\ell} \mathbb{E}\left[ e^{4R(S \wedge \varepsilon_{t,\ell-1}'(\lambda))} \dot{\mu}(\langle x_t^*, \theta^*\rangle)\, \varepsilon_{t,\ell-1}'(\lambda) \right]$$

$$\leq 4 \sum_{t=\mathcal{T}_{\ell-1}+1}^{\mathcal{T}_\ell} \mathbb{E}\Big[ \underbrace{\dot{\mu}(\langle x_t^*, \theta^*\rangle)\, \varepsilon_{t,\ell-1}'(\lambda)}_{\triangleq B_t} \Big] + \mathbb{E}\Big[ \underbrace{\frac{e^{4RS}(S \wedge \varepsilon_{t,\ell-1}'(\lambda))}{S} \dot{\mu}(\langle x_t^*, \theta^*\rangle)\, \varepsilon_{t,\ell-1}'(\lambda)}_{\triangleq C_t} \Big] .$$

The first inequality is a direct application of Lemma 10. The second inequality holds because $z$ lies between $\langle x_{t,a_t}, \theta^*\rangle$ and $\langle x_t^*, \theta^*\rangle$. The third inequality follows from Assumption 1 together with the bound on $\langle x_t^* - x_{t,a_t}, \theta^*\rangle$ derived in (15). Finally, the last inequality is obtained by invoking Lemma 9.

**Bounding $\sum_{t=\mathcal{T}_{\ell-1}+1}^{\mathcal{T}_\ell} \mathbb{E}[B_t]$.**

$$\sum_{t=\mathcal{T}_{\ell-1}+1}^{\mathcal{T}_\ell} \mathbb{E}[B_t] = \beta(\lambda) \sum_{t=\mathcal{T}_{\ell-1}+1}^{\mathcal{T}_\ell} \mathbb{E}\left[ \max_{y \in \mathcal{A}_t^{(\ell-2)}} \|\dot{\mu}(\langle x_t^*, \theta^*\rangle)y\|_{V_{\ell-1}^{-1}} \right]$$

$$= \beta(\lambda)(\mathcal{T}_\ell - \mathcal{T}_{\ell-1}) \mathbb{E}\left[ \max_{y \in \mathcal{A}_t^{(\ell-2)}} \|\dot{\mu}(\langle x_t^*, \theta^*\rangle)y\|_{V_{\ell-1}^{-1}} \right] \qquad (16)$$

$$= \frac{\beta(\lambda)(\mathcal{T}_\ell - \mathcal{T}_{\ell-1})}{\mathcal{T}_{\ell-1} - \mathcal{T}_{\ell-2}} \sum_{s=\mathcal{T}_{\ell-2}+1}^{\mathcal{T}_{\ell-1}} \mathbb{E}\left[ \max_{y \in \mathcal{A}_t^{(\ell-2)}} \|\dot{\mu}(\langle x_t^*, \theta^*\rangle)y\|_{V_{\ell-1}^{-1}} \right]$$

$$\leq \frac{\beta(\lambda)(\mathcal{T}_\ell - \mathcal{T}_{\ell-1})}{\mathcal{T}_{\ell-1} - \mathcal{T}_{\ell-2}} \sum_{s=\mathcal{T}_{\ell-2}+1}^{\mathcal{T}_{\ell-1}} \mathbb{E}\left[ \max_{y \in \mathcal{A}_t^{(\ell-2)}} \|\dot{\mu}(\langle x_t^*, \theta^*\rangle)y\|_{H_{s-1}^{-1}} \right]$$

$$= \frac{\beta(\lambda)(\mathcal{T}_\ell - \mathcal{T}_{\ell-1})}{\mathcal{T}_{\ell-1} - \mathcal{T}_{\ell-2}} \sum_{s=\mathcal{T}_{\ell-2}+1}^{\mathcal{T}_{\ell-1}} \mathbb{E}\left[ \mathbb{E}\left[ \max_{y \in \mathcal{A}_t^{(\ell-2)}} \|\dot{\mu}(\langle x_t^*, \theta^*\rangle)y\|_{H_{s-1}^{-1}} \Big| \mathcal{F}_{s-1} \right] \right]$$

$$= \frac{\beta(\lambda)(\mathcal{T}_\ell - \mathcal{T}_{\ell-1})}{\mathcal{T}_{\ell-1} - \mathcal{T}_{\ell-2}} \sum_{s=\mathcal{T}_{\ell-2}+1}^{\mathcal{T}_{\ell-1}} \mathbb{E}\left[ \mathbb{E}\left[ \max_{y \in \mathcal{A}_s^{(\ell-2)}} \|\dot{\mu}(\langle x_s^*, \theta^*\rangle)y\|_{H_{s-1}^{-1}} \Big| \mathcal{F}_{s-1} \right] \right]$$

$$= \frac{\beta(\lambda)(\mathcal{T}_\ell - \mathcal{T}_{\ell-1})}{\mathcal{T}_{\ell-1} - \mathcal{T}_{\ell-2}} \sum_{s=\mathcal{T}_{\ell-2}+1}^{\mathcal{T}_{\ell-1}} \mathbb{E}\left[ \max_{y \in \mathcal{A}_s^{(\ell-2)}} \|\dot{\mu}(\langle x_s^*, \theta^*\rangle)y\|_{H_{s-1}^{-1}} \right]$$

$$= \frac{\beta(\lambda)(\mathcal{T}_\ell - \mathcal{T}_{\ell-1})}{\mathcal{T}_{\ell-1} - \mathcal{T}_{\ell-2}} \sum_{s=\mathcal{T}_{\ell-2}+1}^{\mathcal{T}_{\ell-1}} \mathbb{E}\Big[ \underbrace{\|\dot{\mu}(\langle x_s^*, \theta^*\rangle)x_{s,a_s}\|_{H_{s-1}^{-1}}}_{\triangleq D_s} \Big] .$$

The second equality mirrors the reasoning in (5), since both $V_{\ell-1}$ and the arm elimination rule—determined by $\hat{\theta}_1, \ldots, \hat{\theta}_{\ell-2}$—are measurable with respect to $\mathcal{F}_{\mathcal{T}_{\ell-1}}$, and can therefore be regarded as fixed conditional on this filtration. Given that the contexts are drawn independently and identically, their values are equal. The first inequality follows from the monotonicity, as $H_{s-1} \preceq V_{\ell-1}$ for all $s$ in the interval. The fourth and sixth equalities use the tower property. The fifth equality relies on the fact that, conditional on $\mathcal{F}_{s-1}$, both $H_{s-1}$ and the arm elimination rule (determined by $\hat{\theta}_1, \ldots, \hat{\theta}_{\ell-2}$) are fixed, while the distribution of the contexts remains unchanged. Finally, the last equality follows from the arm-selection strategy, since the factor $\dot{\mu}(\langle x_s^*, \theta^*\rangle)$ does not affect the maximization and can thus be pulled outside without altering the $\arg\max$.

**Bounding $\sum_{s=\mathcal{T}_{\ell-2}+1}^{\mathcal{T}_{\ell-1}} \mathbb{E}[D_s]$.**

$$
\sum_{s=\mathcal{T}_{\ell-2}+1}^{\mathcal{T}_{\ell-1}} \mathbb{E}[D_s] \leq \sum_{s=\mathcal{T}_{\ell-2}+1}^{\mathcal{T}_{\ell-1}} \mathbb{E}\left[\left\| e^{\frac{R}{2}|\langle x_s^*,\theta^*\rangle - \langle x_{s,a_s},\hat{\theta}_{\ell-2}\rangle|} \sqrt{\dot{\mu}(\langle x_s^*,\theta^*\rangle)\dot{\mu}(\langle x_{s,a_s},\hat{\theta}_{\ell-2}\rangle)} x_{s,a_s}\right\|_{H_{s-1}^{-1}}\right]
$$

$$
\leq \sum_{s=\mathcal{T}_{\ell-2}+1}^{\mathcal{T}_{\ell-1}} \mathbb{E}\left[\left\| e^{\frac{R}{2}(2S\wedge 5\varepsilon_{s,\ell-2}'(\lambda))} \sqrt{\dot{\mu}(\langle x_s^*,\theta^*\rangle)\dot{\mu}(\langle x_{s,a_s},\hat{\theta}_{\ell-2}\rangle)} x_{s,a_s}\right\|_{H_{s-1}^{-1}}\right]
$$

$$
\leq \sum_{s=\mathcal{T}_{\ell-2}+1}^{\mathcal{T}_{\ell-1}} \mathbb{E}\left[\left\| e^{\frac{R}{2}(2S\wedge 5\varepsilon_{s,\ell-2}'(\lambda)) + \frac{R}{2}(2S\wedge \varepsilon_{s,\ell-2}'(\lambda))} \sqrt{\dot{\mu}(\langle x_s^*,\theta^*\rangle)\alpha_{s,\ell-2}(\lambda)\dot{\mu}(\langle x_{s,a_s},\hat{\theta}_{\ell-2}\rangle)} x_{s,a_s}\right\|_{H_{s-1}^{-1}}\right]
$$

$$
\leq \sum_{s=\mathcal{T}_{\ell-2}+1}^{\mathcal{T}_{\ell-1}} \mathbb{E}\left[\left\| e^{3R(S\wedge \varepsilon_{s,\ell-2}'(\lambda))} \sqrt{\dot{\mu}(\langle x_s^*,\theta^*\rangle)\alpha_{s,\ell-2}(\lambda)\dot{\mu}(\langle x_{s,a_s},\hat{\theta}_{\ell-2}\rangle)} x_{s,a_s}\right\|_{H_{s-1}^{-1}}\right]
$$

$$
\leq \sum_{s=\mathcal{T}_{\ell-2}+1}^{\mathcal{T}_{\ell-1}} \mathbb{E}\Bigg[\underbrace{\left\| \sqrt{\dot{\mu}(\langle x_s^*,\theta^*\rangle)\alpha_{s,\ell-2}(\lambda)\dot{\mu}(\langle x_{s,a_s},\hat{\theta}_{\ell-2}\rangle)} x_{s,a_s}\right\|_{H_{s-1}^{-1}}}_{\triangleq E_s}\Bigg]
$$

$$
+ \mathbb{E}\Bigg[\underbrace{\left\| \frac{e^{3RS}(S\wedge \varepsilon_{s,\ell-2}'(\lambda))}{S} \sqrt{\dot{\mu}(\langle x_s^*,\theta^*\rangle)\alpha_{s,\ell-2}(\lambda)\dot{\mu}(\langle x_{s,a_s},\hat{\theta}_{\ell-2}\rangle)} x_{s,a_s}\right\|_{H_{s-1}^{-1}}}_{\triangleq F_s}\Bigg].
$$

c The first inequality is obtained by applying Lemma 10 to compare $\dot{\mu}(\langle x_s^*,\theta^*\rangle)$ and $\dot{\mu}(\langle x_{s,a_s},\hat{\theta}_{\ell-2}\rangle)$. The second inequality follows from Assumption 3 together with the decomposition $|\langle x_s^* - x_{s,a_s},\theta^*\rangle| + |\langle x_{s,a_s},\theta^* - \hat{\theta}_{\ell-2}\rangle| \leq 4\varepsilon_{s,\ell-2}'(\lambda) + \varepsilon_{s,\ell-2}'(\lambda)$, as implied by (15) and the definition of the good event $E'$. The third inequality directly follows from the definition of $\alpha_{s,\ell-2}(\lambda)$ when $\ell \geq 4$. The final bound is obtained by invoking Lemma 9.

**Bounding $\sum_{s=\mathcal{T}_{\ell-2}+1}^{\mathcal{T}_{\ell-1}} \mathbb{E}[E_s]$.**

$$
\sum_{s=\mathcal{T}_{\ell-2}+1}^{\mathcal{T}_{\ell-1}} \mathbb{E}[E_s] \leq \sum_{s=\mathcal{T}_{\ell-2}+1}^{\mathcal{T}_{\ell-1}} \sqrt{\mathbb{E}\left[\dot{\mu}(\langle x_s^*,\theta^*\rangle)\right]} \sqrt{\mathbb{E}\left[\left\| \sqrt{\alpha_{s,\ell-2}(\lambda)\dot{\mu}(\langle x_{s,a_s},\hat{\theta}_{\ell-2}\rangle)} x_{s,a_s}\right\|_{H_{s-1}^{-1}}^2\right]}
$$

$$
\leq \frac{\sqrt{\mathcal{T}_{\ell-1} - \mathcal{T}_{\ell-2}}}{\sqrt{\hat{\kappa}}} \sqrt{\sum_{s=\mathcal{T}_{\ell-2}+1}^{\mathcal{T}_{\ell-1}} \mathbb{E}\left[\left\| \sqrt{\alpha_{s,\ell-2}(\lambda)\dot{\mu}(\langle x_{s,a_s},\hat{\theta}_{\ell-2}\rangle)} x_{s,a_s}\right\|_{H_{s-1}^{-1}}^2\right]}
$$

$$
\leq \frac{\sqrt{\mathcal{T}_{\ell-1} - \mathcal{T}_{\ell-2}}}{\sqrt{\hat{\kappa}}} \sqrt{2\log\left(\frac{\det(H_{\mathcal{T}_{\ell-1}})}{\det(\lambda I)}\right)}
$$

$$
\leq \frac{\sqrt{2d(\mathcal{T}_{\ell-1} - \mathcal{T}_{\ell-2})\log(2T)}}{\sqrt{\hat{\kappa}}}.
$$

The first two inequalities are obtained by successive applications of the Cauchy-Schwarz inequality. The third inequality follows directly from Corollary 2. Finally, the last inequality is derived using the same reasoning as in (13), where the determinant is upper bounded by a trace argument, yielding a logarithmic dependence on $T$.

**Bounding $\sum_{s=\mathcal{T}_{\ell-2}+1}^{\mathcal{T}_{\ell-1}} \mathbb{E}[F_s]$.**

$$
\sum_{s=\mathcal{T}_{\ell-2}+1}^{\mathcal{T}_{\ell-1}} \mathbb{E}[F_s] \leq \frac{e^{3RS}}{S} \sum_{s=\mathcal{T}_{\ell-2}+1}^{\mathcal{T}_{\ell-1}} \mathbb{E}\left[\varepsilon_{s,\ell-2}(\lambda)\left\| \sqrt{\dot{\mu}(\langle x_s^*,\theta^*\rangle)\alpha_{s,\ell-2}(\lambda)\dot{\mu}(\langle x_{s,a_s},\hat{\theta}_{\ell-2}\rangle)} x_{s,a_s}\right\|_{H_{s-1}^{-1}}\right]
$$

$$= \frac{e^{3RS}\beta(\lambda)}{S} \sum_{s=\mathcal{T}_{\ell-2}+1}^{\mathcal{T}_{\ell-1}} \mathbb{E}\left[\max_{y\in\mathcal{A}_s^{(\ell-3)}} \|y\|_{V_{\ell-2}^{-1}} \left\|\sqrt{\dot\mu(\langle x_s^*,\theta^*\rangle)}\alpha_{s,\ell-2}(\lambda)\dot\mu(\langle x_{s,a_s},\hat\theta_{\ell-2}\rangle)x_{s,a_s}\right\|_{H_{s-1}^{-1}}\right]$$

$$\leq \frac{e^{3RS}\beta(\lambda)}{S} \sum_{s=\mathcal{T}_{\ell-2}+1}^{\mathcal{T}_{\ell-1}} \sqrt{\mathbb{E}\left[\max_{y\in\mathcal{A}_s^{(\ell-3)}} \|\sqrt{\dot\mu(\langle x_s^*,\theta^*\rangle)}y\|_{V_{\ell-2}^{-1}}^2\right]} \sqrt{\mathbb{E}\left[\left\|\sqrt{\alpha_{s,\ell-2}(\lambda)\dot\mu(\langle x_{s,a_s},\hat\theta_{\ell-2}\rangle)}x_{s,a_s}\right\|_{H_{s-1}^{-1}}^2\right]}$$

$$\leq \frac{e^{3RS}\beta(\lambda)\sqrt{2d(\mathcal{T}_{\ell-1}-\mathcal{T}_{\ell-2})\log(2T)}}{S} \sqrt{\underbrace{\mathbb{E}\left[\max_{y\in\mathcal{A}_s^{(\ell-3)}} \|\sqrt{\dot\mu(\langle x_s^*,\theta^*\rangle)}y\|_{V_{\ell-2}^{-1}}^2\right]}_{\triangleq G_s}} .$$

The second inequality follows from an application of the Cauchy-Schwarz inequality. The final inequality uses the fact that the expectation $\mathbb{E}\left[\max_{y\in\mathcal{A}_s^{(\ell-3)}} \|\sqrt{\dot\mu(\langle x_s^*,\theta^*\rangle)}y\|_{V_{\ell-2}^{-1}}^2\right]$ takes the same value for all $s \in [\mathcal{T}_{\ell-2}+1, \mathcal{T}_{\ell-1}]$, as established in (16), together with Corollary 2, which bounds the quadratic form by a log-determinant expression.

**Bounding $\mathbb{E}[G_s]$ for $s \in [\mathcal{T}_{\ell-2}+1, \mathcal{T}_{\ell-1}]$.**

$$\mathbb{E}[G_s] = \frac{1}{\mathcal{T}_{\ell-2}-\mathcal{T}_{\ell-3}} \sum_{u=\mathcal{T}_{\ell-3}+1}^{\mathcal{T}_{\ell-2}} \mathbb{E}\left[\max_{y\in\mathcal{A}_s^{(\ell-3)}} \|\sqrt{\dot\mu(\langle x_s^*,\theta^*\rangle)}y\|_{V_{\ell-2}^{-1}}^2\right]$$

$$\leq \frac{1}{\mathcal{T}_{\ell-2}-\mathcal{T}_{\ell-3}} \sum_{u=\mathcal{T}_{\ell-3}+1}^{\mathcal{T}_{\ell-2}} \mathbb{E}\left[\max_{y\in\mathcal{A}_s^{(\ell-3)}} \|\sqrt{\dot\mu(\langle x_s^*,\theta^*\rangle)}y\|_{H_{u-1}^{-1}}^2\right]$$

$$= \frac{1}{\mathcal{T}_{\ell-2}-\mathcal{T}_{\ell-3}} \sum_{u=\mathcal{T}_{\ell-3}+1}^{\mathcal{T}_{\ell-2}} \mathbb{E}\left[\mathbb{E}\left[\max_{y\in\mathcal{A}_s^{(\ell-3)}} \|\sqrt{\dot\mu(\langle x_s^*,\theta^*\rangle)}y\|_{H_{u-1}^{-1}}^2\,\middle|\,\mathcal{F}_{u-1}\right]\right]$$

$$= \frac{1}{\mathcal{T}_{\ell-2}-\mathcal{T}_{\ell-3}} \sum_{u=\mathcal{T}_{\ell-3}+1}^{\mathcal{T}_{\ell-2}} \mathbb{E}\left[\mathbb{E}\left[\max_{y\in\mathcal{A}_u^{(\ell-3)}} \|\sqrt{\dot\mu(\langle x_u^*,\theta^*\rangle)}y\|_{H_{u-1}^{-1}}^2\,\middle|\,\mathcal{F}_{u-1}\right]\right]$$

$$= \frac{1}{\mathcal{T}_{\ell-2}-\mathcal{T}_{\ell-3}} \sum_{u=\mathcal{T}_{\ell-3}+1}^{\mathcal{T}_{\ell-2}} \mathbb{E}\left[\max_{y\in\mathcal{A}_u^{(\ell-3)}} \|\sqrt{\dot\mu(\langle x_u^*,\theta^*\rangle)}y\|_{H_{u-1}^{-1}}^2\right]$$

$$= \frac{1}{\mathcal{T}_{\ell-2}-\mathcal{T}_{\ell-3}} \sum_{u=\mathcal{T}_{\ell-3}+1}^{\mathcal{T}_{\ell-2}} \mathbb{E}\left[\underbrace{\|\sqrt{\dot\mu(\langle x_u^*,\theta^*\rangle)}x_{u,a_u}\|_{H_{u-1}^{-1}}^2}_{\triangleq I_u}\right] .$$

The first inequality follows from the monotonicity of the matrices, since $H_{u-1} \preceq V_{\ell-2}$ for all $u$ in the summation range. The second and fourth equalities are applications of the tower property. The third equality holds because, conditional on $\mathcal{F}_{u-1}$, both $H_{u-1}$ and the arm elimination rule (determined by $\hat\theta_1,\ldots,\hat\theta_{\ell-3}$) are fixed, while the distribution of the contexts remains unchanged. Finally, the last equality follows from the arm-selection strategy: the multiplicative factor $\dot\mu(\langle x_u^*,\theta^*\rangle)$ does not affect the maximization and can therefore be factored out without altering the $\arg\max$.

**Bounding $\sum_{u=\mathcal{T}_{\ell-3}+1}^{\mathcal{T}_{\ell-2}} \mathbb{E}[I_u]$.**

$$\sum_{u=\mathcal{T}_{\ell-3}+1}^{\mathcal{T}_{\ell-2}} \mathbb{E}[I_u] \leq \sum_{u=\mathcal{T}_{\ell-3}+1}^{\mathcal{T}_{\ell-2}} \mathbb{E}\left[e^{R|\langle x_u^*,\theta^*\rangle - \langle x_{u,a_u},\hat\theta_{\ell-3}\rangle|} \left\|\sqrt{\dot\mu(\langle x_{u,a_u},\hat\theta_{\ell-3}\rangle)}x_{u,a_u}\right\|_{H_{u-1}^{-1}}^2\right]$$

$$\leq \sum_{u=\mathcal{T}_{\ell-3}+1}^{\mathcal{T}_{\ell-2}} \mathbb{E}\left[e^{4RS}\left\|\sqrt{\alpha_{s,\ell-3}(\lambda)\dot\mu(\langle x_{u,a_u},\hat\theta_{\ell-3}\rangle)}x_{u,a_u}\right\|_{H_{u-1}^{-1}}^2\right]$$

$$\leq 2e^{4RS}d\log(2T) .$$

The first inequality is obtained by applying Lemma 10, which allows us to compare $\dot{\mu}(\langle x_u^*, \theta^* \rangle)$ and $\dot{\mu}(\langle x_{u,a_u}, \hat{\theta}_{\ell-3} \rangle)$. The second inequality uses Assumption 1, together with the fact that $\alpha_{s,\ell-3}(\lambda) \geq e^{-2RS}$ for all $\ell \geq 4$. Finally, the last inequality follows from Corollary 2, which bounds the quadratic form in terms of a log-determinant expression and yields the stated order.

We now combine the previous bounds step by step. First, from the result on $\mathbb{E}[G_s]$, we obtain

$$\mathbb{E}[G_s] \leq \frac{1}{\mathcal{T}_{\ell-2} - \mathcal{T}_{\ell-3}} \sum_{u=\mathcal{T}_{\ell-3}+1}^{\mathcal{T}_{\ell-2}} \mathbb{E}[I_u] \leq \frac{2e^{4RS} d \log(2T)}{\mathcal{T}_{\ell-2} - \mathcal{T}_{\ell-3}} \ .$$

Next, substituting this into the bound for $\sum_{s=\mathcal{T}_{\ell-2}+1}^{\mathcal{T}_{\ell-1}} \mathbb{E}[F_s]$, we obtain

$$\sum_{s=\mathcal{T}_{\ell-2}+1}^{\mathcal{T}_{\ell-1}} \mathbb{E}[F_s] \leq \frac{e^{3RS} \beta(\lambda) \sqrt{2d(\mathcal{T}_{\ell-1} - \mathcal{T}_{\ell-2}) \log(2T)}}{S} \sqrt{\mathbb{E}[G_s]}$$

$$\leq \frac{2e^{5RS} \beta(\lambda) d \log(2T) \sqrt{\mathcal{T}_{\ell-1} - \mathcal{T}_{\ell-2}}}{S \sqrt{\mathcal{T}_{\ell-2} - \mathcal{T}_{\ell-3}}} \ .$$

Furthermore, combining this with the result on $\sum_{s=\mathcal{T}_{\ell-2}+1}^{\mathcal{T}_{\ell-1}} \mathbb{E}[E_s]$, we can bound $\sum_{s=\mathcal{T}_{\ell-2}+1}^{\mathcal{T}_{\ell-1}} \mathbb{E}[D_s]$ as

$$\sum_{s=\mathcal{T}_{\ell-2}+1}^{\mathcal{T}_{\ell-1}} \mathbb{E}[D_s] \leq \sum_{s=\mathcal{T}_{\ell-2}+1}^{\mathcal{T}_{\ell-1}} \mathbb{E}[E_s] + \sum_{s=\mathcal{T}_{\ell-2}+1}^{\mathcal{T}_{\ell-1}} \mathbb{E}[F_s]$$

$$\leq \frac{\sqrt{2d(\mathcal{T}_{\ell-1} - \mathcal{T}_{\ell-2}) \log(2T)}}{\sqrt{\hat{\kappa}}} + \frac{2e^{5RS} \beta(\lambda) d \log(2T) \sqrt{\mathcal{T}_{\ell-1} - \mathcal{T}_{\ell-2}}}{S \sqrt{\mathcal{T}_{\ell-2} - \mathcal{T}_{\ell-3}}} \ .$$

Finally, substituting this bound into the expression for $\sum_{t=\mathcal{T}_{\ell-1}+1}^{\mathcal{T}_{\ell}} \mathbb{E}[B_t]$ yields

$$\sum_{t=\mathcal{T}_{\ell-1}+1}^{\mathcal{T}_{\ell}} \mathbb{E}[B_t] \leq \frac{\beta(\lambda)(\mathcal{T}_{\ell} - \mathcal{T}_{\ell-1})}{\mathcal{T}_{\ell-1} - \mathcal{T}_{\ell-2}} \sum_{s=\mathcal{T}_{\ell-2}+1}^{\mathcal{T}_{\ell-1}} \mathbb{E}[D_s]$$

$$\leq \frac{\beta(\lambda)(\mathcal{T}_{\ell} - \mathcal{T}_{\ell-1})}{\sqrt{\mathcal{T}_{\ell-1} - \mathcal{T}_{\ell-2}}} \left( \frac{\sqrt{2d \log(2T)}}{\sqrt{\hat{\kappa}}} + \frac{2e^{5RS} \beta(\lambda) d \log(2T)}{S \sqrt{\mathcal{T}_{\ell-2} - \mathcal{T}_{\ell-3}}} \right) \ .$$

**Bounding $\sum_{t=\mathcal{T}_{\ell-1}+1}^{\mathcal{T}_{\ell}} \mathbb{E}[C_t]$.**

$$\sum_{t=\mathcal{T}_{\ell-1}+1}^{\mathcal{T}_{\ell}} \mathbb{E}[C_t] \leq \frac{e^{4RS}}{S} \sum_{t=\mathcal{T}_{\ell-1}+1}^{\mathcal{T}_{\ell}} \mathbb{E}\left[ \dot{\mu}(\langle x_t^*, \theta^* \rangle) \varepsilon'_{t,\ell-1}(\lambda)^2 \right]$$

$$= \frac{e^{4RS} \beta(\lambda)^2}{S} \sum_{t=\mathcal{T}_{\ell-1}+1}^{\mathcal{T}_{\ell}} \mathbb{E}\left[ \max_{z \in \mathcal{A}_t^{(\ell-2)}} \| \sqrt{\dot{\mu}(\langle x_t^*, \theta^* \rangle)} z \|_{V_{\ell-1}^{-1}}^2 \right]$$

$$= \frac{e^{4RS} \beta(\lambda)^2 (\mathcal{T}_{\ell} - \mathcal{T}_{\ell-1})}{S} \mathbb{E}\left[ \max_{z \in \mathcal{A}_t^{(\ell-2)}} \| \sqrt{\dot{\mu}(\langle x_t^*, \theta^* \rangle)} z \|_{V_{\ell-1}^{-1}}^2 \right]$$

$$= \frac{e^{4RS} \beta(\lambda)^2 (\mathcal{T}_{\ell} - \mathcal{T}_{\ell-1})}{S(\mathcal{T}_{\ell-1} - \mathcal{T}_{\ell-2})} \sum_{s=\mathcal{T}_{\ell-2}+1}^{\mathcal{T}_{\ell-1}} \mathbb{E}\left[ \max_{z \in \mathcal{A}_t^{(\ell-2)}} \| \sqrt{\dot{\mu}(\langle x_t^*, \theta^* \rangle)} z \|_{V_{\ell-1}^{-1}}^2 \right]$$

$$\leq \frac{e^{4RS} \beta(\lambda)^2 (\mathcal{T}_{\ell} - \mathcal{T}_{\ell-1})}{S(\mathcal{T}_{\ell-1} - \mathcal{T}_{\ell-2})} \sum_{s=\mathcal{T}_{\ell-2}+1}^{\mathcal{T}_{\ell-1}} \mathbb{E}\left[ \max_{z \in \mathcal{A}_t^{(\ell-2)}} \| \sqrt{\dot{\mu}(\langle x_t^*, \theta^* \rangle)} z \|_{H_{s-1}^{-1}}^2 \right]$$

$$= \frac{e^{4RS} \beta(\lambda)^2 (\mathcal{T}_{\ell} - \mathcal{T}_{\ell-1})}{S(\mathcal{T}_{\ell-1} - \mathcal{T}_{\ell-2})} \sum_{s=\mathcal{T}_{\ell-2}+1}^{\mathcal{T}_{\ell-1}} \mathbb{E}\left[ \mathbb{E}\left[ \max_{z \in \mathcal{A}_t^{(\ell-2)}} \| \sqrt{\dot{\mu}(\langle x_t^*, \theta^* \rangle)} z \|_{H_{s-1}^{-1}}^2 \bigg| \mathcal{F}_{s-1} \right] \right]$$

$$= \frac{e^{4RS}\beta(\lambda)^2(\mathcal{T}_\ell - \mathcal{T}_{\ell-1})}{S(\mathcal{T}_{\ell-1} - \mathcal{T}_{\ell-2})} \sum_{s=\mathcal{T}_{\ell-2}+1}^{\mathcal{T}_{\ell-1}} \mathbb{E}\left[\mathbb{E}\left[\max_{z \in \mathcal{A}_s^{(\ell-2)}} \|\sqrt{\dot{\mu}(\langle x_s^*, \theta^* \rangle)}z\|_{H_{s-1}^{-1}}^2 \Big| \mathcal{F}_{s-1}\right]\right]$$

$$= \frac{e^{4RS}\beta(\lambda)^2(\mathcal{T}_\ell - \mathcal{T}_{\ell-1})}{S(\mathcal{T}_{\ell-1} - \mathcal{T}_{\ell-2})} \sum_{s=\mathcal{T}_{\ell-2}+1}^{\mathcal{T}_{\ell-1}} \mathbb{E}\left[\max_{z \in \mathcal{A}_s^{(\ell-2)}} \|\sqrt{\dot{\mu}(\langle x_s^*, \theta^* \rangle)}z\|_{H_{s-1}^{-1}}^2\right]$$

$$= \frac{e^{4RS}\beta(\lambda)^2(\mathcal{T}_\ell - \mathcal{T}_{\ell-1})}{S(\mathcal{T}_{\ell-1} - \mathcal{T}_{\ell-2})} \sum_{s=\mathcal{T}_{\ell-2}+1}^{\mathcal{T}_{\ell-1}} \mathbb{E}\left[\|\sqrt{\dot{\mu}(\langle x_s^*, \theta^* \rangle)}x_{s,a_s}\|_{H_{s-1}^{-1}}^2\right]$$

$$= \frac{e^{4RS}\beta(\lambda)^2(\mathcal{T}_\ell - \mathcal{T}_{\ell-1})}{S(\mathcal{T}_{\ell-1} - \mathcal{T}_{\ell-2})} \sum_{s=\mathcal{T}_{\ell-2}+1}^{\mathcal{T}_{\ell-1}} \mathbb{E}\left[I_s\right]$$

$$\leq \frac{2e^{8RS}\beta(\lambda)^2(\mathcal{T}_\ell - \mathcal{T}_{\ell-1})d\log(2T)}{S(\mathcal{T}_{\ell-1} - \mathcal{T}_{\ell-2})} \ .$$

The second equality follows from the same reasoning as in (5), since both $V_{\ell-1}$ and the arm elimination rule—determined by $\hat{\theta}_1, \ldots, \hat{\theta}_{\ell-2}$—are measurable with respect to $\mathcal{F}_{\mathcal{T}_{\ell-1}}$ and can therefore be treated as fixed conditional on this filtration; given that the contexts are drawn independently and identically, their values coincide. The second inequality uses the monotonicity of the matrices, as $H_{s-1} \preceq V_{\ell-1}$ for all $s$ in the interval. The fourth and sixth equalities apply the tower property. The fifth equality holds because, conditional on $\mathcal{F}_{s-1}$, both $H_{s-1}$ and the arm elimination rule (determined by $\hat{\theta}_1, \ldots, \hat{\theta}_{\ell-2}$) are fixed, while the distribution of the contexts remains unchanged. The seventh equality is justified by the arm-selection strategy, as the multiplicative factor $\dot{\mu}(\langle x_s^*, \theta^* \rangle)$ does not affect the maximization and hence does not alter the $\arg\max$. The final inequality follows from the previously established bound on $\sum_{s=\mathcal{T}_{\ell-2}+1}^{\mathcal{T}_{\ell-1}} \mathbb{E}[I_s]$.

Combining the previously derived bounds for $\sum_{t=\mathcal{T}_{\ell-1}+1}^{\mathcal{T}_\ell} \mathbb{E}[B_t]$ and $\sum_{t=\mathcal{T}_{\ell-1}+1}^{\mathcal{T}_\ell} \mathbb{E}[C_t]$, we obtain

$$\sum_{t=\mathcal{T}_{\ell-1}+1}^{\mathcal{T}_\ell} \mathbb{E}[A_t] \leq 4 \sum_{t=\mathcal{T}_{\ell-1}+1}^{\mathcal{T}_\ell} \mathbb{E}[B_t] + 4 \sum_{t=\mathcal{T}_{\ell-1}+1}^{\mathcal{T}_\ell} \mathbb{E}[C_t]$$

$$\leq \frac{4\beta(\lambda)(\mathcal{T}_\ell - \mathcal{T}_{\ell-1})}{\sqrt{\mathcal{T}_{\ell-1} - \mathcal{T}_{\ell-2}}} \left(\frac{\sqrt{2d\log(2T)}}{\sqrt{\hat{\kappa}}} + \frac{2e^{5RS}\beta(\lambda)d\log(2T)}{S\sqrt{\mathcal{T}_{\ell-2} - \mathcal{T}_{\ell-3}}}\right) + \frac{8e^{8RS}\beta(\lambda)^2(\mathcal{T}_\ell - \mathcal{T}_{\ell-1})d\log(2T)}{S(\mathcal{T}_{\ell-1} - \mathcal{T}_{\ell-2})} \ .$$

For the case $\ell = 4$, this simplifies to

$$\sum_{t=\mathcal{T}_{\ell-1}+1}^{\mathcal{T}_\ell} \mathbb{E}[A_t] = \mathcal{O}\left(\frac{RS\sqrt{d+\log T} \cdot \frac{T^{\frac{2}{3}}}{\log\log T}}{\sqrt{\frac{\sqrt[3]{T}}{\log\log T}}} \left(\frac{\sqrt{d\log T}}{\sqrt{\hat{\kappa}}} + \frac{Re^{5RS}d\log T\sqrt{d+\log T}}{\sqrt{\frac{\sqrt[3]{T}}{\log\log T}}}\right)\right)$$

$$+ \mathcal{O}\left(\frac{R^2Se^{8RS} \cdot \frac{dT^{\frac{2}{3}}(d+\log T)\log T}{\log\log T}}{\frac{\sqrt[3]{T}}{\log\log T}}\right)$$

$$= \mathcal{O}\left(\frac{RS\sqrt{d(d+\log T)T}\log T}{\sqrt{\hat{\kappa}}\log\log T} + R^2Se^{8RS}d(d+\log T)T^{\frac{1}{3}}\log T\right) \ .$$

For all subsequent batches with $\ell \geq 5$, we similarly obtain

$$\sum_{t=\mathcal{T}_{\ell-1}+1}^{\mathcal{T}_\ell} \mathbb{E}[A_t] = \mathcal{O}\left(\frac{RS\sqrt{d+\log T} \cdot \frac{T^{1-\frac{1}{3\cdot2^{\ell-4}}}}{\log\log T}}{\sqrt{\frac{T^{1-\frac{1}{3\cdot2^{\ell-5}}}}{\log\log T}}} \left(\frac{\sqrt{d\log T}}{\sqrt{\hat{\kappa}}} + \frac{Re^{5RS}d\log T\sqrt{d+\log T}}{\sqrt{\frac{T^{1-\frac{1}{3\cdot2^{\ell-6}}}}{\log\log T}}}\right)\right)$$

$$+ \mathcal{O}\left(\frac{R^2Se^{8RS} \cdot \frac{dT^{1-\frac{1}{3\cdot2^{\ell-4}}}(d+\log T)\log T}{\log\log T}}{\frac{T^{1-\frac{1}{3\cdot2^{\ell-5}}}}{\log\log T}}\right)$$

$$= \mathcal{O}\left(\frac{RS\sqrt{d(d+\log T)T\log T}}{\sqrt{\hat{\kappa}}\log\log T} + R^2 Se^{8RS}d(d+\log T)T^{\frac{1}{3\cdot 2^{\ell-5}}}\log T\right)$$

$$= \mathcal{O}\left(\frac{RS\sqrt{d(d+\log T)T\log T}}{\sqrt{\hat{\kappa}}\log\log T} + R^2 Se^{8RS}d(d+\log T)T^{\frac{1}{3}}\log T\right) .$$

Therefore, the total worst-case regret is bounded as

$$\mathcal{R}(T) = \sum_{\ell=1}^{3}\mathrm{Regret}_\ell + \sum_{\ell=4}^{B}\mathrm{Regret}_\ell$$

$$= \mathcal{O}\left(\frac{RT^{\frac{1}{3}}}{\log\log T}\right) + \sum_{\ell=4}^{B}\sum_{t=\mathcal{T}_{\ell-1}+1}^{\mathcal{T}_\ell}\mathbb{E}[\mu(\langle x_t^*, \theta^*\rangle) - \mu(\langle x_{t,a_t}, \theta^*\rangle)]$$

$$\leq \mathcal{O}\left(\frac{RT^{\frac{1}{3}}}{\log\log T}\right) + \sum_{\ell=4}^{B}\sum_{t=\mathcal{T}_{\ell-1}+1}^{\mathcal{T}_\ell}\mathbb{E}[A_t]$$

$$= \mathcal{O}\left(RS\sqrt{\frac{d(d+\log T)T\log T\log\log T}{\hat{\kappa}}} + \left(R^2 Se^{8RS}d(d+\log T)\log T\log\log T + \frac{R}{\log\log T}\right)T^{\frac{1}{3}}\right)$$

$$= \tilde{\mathcal{O}}\left(\frac{dRS\sqrt{T}}{\sqrt{\hat{\kappa}}} + (R^2 Se^{8RS}d^2 + R)T^{\frac{1}{3}}\right) .$$

$\square$

# D  ADDITIONAL EXPERIMENTS

## D.1  LINEAR CONTEXTUAL BANDITS: EXPERIMENTAL RESULTS WITH NORMAL CONTEXTS

We evaluate the performance of `BLCE-G` and `BLCE` by measuring the cumulative regret over $T = 10,000$ rounds. At each iteration, $K$ arms are independently sampled from a $d$-dimensional normal distribution, and the parameter vector $\theta^*$ is drawn from a $d$-dimensional normal distribution. Each experiment is repeated 10 times. We consider $(K, d) \in \{(1000, 5), (5000, 10), (50, 20), (100, 30)\}$, where the first two pairs represent the large-$K$ regime and the latter two correspond to the small-$K$ regime.

For comparison, we benchmark against state-of-the-art algorithms: *Rarely Switching OFUL* (`RS-OFUL`; Abbasi-Yadkori et al. 2011), `BatchLinUCB-DG` (Ruan et al. 2021), *Efficient Batched Algorithm for linear contextual Bandits* (`SoftBatch`; Hanna et al. 2023b), and `BatchLearning` (Zhang et al. 2025). The within-batch allocation rate for `BLCE-G` and `BLCE` is set to $c = 0.5$. For `RS-OFUL`, the switching parameter is $C = 3$, and for `SoftBatch`, the discretization parameter is $q = 1/(8\sqrt{d})$. Algorithms of Hanna et al. (2023b) incur substantial computational overhead, as reflected in their time complexity reported in Table 1; we therefore omit their regret plots. Importantly, `BLCE-G` and `BLCE` are implemented with the exact theoretical hyperparameters specified in our main results, without additional tuning.

We report three types of figures. First, the average cumulative regret (solid line) together with its standard deviation (shaded region) over 10 runs. Second, zoomed-in views of the regret curves to highlight the differences between `BLCE-G` and `BLCE`. Third, the average batch complexity across 10 runs, showing how frequently each algorithm updates its policy.

As illustrated in Figure 2, both `BLCE-G` and `BLCE` consistently outperform all baselines in both the large-$K$ and small-$K$ regimes, achieving lowest regret and exhibiting greater stability. These results confirm that `BLCE-G` and `BLCE` not only attain the tightest theoretical guarantees but also deliver strong empirical performance, thereby fulfilling their design objectives of near-optimal regret and minimal batch complexity. Furthermore, runtime comparisons in Table 4 show that `BLCE-G` and `BLCE` incur substantially lower computation cost than other theoretically optimal algorithms. In particular, `BLCE`, which eliminates reliance on G-optimal design, attains the fastest runtime—comparable even to suboptimal algorithms.

Overall, these experiments demonstrate a distinctive advantage of our approach: `BLCE-G` and `BLCE` combine minimax-optimal regret guarantees with practical efficiency. This dual benefit of theoretical optimality and empirical superiority sets them apart from all prior methods for batched linear contextual bandits.

Table 4: Average runtime (seconds) over 10 runs.

| $(K, d)$ | Suboptimal algorithms | | Optimal algorithms | | | | |
|---|---|---|---|---|---|---|---|
| | RS-OFUL | SoftBatch | BatchLinUCB-DG | Hanna et al. (2023b) | BatchLearning | BLCE-G | BLCE |
| $(1000, 5)$ | 1.37 | 1.15 | 148.19 | Exponential | 143.07 | 3.58 | 2.17 |
| $(5000, 10)$ | 10.46 | 12.62 | 555.11 | Exponential | 590.69 | 9.16 | 6.19 |
| $(50, 20)$ | 0.54 | 1.83 | 981.19 | Exponential | 46.24 | 1.39 | 1.05 |
| $(100, 30)$ | 0.95 | 3.51 | 2773.66 | Exponential | 75.07 | 1.97 | 1.41 |

## D.2  EXPERIMENTAL RESULTS FOR GENERALIZED LINEAR CONTEXTUAL BANDITS

We evaluate the performance of `BGLE` by measuring the cumulative regret over a horizon of $T = 10,000$ rounds. At each iteration, $K$ arms are independently sampled from either a $d$-dimensional uniform or normal distribution, and the parameter vector $\theta^*$ is drawn from a $d$-dimensional normal distribution. Each experiment is repeated 20 times for the parameter pairs $(K, d) \in \{(20, 2), (50, 3)\}$, considering both uniform and normal contexts.

For comparison, we benchmark against state-of-the-art algorithm: `B-GLinCB` (Sawarni et al. 2024). The within-batch allocation rate for `BGLE` is set to $c = 0.5$, and we conduct experiments on logistic bandits with $R = S = 1$. Importantly, `BGLE` is implemented with the exact theoretical hyperparameters from our main results, without any tuning.

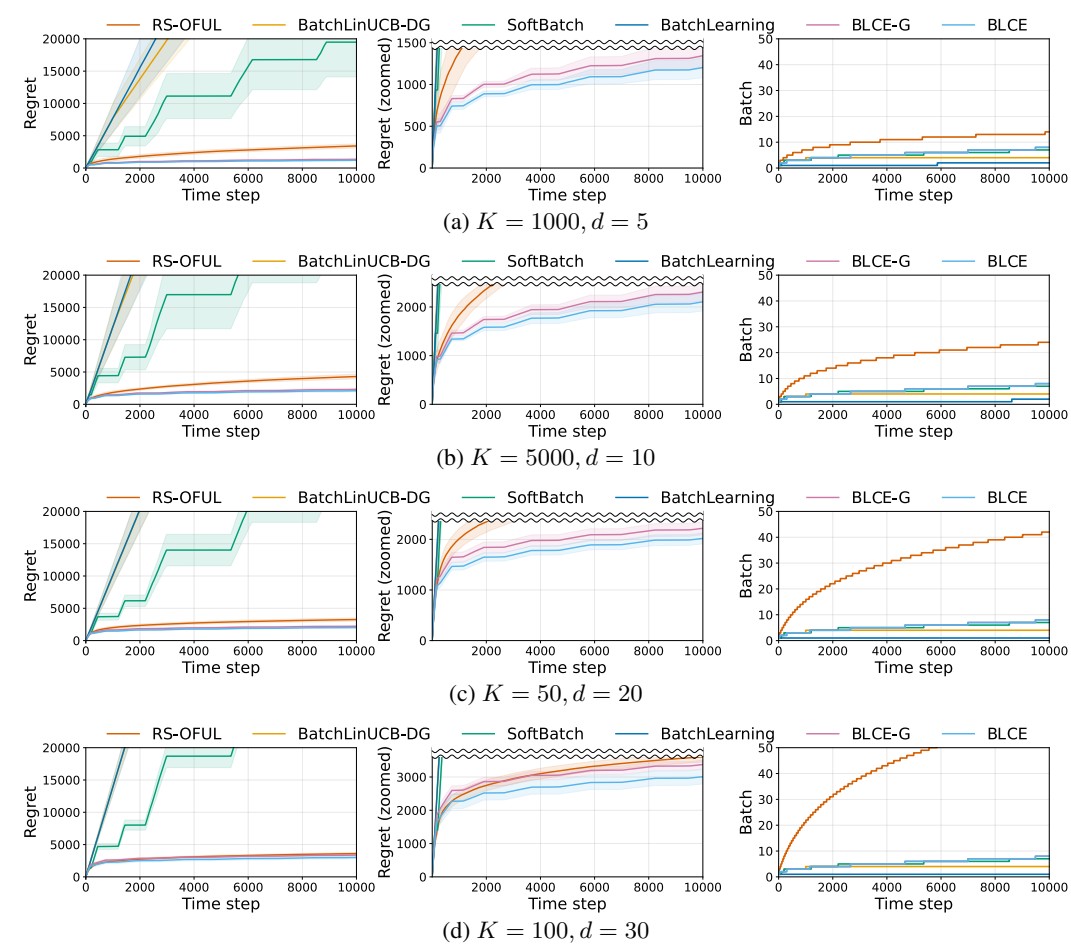

Figure 2: Regret, zoomed-in regret, and batch complexity over time for different values of $K$ and $d$.

We report two types of figures. First, the average cumulative regret (solid line) together with its standard deviation (shaded region) over 20 runs. Second, the average batch complexity across 20 runs, showing how frequently each algorithm updates its policy.

As shown in Figure 3, `BGLE` consistently outperforms the baseline, achieving lowest regret and demonstrating stable performance. Runtime comparisons in Table 5 further show that `BGLE` incur lower computation cost than `B-GLinCB`.

An additional limitation of `B-GLinCB` is that it often uses only one batch (see Figure 3), even for small values of $K$ and $d$. This behavior arises because its first batch length is determined by

$$\left(900R^2 S\sqrt{\kappa}e^{3RS}d^3\log T\sqrt{T}\right)^{\frac{2}{3}},$$

which easily exceeds $T = 10,000$ when $d$ is small, thereby preventing meaningful batching.

Overall, these experiments highlight a clear advantage of our approach: BGLE combines optimal regret guarantees with practical efficiency. This dual benefit of theoretical optimality and empirical superiority distinguishes our method from prior approaches to batched generalized linear contextual bandits.

### D.3 PERFORMANCE COMPARISON BETWEEN `BLCE-G` AND `BLCE`

We evaluate the performance of `BLCE-G` and `BLCE` over $T = 10,000$ rounds with $K = 1000$ arms and feature dimension $d = 5$. At the beginning of each run, the parameter vector $\theta^*$ is sampled

Table 5: Average runtime (seconds) over 20 runs.

| $(K, d)$ | Uniform distribution | | Normal distribution | |
| | B-GLinCB | BGLE | B-GLinCB | BGLE |
|---|---|---|---|---|
| $(20, 2)$ | 24.94 | 3.62 | 25.82 | 3.88 |
| $(50, 3)$ | 27.45 | 3.91 | 28.78 | 4.06 |

from a $d$-dimensional normal distribution. We then conduct *four independent experiments*, each corresponding to a different fixed context distribution. In each experiment, and for every round, the $d$-dimensional feature vectors are drawn i.i.d. from one of the following distributions: (i) a Student-$t$ distribution with 1.5 degrees of freedom, (ii) a Beta$(0.5, 0.5)$ distribution supported on $(0, 1)$, (iii) a Laplace distribution with location 0 and scale $1/\sqrt{2}$, or (iv) an Exponential distribution with unit scale supported on $[0, \infty)$. Thus, the four experiments differ only in the underlying feature distribution, while $T$, $K$, and $d$ remain fixed across all settings.

The within-batch allocation rate for both BLCE-G and BLCE is fixed at $c = 0.5$. As in Appendix D.1, we report cumulative regret, zoomed-in regret, and batch complexity, averaged over 10 independent runs for each distributional setting.

As shown in Figure 4, BLCE-G and BLCE exhibit remarkably similar regret performance, with neither algorithm consistently dominating the other. Across most distributional settings, BLCE achieves slightly lower regret, whereas in the heavy-tailed Student-$t$ distribution case this difference nearly vanishes and BLCE-G matches or occasionally exceeds the performance of BLCE. While theory guarantees that BLCE-G has a strictly smaller minimax regret bound, these guarantees characterize worst-case regimes and do not necessarily arise in benign stochastic environments. Hence, the empirical results do not contradict the theoretical results.

Finally, the runtime comparison in Table 6 shows that BLCE incurs lower computational cost than BLCE-G. By removing the G-optimal design component, BLCE achieves a simpler computational structure and correspondingly faster execution in practice.

Table 6: Average runtime (seconds) over 10 runs.

| Method | Student-$t$ | Beta | Laplace | Exponential |
|---|---|---|---|---|
| BLCE-G | 9.33 | 47.57 | 9.91 | 48.96 |
| BLCE | 3.56 | 8.16 | 4.11 | 7.96 |

### D.4 How the Performance of BLCE-G, BLCE, and BGLE Changes with the Allocation Rate $c$

We evaluate the performance of BLCE-G, BLCE, and BGLE over $T = 10{,}000$ rounds. For BLCE-G and BLCE, we set $K = 1000$ arms with feature dimension $d = 5$, where the arm contexts are independently sampled from either a $d$-dimensional uniform or normal distribution. For BGLE, we set $K = 50$ arms with feature dimension $d = 3$, again sampled independently from either a $d$-dimensional uniform or normal distribution. At the beginning of each run, the parameter vector $\theta^*$ is sampled from a $d$-dimensional normal distribution. For each algorithm, we vary the within-batch allocation rate $c \in \{0.1, 0.3, 0.5, 0.7, 0.9\}$ and compare the resulting performance.

As in Appendix D.1, we report cumulative regret, zoomed-in regret, and batch complexity averaged over 10 independent runs for BLCE-G and BLCE. For BGLE, following Appendix D.2, we report cumulative regret and batch complexity averaged over 20 independent runs.

As shown in Figure 5 and Figure 6, decreasing the value of $c$ consistently improves the empirical performance of BLCE-G, BLCE, and BGLE. This indicates that more greedy within-batch arm selection tends to yield better regret. Our theoretical analysis only requires that $c$ be chosen in the interval $(0, 1]$ and does not predict how performance varies with $c$. Understanding why smaller $c$—that is, more greedy selection—leads to improved performance in practice remains an interesting

direction for future work. The empirical findings suggest that, under stochastic contextual variation, prioritizing greedy exploitation may already provide strong performance even in the batched setting. Developing a rigorous mathematical explanation for this behavior would be an important step toward a deeper theoretical understanding of batched contextual bandits.

# E    DISCUSSION OF WITHIN-BATCH CONTEXT USAGE

This appendix clarifies the role of within-batch context information in batched (generalized) linear contextual bandits and explains why our comparisons with existing methods are methodologically fair, even though some prior algorithms do not explicitly use within-batch contexts in their policies.

**Our formulation.**    Our algorithms (BLCE-G, BLCE, and BGLE) operate in a standard batched / limited-adaptivity setting. The time horizon is partitioned into batches, and during each batch the agent observes the context at every round and is allowed to select actions as a function of all contexts observed so far and previous batch rewards.

By contrast, several recent algorithms for batched linear contextual bandits (Ruan et al., 2021; Hanna et al., 2023a;b; Zhang et al., 2025) choose not to exploit within-batch context information in their policy updates: their policies within each batch are designed to depend only on reward and context information from past batches together with the current context. Importantly, this is an *algorithmic design choice*, not a different information structure; all of these works still assume that the agent observes the context at every round.

**Evidence from existing formulations.**    The distinction between restricting reward adaptivity and restricting context usage is already visible in prior formulations of batched bandits.

Ruan et al. (2021) formally define both a *batch-learning* model and a *rare-policy-switch* model. In their batch-learning model, the time horizon is partitioned into batches (according to a static grid), the policy remains fixed throughout each batch, and rewards from that batch are revealed only at the batch boundary. The rare-policy-switch model relaxes this by allowing the learner to change its policy a limited number of times, and measures adaptivity by the number of such switches. In Section 1.1, Ruan et al. (2021) explicitly identify the Rarely Switching OFUL (RS-OFUL) algorithm of Abbasi-Yadkori et al. (2011, Section 5.1) as an instance of the rare-policy-switch model. RS-OFUL determines batch boundaries in an adaptive way (based on the determinant of the Gram matrix) and updates the Gram matrix at every round, so its action sequence within a batch depends on the full history of observed contexts, while rewards from the current batch are only incorporated at batch boundaries. Nevertheless, Ruan et al. (2021) treat RS-OFUL as a prototypical example of a batched algorithm.

A similar perspective appears in Hanna et al. (2023b), who define a batch-learning setting in which the learner must split the horizon into a fixed number of batches, use a pre-determined policy within each batch, and only update this policy using outcome feedback at batch boundaries. In their related work discussion, Hanna et al. (2023b) clearly classify RS-OFUL as a batched bandit algorithm under this formulation.

Yu & Oh (2025), the most recent work on batched (non-contextual) linear bandits, adopt a closely related definition: in their Section 3.2, a batch is a contiguous block of rounds during which the agent must use a fixed policy, and reward feedback from the batch is only incorporated at the end of the block. Yu & Oh (2025) also include RS-OFUL as a batched algorithm in their comparisons, both theoretically and empirically.

Taken together, these works show that (i) the standard batched / limited-adaptivity framework constrains when reward feedback is used, and (ii) algorithms such as RS-OFUL, whose actions within a batch depend on all observed contexts, are nonetheless regarded as canonical batched methods. Our algorithms follow exactly this information structure and, in fact, are even *less* adaptive than RS-OFUL, since we only update at pre-specified batch boundaries.

**Broader evidence from the batched bandit literature.**    Allowing policies to depend on all observed contexts within a batch, while incorporating only past-batch rewards, is common in the broader batched bandit literature. This problem setting appears in Abbasi-Yadkori et al. (2011);

Gao et al. (2019); Ren et al. (2020); Han et al. (2020); Kalkanli & Ozgur (2021); Karbasi et al. (2021); Hanna et al. (2023a); Ren & Zhou (2024); Gu et al. (2024); Jiang & Ma (2025b;a), among others. These works are published in top venues and come from well-established research groups in the bandit community. Sequential dependence on contexts within a batch is therefore neither unusual nor non-standard; rather, it is a natural and practically motivated modeling choice that fits squarely within the batched framework.

**Implications for comparisons and computational considerations.**    Within this common framework, there are at least two widely used algorithmic design patterns:

1. policies that remain fixed within each batch and are updated only at batch boundaries using past-batch rewards; and

2. policies that may depend on all contexts observed within a batch, while still updating only at batch boundaries using past-batch rewards.

Both patterns reduce adaptivity by limiting how often reward feedback can influence the policy. The algorithms in Ruan et al. (2021); Hanna et al. (2023a;b); Zhang et al. (2025) fall into the first category, whereas RS-OFUL (Abbasi-Yadkori et al., 2011) and our BLCE-G, BLCE, and BGLE algorithms fall into the second. The difference is therefore algorithmic rather than a change of problem setting.

We also note that Zhang et al. (2025), a recent batched linear contextual bandit method, must maintain $\mathcal{O}(d \log T)$ matrices and all previous batch estimators at each time step, whereas our methods maintain only a single Gram matrix per step together with batch-wise estimates. In both cases, however, the algorithms observe and process the context at every round and select actions at each step. Imposing an additional constraint that the policy *must* ignore readily available within-batch contexts (while still not using current-batch rewards) would increase computational and memory costs for many existing algorithms and appears more like an artificial restriction than a fundamental property of the batched framework.

Therefore, allowing our algorithms to use readily available within-batch context information does not change the underlying batched setting, nor does it trivialize the problem. Rather, it aligns with a well-established and widely accepted formulation in the batched bandit literature.

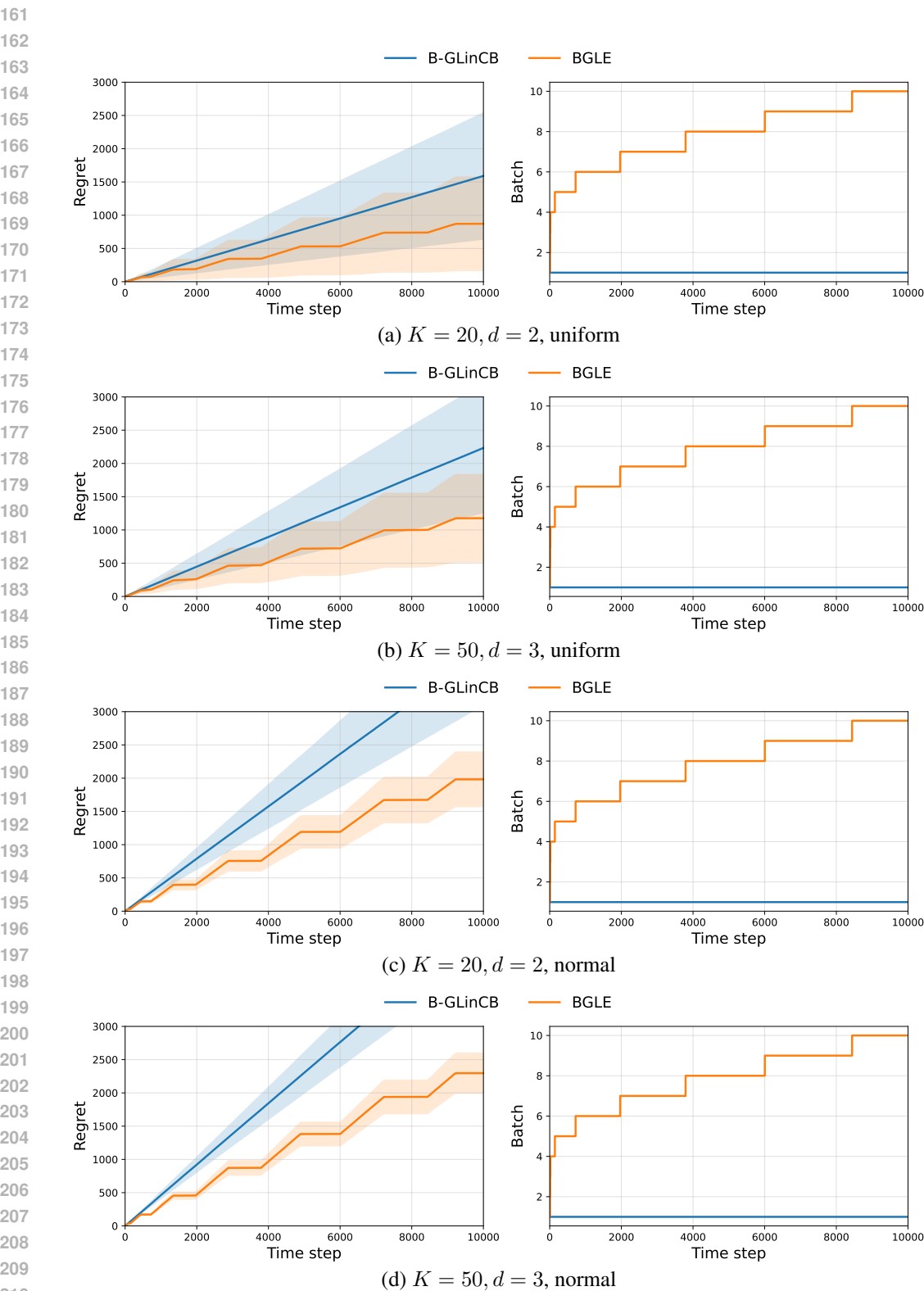

Figure 3: Regret and batch complexity over time for different values of $K$ and $d$.

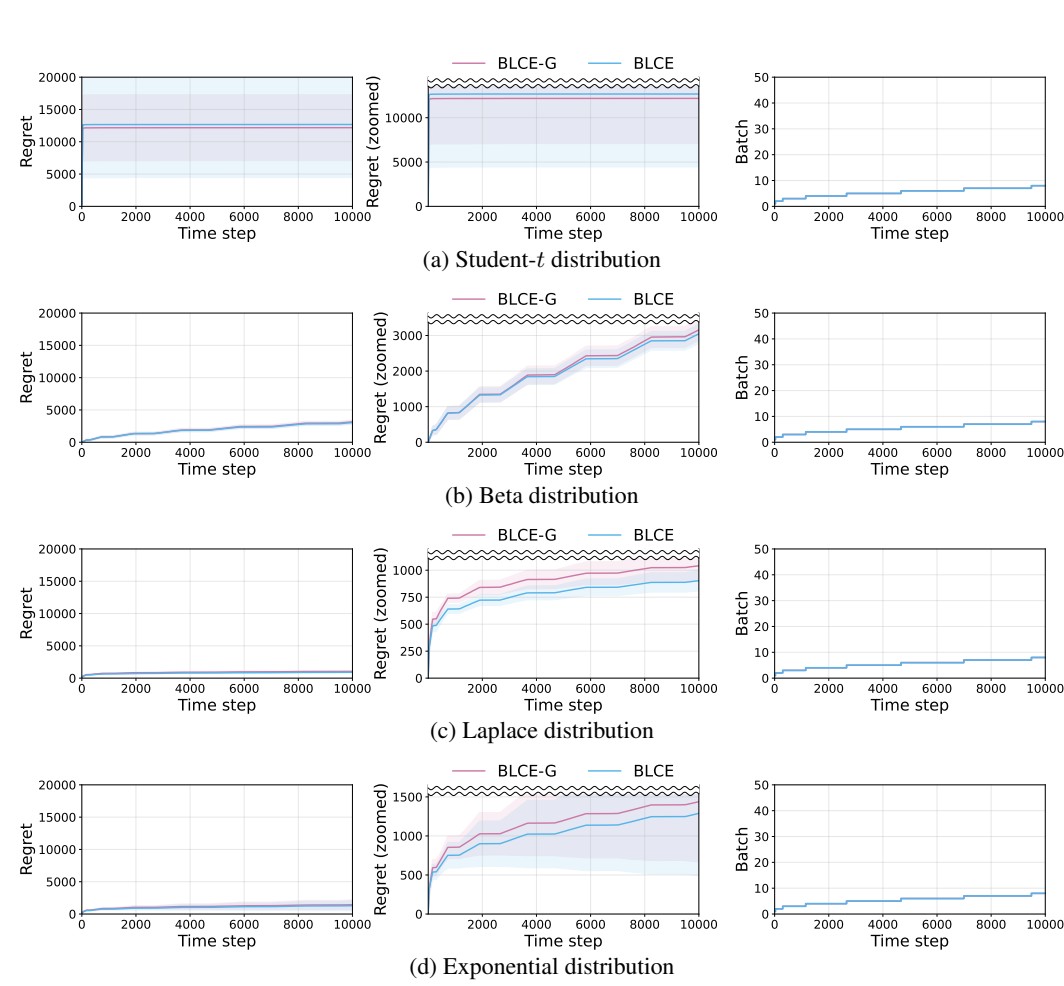

Figure 4: Regret, zoomed-in regret, and batch complexity over time for various context distributions.

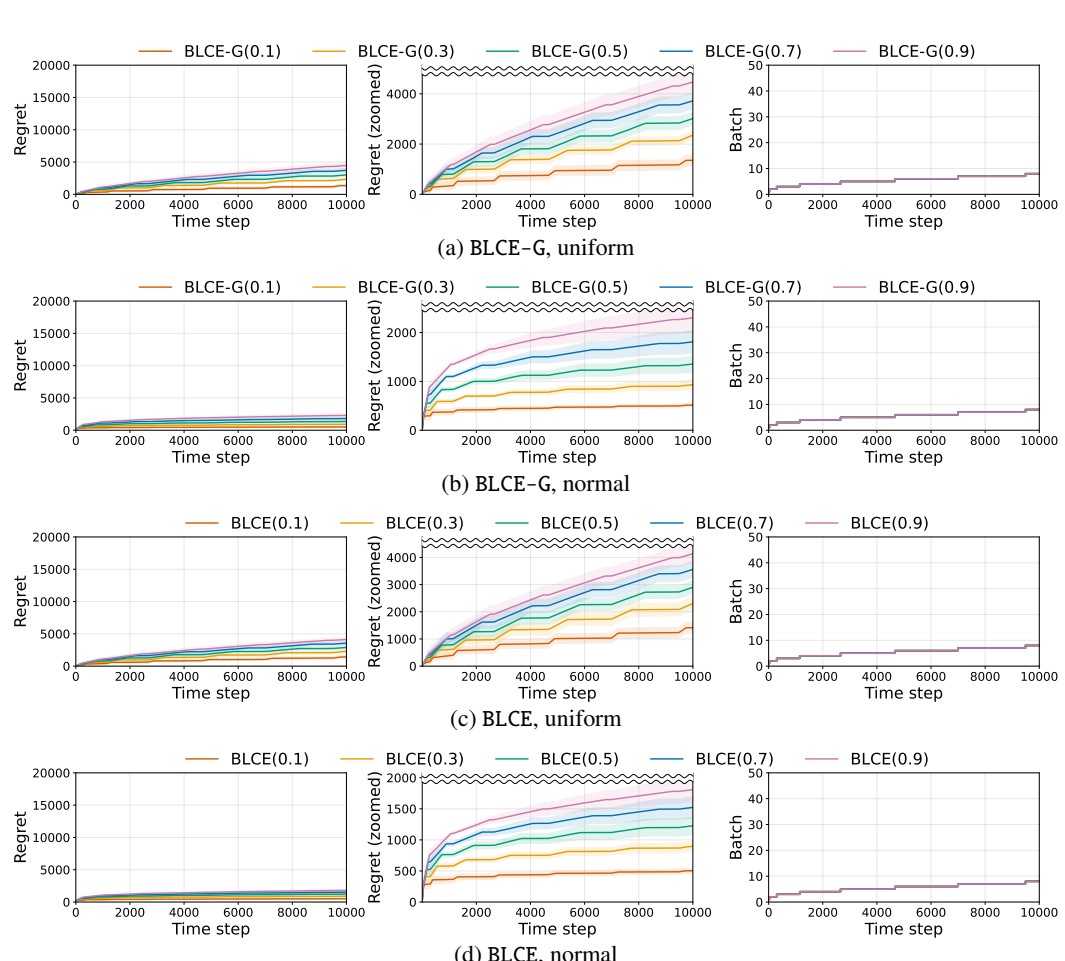

Figure 5: Regret and batch complexity over time for different values of $c$.

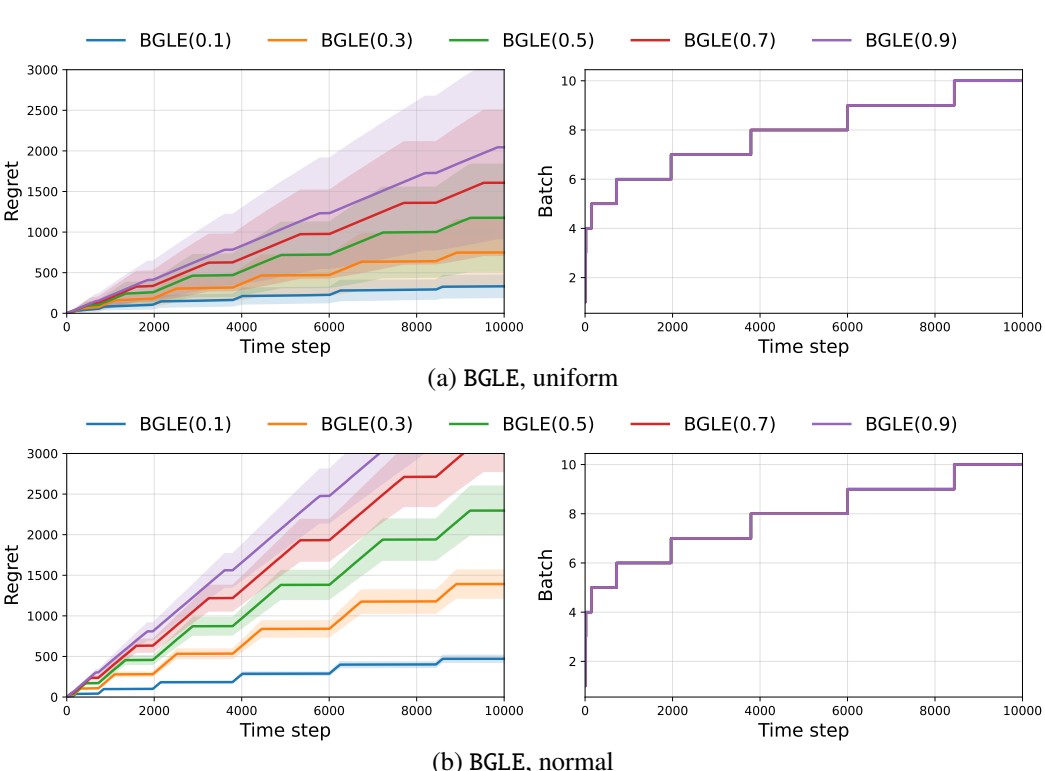

Figure 6: Regret and batch complexity over time for different values of $c$.

