# OpenReview forum: "Optimal Batched (Generalized) Linear Contextual Bandit Algorithm"
_ICLR.cc/2026/Conference — Submitted to ICLR 2026_

### Official Review · Reviewer_np75 · 2025-10-27

**Soundness:** 3
**Presentation:** 3
**Contribution:** 3
**Rating:** 6
**Confidence:** 2

**Summary:**

This paper studies the problem of learning in batched linear and generalized linear contextual bandits, where decision updates can only occur at a limited number of time points. The goal is to design algorithms that preserve the statistical efficiency of fully adaptive methods while remaining computationally practical under restricted adaptivity.

To this end, the authors develop three algorithms—BLCE-G, BLCE, and BGLE—that together achieve near-optimal regret with minimal batch complexity. The first method, BLCE-G, integrates a near G-optimal design with an arm-elimination procedure and achieves an regret matching the known minimax lower bounds across both small- and large-$K$ regimes while requiring only $O(loglog T)$ batches.
Building upon this, BLCE removes the computationally expensive G-optimal design step and replaces it with an uncertainty-driven exploration mechanism. Remarkably, this simplification does not compromise statistical performance: BLCE retains the same minimax-optimal regret while reducing the overall computational cost to $O(Kd^2 T log log T)$, which is asymptotically the lowest among existing optimal algorithms.

The third contribution, BGLE, generalizes the approach to generalized linear contextual bandits, where rewards depend on a nonlinear link function. A key innovation lies in removing all dependence on the curvature parameter κκ, a challenging quantity that governs the local geometry of the link function. The corresponding regret bound is the tightest known for batched generalized linear bandits and is entirely κκ-free in both leading and transient terms.

Empirical results further support these theoretical claims: the proposed algorithms consistently outperform existing batched bandit methods in cumulative regret and runtime efficiency across diverse regimes. Overall, the paper offers the first unified framework that achieves minimax-optimal regret, $O(log log T)$ adaptivity, and practical computational scalability for both linear and generalized linear contextual bandits.

**Strengths:**

**Originality**
The paper’s originality is mainly from combining theoretical optimality with computational practicality for batched contextual bandits. Previous methods achieving minimax regret usually relied on computationally heavy G-optimal design or assumed access to curvature parameters (e.g. $\kappa$) as algorithm input in the generalized linear setting. Regarding the proposed algorithms: BLCE eliminates G-optimal design, and BGLE removes dependence on the curvature constant $\kappa$.

**Quality**
This is a theoretical paper with good quality. The regret analysis seems complete and well matched with existing lower bounds in the related field. The mathematical proof indeed builds on prior techniques, but it also extends existing techniques to cover new cases such as the $\kappa$-free GLM analysis. The experimental evaluation on synthetic data corroborates with the statistical theory and provides credible empirical validation of both statistical and computational claims. Overall, the quality of the paper is good.

**Clarity**
The paper is clearly written and well structured. The motivation is clear and directly linked to practical limitations of existing literature in the field. Key results are presented in a clear way. For example, tables containing comparison of regret, batch complexity, and runtime are presented. Algorithm pseudocode is not too hard to understand.

**Significance**
The work has significance for both theoretical contributions in bandit, an important topic in machine learning and statistical theory. Importantly, it effectively closes an open gap in the literature by showing that minimax-optimal regret can be achieved under minimal adaptivity without relying on computationally infeasible steps.
The result also guides the direction for real world applications. To be specific, the removal of G-optimal design and the independence from $\kappa$ make the proposed algorithms very implementable in important real-world settings including statistical decision systems (e.g. recommendation systems or A/B testing platforms) that require good model generalization subject to limited adaptivity. The paper seems of interest to bandit theorists and also practitioners working on sequential decision-making under resource constraints.

**Weaknesses:**

I think this is a fine paper without major technical weakness. However, the following could be noted.

(1) The theoretical contribution is a bit incremental. The minimax-optimal regret and double-log batch complexity are existing results. I feel the major novelty of the paper’s is removing G-optimal design and achieving $\kappa$-independence. As mentioned in the ‘Strengths’ section, I think these are good refinement that have quite significant implications for real-world use cases.
Therefore, I feel it could be more appropriate to position the paper as a clever technical refinement.

(2) Absence of proof sketches or intuition.
Theoretical arguments are hard to follow. I think some proof sketches highlighting the key ideas behind Theorems 1–3 would improve readability.

**Questions:**

(1) The synthetic experiments show that the regret for BLCE-G and BLCE are very similar. This means the G-optimal step adds little in their specific experimental setup. I am curious under what setups one significantly outperforms the other.

(2) Regarding $\kappa$-independence: The proposed algorithm BGLE is shown to be $\kappa$-free, but in practice the bound still involves the expected inverse curvature $\hat{\kappa}$. In saturated regimes where $\mu’(<x, \theta^*>)$ is small, $\hat{\kappa}$ can still be large and may degrade the constant factor in the regret. This means $\kappa$ can still have an indirect impact on the statistical guarantee, is the understanding correct?

(3) The theory defines $\hat{\kappa}$ using the curvature at the optimal arm, but in practice, non-optimal arms might have very different curvature. Does the regret guarantee assume that curvature does not vary too sharply across the arm set?

(4) The algorithms rely on an exploration–exploitation split controlled by $c$. How is this parameter chosen, and how does it affect the empirical regret?

---

> ### Author Response · Authors · 2025-11-21
>
> We sincerely thank you for taking the time to review our paper and for providing thoughtful and valuable feedback. We greatly appreciate your recognition of our work and your constructive comments. Below, we address each of your comments and questions in detail:
>
> ---
>
> ## **Proof Sketches of Theorems**
>
> For the batched linear contextual bandit setting, our elimination rule builds on the arm elimination framework of Yu & Oh [1]. Under our elimination rule, it suffices to control the quantity $\sum_{t=\mathcal{T}\_{\ell}+1}^{\mathcal{T}\_{\ell+1}}\mathbb{E}\left[\max_{y \in \mathcal{A}\_t^{(\ell-1)}}y^\top V_{\ell}^{-1}y\right]$. Because the contexts are drawn i.i.d., the elimination rule is monotone across batches, and $\ell$-th batch information matrix $V_\ell$ always dominates (in the PSD order) the intermediate matrices $H_s$ for all $s \in [\mathcal{T}\_{\ell-1}+1,\mathcal{T}\_{\ell}]$, we can compare the $(\ell+1)$-th batch quantity $\sum_{t=\mathcal{T}\_{\ell}+1}^{\mathcal{T}\_{\ell+1}}\mathbb{E}\left[\max_{y \in \mathcal{A}\_t^{(\ell-1)}}y^\top V_{\ell}^{-1}y\right]$ with the $\ell$-th batch quantity $\sum_{s=\mathcal{T}\_{\ell-1}+1}^{\mathcal{T}\_{\ell}'}\mathbb{E}\left[\max_{y \in \mathcal{A}\_s^{(\ell-1)}}y^\top H_{s-1}^{-1}y\right]$. During Phase 1 of BLCE, the algorithm selects $x_{s,a_s} \in \arg\max_{x\in\mathcal{A}\_s^{(\ell-1)}} x^\top H_{s-1}^{-1}x$, so the $(\ell+1)$-th batch quantity is further upper-bounded by $\sum_{s=\mathcal{T}\_{\ell-1}+1}^{\mathcal{T}\_{\ell}'}\mathbb{E}\left[x_{s,a_s}^\top H_{s-1}^{-1}x_{s,a_s}\right]$. This is exactly the setting where the elliptical potential lemma applies, yielding an $\mathcal{O}(d\log T)$ bound. Hence, the key exploration term can be controlled without explicitly constructing a G-optimal design, which shows that **G-optimal design is not essential** for achieving minimax-optimal regret.
>
> On the other hand, when we employ a (near) G-optimal design together with an i.i.d. matrix concentration inequality, we can ensure that the Gram matrix in Phase 2 of BLCE-G does not grow excessively. This yields a sharper bound of $\mathcal{O}(d\log d)$, which explains why the regret bounds of BLCE and BLCE-G differ only by a $\log T$ versus $\log d$ factor.
>
> For the batched generalized linear contextual bandit setting, we develop a novel arm elimination rule inspired by Yu & Oh [1]. In this setting, we must control a curvature-weighted quantity of the form $\sum_{t=\mathcal{T}\_{\ell-1}+1}^{\mathcal{T}\_{\ell}}\mathbb{E}\left[\dot\mu(z_t)\max_{y \in \mathcal{A}\_t^{(\ell-2)}}y^\top V_{\ell-1}^{-1}y\right]$, where each $z_t$ lies between $\langle x_{t,a_t}, \theta^\star \rangle$ and $\langle x\_t^\star, \theta^\star \rangle$. A direct application of the elliptical potential lemma is not possible because the unknown curvature weight $\dot\mu(z_t)$ prevents us from constructing the weighted design matrix $\sum_t\dot\mu(z_t)x_{t,a_t}x_{t,a_t}^\top$.
>
> To address this difficulty, we upper bound $\dot\mu(z_t)$ using $\dot\mu(\langle x_t^\star,\theta^\star \rangle)$ term, which is i.i.d. due to the i.i.d. nature of contexts. This allows us to compare $\mathbb{E}\left[\dot\mu(\langle x_t^\star,\theta^\star \rangle)\max\_{y \in \mathcal{A}\_t^{(\ell-2)}}y^\top V_{\ell-1}^{-1}y\right]$ with the corresponding quantity from the previous batch, as similar as in the linear case. Next, we relate the $\dot\mu(\langle x_t^\star,\theta^\star \rangle)$ to the update weight used in the previous batch via $\alpha_{t,\ell-2}(\lambda)\dot\mu(\langle x_{s,a_s}, \hat\theta_{\ell-2} \rangle)$ which is precisely the weight governing the $(\ell-2)$-th batch information matrix. This correspondence allows us to again apply the elliptical potential lemma.
>
> Finally, using the boundedness of both $\hat\theta$ and $\theta^\star$, we carefully separate the resulting approximation error and show that all exponential factors can be absorbed into the transient part of the regret, without affecting the leading term.
>
> >[1] Sanghoon Yu and Min-hwan Oh. *Optimal and practical batched linear bandit algorithm.* In The Forty-Second International Conference on Machine Learning, 2025.
>
> ---

---

> ### Author Response · Authors · 2025-11-21
>
> ---
>
> ## **Conditions Under Which BLCE-G or BLCE Dominates the Other**
>
> To further investigate the empirical differences between BLCE-G and BLCE, we conducted four additional experiments using fixed context distributions: Student-$t$, Beta, Laplace, and Exponential distributions. These results are now included in Appendix Section D.3.
>
> Across all distributions tested—including uniform and normal distributions from our paper—BLCE-G and BLCE exhibit highly similar regret performance. In most settings, BLCE achieves slightly lower regret, whereas in the heavy-tailed Student-$t$ distribution, this difference nearly vanishes and BLCE-G matches or occasionally outperforms BLCE. While the theory guarantees that BLCE-G has a strictly smaller worst-case regret, these guarantees characterize worst-case regimes and do not necessarily arise in benign stochastic environments. Thus, the empirical results do not contradict the theoretical results.
>
> Despite conducting extensive experiments across a wide range of distributions, we were not able to find a regime in which one algorithm consistently and significantly outperforms the other. Based on our observations, neither BLCE-G nor BLCE dominates uniformly across distributions, we believe their empirical performance remains comparable in most practical settings.
>
> ---
>
> ## **Does a Large $\hat{\kappa}$ in Saturated Regime Degrade the Regret?**
>
> We respectfully clarify that a **large $\hat{\kappa}$ does not degrade the regret** in BGLE. In our regret bound, $\hat{\kappa}$ appears only in the leading term, and the dependence is through the factor $\sqrt{1/\hat\kappa}$. Thus, a larger $\hat{\kappa}$ reduces the regret upper bound—representing a more favorable regime. As the reviewer noted, in saturated regimes where $\dot\mu(\langle x, \theta^* \rangle)$ is small, $\hat{\kappa}$ indeed becomes large, but this directly improves the regret guarantee rather than worsening it.
>
> Notably, even in the fully adaptive generalized linear bandit literature, the recent work of Zhang et al. [1] still incurs $\kappa$-dependence in transient term, which indeed deteriorates in saturated regimes, exactly as the reviewer mentioned. In contrast, despite operating under limited adaptivity, the batched setting benefits from the i.i.d. nature of contexts—a structural advantage absent in fully adaptive (and potentially adversarial) settings—which allows us to obtain a completely $\kappa$-independent regret bound even with weaker adaptivity.
>
> >[1] Yu-Jie Zhang, Sheng-An Xu, Peng Zhao, and Masashi Sugiyama. *Generalized linear bandits: Almost optimal regret with one-pass update.* The Thirty-Ninth Annual Conference on Neural Information Processing Systems, 2025.
>
> ---

---

> ### Author Response · Authors · 2025-11-21
>
> ---
>
> ## **Assumptions on Curvature Variation Across the Arm Set**
>
> We first clarify that our regret guarantee does not require the curvature to be similar across arms. Recall that we define $\kappa := \max_{\mathcal{A}\in \mathrm{supp}(\mathcal{D})} \max_{x\in \mathcal{A}} \frac{1}{\dot{\mu}(\langle x,\theta^\star\rangle)}$ as a global upper bound on curvature, and $\hat{\kappa} := \frac{1}{ \mathbb{E}_{\mathcal{A}\sim\mathcal{D}}[\dot{\mu}(\langle x^\star,\theta^\star\rangle)]}$, where $x^\star$ denotes the optimal arm within $\mathcal{A}$. Thus $\hat{\kappa}$ depends only on the curvature at the optimal arm (averaged over $\mathcal{A}\sim\mathcal{D}$), while $\kappa$ captures the global worst-case curvature over all (possibly highly suboptimal) arms.
>
> A key technical challenge in generalized linear contextual bandits is that the Fisher information depends on the $\dot\mu$-weighted Gram matrix. To remove all dependence on the global curvature bound $\kappa$, this $\dot\mu$-weighted Gram matrix must be properly updated within batches, and all quantities in the analysis must be controlled with respect to this matrices. In prior batched work (Sawarni et al. [1]), the information term is upper bounded using the first batch's unweighted (weight-1) Gram matrix for all subsequent batches. This discards curvature information and inevitably leads to $\kappa$-dependent regret bounds.
>
> To eliminate any $\kappa$-dependence in both the algorithm and the regret analysis, we deliberately split the warm-up phase into three batches. In Batch 1, where curvature information is unavailable, we update only the unweighted Gram matrix. In Batch 2, we use information collected in Batch 1 to begin updating the $\dot\mu$-weighted Gram matrix. In Batch 3, we apply the arm elimination rule while continuing to update the $\dot\mu$-weighted Gram matrix. By combining the three-batch warm-up structure with the i.i.d. nature of the contexts, our elimination rule guarantees that later batches never need to rely on Batch 1 quantities, thereby removing any dependence on the global curvature bound $\kappa$ in the final regret guarantee.
>
> **Importantly, at no point in the analysis do we require the curvature of suboptimal arms to resemble that of the optimal arm.**
>
> >[1] Ayush Sawarni, Nirjhar Das, Siddharth Barman, and Gaurav Sinha. *Generalized linear bandits with limited adaptivity.* Advances in Neural Information Processing Systems, 37:8329–8369, 2024.
>
> ---
>
> ## **How the Within-Batch Allocation Rate $c$ Affects Empirical Regret**
>
> In our numerical experiments, we use $c=0.5$, meaning that each batch allocates half of its rounds to exploration and the remaining half to exploitation. To examine how this parameter influences empirical performance, we additionally evaluate BLCE-G, BLCE, and BGLE over $T = 10{,}000$ rounds across a range of values $c \in [0.1, 0.3, 0.5,0.7,0.9] $. The full results are provided in Appendix Section D.4.
>
> For BLCE-G and BLCE, we consider $K = 1000$ arms with feature dimension $d = 5$, where contexts are independently sampled from either a $d$-dimensional uniform or normal distribution. For BGLE, we set $K = 50$ and $d = 3$ with the same sampling scheme.
>
> As shown in Figure 5 and 6, smaller values of $c$ consistently yield better empirical regret across all three algorithms. This suggests that more greedy within-batch selection tends to perform better in practice. Our theoretical guarantees only require choosing any $c \in (0,1]$ and do not prescribe how the regret should vary with this parameter.  Understanding why more aggressive exploitation (i.e., smaller $c$) improves performance under stochastic contexts appears to be an interesting direction for future work. These findings indicate that, in stochastic contextual environments, prioritizing greedy selection may already offer strong performance even in the batched setting.
>
> ---

---

> ### Author Response · Authors · 2025-11-21
>
> ---
>
> ## **Theoretical Contribution in Batched (Generalized) Linear Contextual Bandits**
>
> We are happy to provide clarification on our theoretical contribution, which we believe is significant. In the batched linear contextual bandit setting, our regret bounds improve over prior work in both the large-$K$ regime $(K \geq \Omega(e^d))$ and the small-$K$ regime $(K \leq \mathcal{O}(e^d))$. To the best of our knowledge, no existing algorithm **achieves minimax-optimal regret in both regimes simultaneously;** thus, no prior work provides optimal guarantees across the full range of arm sizes $K$. For example, Zhang et al. [1]—the most recent work on batched linear contextual bandits—obtains regret bounds that are actually worse than earlier results in the small-$K$ regime. Their primary contribution is a partial reduction in the use of G-optimal design for improved practicality. In contrast, our algorithms attain the tightest regret bounds in both regimes while being strictly more practical; in particular, BLCE entirely removes the need for any G-optimal design.
>
> We also note that Zhang et al. [1], despite offering improved time complexity relative to previous work, still requires maintaining $\mathcal{O}(d\log T)$ matrices and all previous batch estimators at every round, whereas our method maintains **only a single Gram matrix** per round together with the batch estimators. Furthermore, prior works (Zhang et al. [1], Ruan et al. [2], Hanna et al. [3,4]) fundamentally rely on G-optimal design and i.i.d.-based expectation inequalities—or reduce the contextual setting to a non-contextual one using i.i.d. assumptions—and therefore cannot relax the i.i.d. assumption, which can be somewhat restrictive in certain settings. In contrast, BLCE, which does not require G-optimal design nor i.i.d.-based inequalities, **can leverage the relaxed i.i.d. assumptions** described in Remark 1.
>
> In addition, prior batched algorithms (Zhang et al. [1], Sawarni et al. [5]) adopt batch schedules that depend on problem-specific parameters beyond the horizon $T$. This often leads to unstable empirical performance, most notably when the first batch becomes extremely large, forcing the agent to explore blindly for an extended period without receiving any reward information. In contrast, our algorithms introduce a **novel batch schedule** that depends solely on $T$, making the method both theoretically optimal and more practical, and resulting in far more predictable and stable behavior across a wide range of problem instances.
>
> Moreover, several prior works rely on elimination-driven exploration (Zhang et al. [1], Ruan et al. [2], Sawarni et al. [5]). When arm elimination does not proceed effectively, these methods can be pushed into overly exploratory behavior, again causing significant empirical instability. Our algorithms avoid this problem by **explicitly incorporating a greedy within-batch exploitation step**, which prevents excessive exploration and consistently yields strong empirical performance without sacrificing theoretical guarantees.
>
> Taken together—improved regret guarantees across all $K$ regimes, strictly lower computational and memory requirements, the ability to relax context assumptions, and substantially more stable and strong empirical performance without compromising any theoretical guarantees—we believe these contributions go well beyond a minor technical refinement.
>
> In the batched generalized linear contextual bandit setting, our improvements go well beyond removing the dependence on $\kappa$. More importantly, to the best of our knowledge, we present the first batched algorithm that achieves the minimax-optimal regret.
>
> The **removal of all $\kappa$-dependence** is a central component of this result. Even in the fully adaptive generalized linear bandit literature, the most recent work of Zhang et al. [6] still incurs $\kappa$-dependence in transient terms. By contrast, despite operating under limited adaptivity, the batched setting benefits from the i.i.d. nature of contexts—a structural advantage absent in fully adaptive (and potentially adversarial) settings—which allows us to obtain a completely $\kappa$-independent regret bound while maintaining weaker adaptivity.
>
> Furthermore, prior batched work could not reach the optimal $\widetilde{\mathcal{O}}(d\sqrt{T})$ regret due to unavoidable terms such as $(\sqrt{d/\hat\kappa} \wedge \sqrt{R_{\dot\mu}\log d})$. Our algorithm BGLE overcomes this limitation: it **achieves the minimax-optimal regret** and reduces computational cost by eliminating all reliance on G-optimal design.
>
> ---

---

> ### Author Response · Authors · 2025-11-21
>
> >[1] Zihan Zhang, Xiangyang Ji, and Yuan Zhou. *Almost optimal batch-regret tradeoff for batch linear contextual bandits.* In The Thirteenth International Conference on Learning Representations, 2025.
> >[2] Yufei Ruan, Jiaqi Yang, and Yuan Zhou. *Linear bandits with limited adaptivity and learning distributional optimal design.* In Proceedings of the 53rd Annual ACM SIGACT Symposium on Theory of Computing, pp. 74–87, 2021.
> >[3] Osama A Hanna, Lin Yang, and Christina Fragouli. *Contexts can be cheap: Solving stochastic contextual bandits with linear bandit algorithms.* In The Thirty Sixth Annual Conference on Learning Theory, pp. 1791–1821. PMLR, 2023b.
> >[4] Osama Hanna, Lin Yang, and Christina Fragouli. *Efficient batched algorithm for contextual linear bandits with large action space via soft elimination.* Advances in Neural Information Processing Systems, 36:56772–56783, 2023a.
> >[5] Ayush Sawarni, Nirjhar Das, Siddharth Barman, and Gaurav Sinha. *Generalized linear bandits with limited adaptivity.* Advances in Neural Information Processing Systems, 37:8329–8369, 2024.
> >[6] Yu-Jie Zhang, Sheng-An Xu, Peng Zhao, and Masashi Sugiyama. *Generalized linear bandits: Almost optimal regret with one-pass update.* The Thirty-Ninth Annual Conference on Neural Information Processing Systems, 2025.
>
> ---
>
> Please don’t hesitate to reach out if you have any further questions!

---

> ### Comment · Reviewer_np75 · 2025-11-26
> **Re: rebuttal**
>
> Thank the authors for their detailed rebuttal. The authors’ explanations adequately addressed my concerns.
>
> The proof sketches clarified how the elimination rules in both the linear and generalized linear settings avoid requiring explicit G-optimal designs. Their use of batch monotonicity and the elliptical potential lemma made the key steps clear, and the treatment of curvature weights in the GLM setting resolved the technical questions I raised.
>
> The additional experiments comparing BLCE and BLCE-G seem helpful to me. The authors’ clarification that neither algorithm uniformly dominates the other in practice—despite BLCE-G having a tighter worst-case guarantee—makes the empirical results corroborate with the theory.
>
> The discussion regarding large in saturated regimes was also useful. The distinction between transient-term degradation in fully adaptive settings and the -independent guarantees achievable in the batched i.i.d. setting clarified why the regret does not worsen.
>
> Overall, the rebuttal satisfactorily addressed all comments. I will maintain my positive assessment and give a score of 8 and recommend the paper for acceptance.

---

> > ### Author Response · Authors · 2025-11-26
> >
> > We sincerely thank the reviewer for the careful reading of our paper and for the continued and strong support. We are glad that our response clarified your concerns/questions.
> > We are grateful for your recommendation for acceptance and will incorporate your feedback into the final version.

---

### Official Review · Reviewer_4X1Q · 2025-10-30

**Soundness:** 3
**Presentation:** 3
**Contribution:** 4
**Rating:** 8
**Confidence:** 3

**Summary:**

The paper considers batched linear and generalized linear contextual bandits.
For linear contextual bandits, the proposed algorithm (BLCE-G) achieves the minmax regret up to log factors in both large and small number of actions, and only uses $O(\log \log T)$ batches.
The second algorithm (BLCE) does not require the computation of G-opt design while keeping the statistical guarantees. They further study the generalized linear contextual bandits. The proposed algorithm (BGLE) works without knowledge of the curvature parameter $\kappa$ and the regret bound is independent of it. The algorithm again only uses $O(\log \log T)$ batches and attains the nearly-optimal regret.

**Strengths:**

1. BLCE-G is the first algorithm to simultaneously match the minimax lower bounds in both large $K$-regime and the small $K$ regime up to logarithmic factors.

2. Keeping $O(\log log T)$ batches and min-max regret for both contextual linear and generalized linear bandits improve over existing results.

3. Empirical performance is promising in various regimes $(K,d)$.

**Weaknesses:**

The result is the extension of Yu & Oh (2025) and see questions below.

**Questions:**

1: Yu & Oh (2025) is one of the most related works. The paper achieves, in the linear bandit,  the minimax regret by integrating arm elimination with a regularized G-optimal design. I understand that the different techniques are required to handle contextual setting, and also remove the G-optimal design in BLCE. It is surprising that results are almost similar with non-contextual settings. I could not fully understand why removing G-optimal design is still fine with theoretical guarantees . You are writing “BLCE instead lengthens the uncertainty-driven exploration phase to occupy those rounds.” in line 257, but I would appreciate a more intuitive and also rigorous explanation behind it.

In the regret analysis, we need to control $\sum_{s} E[ \max_{y \in A^{\ell-1}} y ^{\top} H^{-1}_{s-1}y]$ (as in (8)) and we need to care about making Gram matrix not to grow excessively during Phase 2 of the algorithm. In order to have such a “good”  random event (line 952), employing G-optimal design seems essential.

If I understand correctly, (13) is enough to remove G-optimal design and somehow surprising as the RHS in (13) is $O(d \log T)$ while in (8) the bound (dominating term) is $ $O(\frac{d \log T}{T})$.
Can you comment on the critical idea in analysis for removing G-opt?
It may also be better to add the proof sketch in the main text.

2: What is the computational bottleneck of Zhang et al. (2025) which has time complexity dependent on $d^{7/2}$?

---

> ### Author Response · Authors · 2025-11-21
>
> We sincerely thank you for taking the time to review our paper and for providing thoughtful and valuable feedback. We greatly appreciate your recognition of our work and your constructive comments. Below, we address each of your comments and questions in detail:
>
> ---
>
> ## **Why Removing G-optimal Design Still Preserves Theoretical Guarantees in the Batched Linear Setting**
>
> We are more than happy to elaborate on this.
> For the batched linear contextual bandit setting, our elimination rule builds on the arm elimination framework of Yu & Oh [1]. Under our elimination rule, it suffices to control the quantity $\sum\_{t=\mathcal{T}\_{\ell}+1}^{\mathcal{T}\_{\ell+1}}\mathbb{E}\left[\max\_{y \in \mathcal{A}\_t^{(\ell-1)}}y^\top V\_{\ell}^{-1}y\right]$. Because the contexts are drawn i.i.d., the elimination rule is monotone across batches, and the $\ell$-th batch information matrix $V\_\ell$ always dominates (in the PSD order) the intermediate matrices $H\_s$ for all $s \in [\mathcal{T}\_{\ell-1}+1,\mathcal{T}\_{\ell}]$, we can compare the $(\ell+1)$-th batch quantity $\sum_{t=\mathcal{T}\_{\ell}+1}^{\mathcal{T}\_{\ell+1}}\mathbb{E}\left[\max\_{y \in \mathcal{A}\_t^{(\ell-1)}}y^\top V\_{\ell}^{-1}y\right]$ with the $\ell$-th batch quantity $\sum_{s=\mathcal{T}\_{\ell-1}+1}^{\mathcal{T}\_{\ell}'}\mathbb{E}\left[\max\_{y \in \mathcal{A}\_s^{(\ell-1)}}y^\top H\_{s-1}^{-1}y\right]$. During Phase 1 of BLCE, the algorithm selects $x\_{s,a\_s} \in \arg\max\_{x\in\mathcal{A}\_s^{(\ell-1)}} x^\top H\_{s-1}^{-1}x$, so the $(\ell+1)$-th batch quantity is upper-bounded by $\sum\_{s=\mathcal{T}\_{\ell-1}+1}^{\mathcal{T}\_{\ell}'}\mathbb{E}\left[x\_{s,a\_s}^\top H\_{s-1}^{-1}x\_{s,a\_s}\right]$. This is exactly the setting where the elliptical potential lemma applies, yielding an $\mathcal{O}(d\log T)$ bound. Hence, the key exploration term can be controlled without explicitly constructing a G-optimal design, showing that **G-optimal design is not essential** for achieving minimax-optimal regret.
>
> On the other hand, when we employ a (near) G-optimal design together with an i.i.d. matrix concentration inequality, we can ensure that the Gram matrix in Phase 2 of BLCE-G does not grow excessively. This yields a sharper bound of $\mathcal{O}(d\log d)$, which explains why the regret bounds of BLCE and BLCE-G differ only by a $\log T$ versus $\log d$ factor. Additionally, the prior works (Ruan et al. [2], Hanna et al. [3,4], Zhang et al. [5]) fundamentally rely on G-optimal design and i.i.d.-based expectation inequalities—or reduce the contextual setting to a non-contextual one under i.i.d. assumptions—and therefore cannot relax the i.i.d. assumption, which can be somewhat restrictive in certain settings. In contrast, BLCE, which does not require G-optimal design nor i.i.d.-based inequalities, can leverage the relaxed i.i.d. assumptions described in Remark 1.
>
> Finally, we clarify a potential misunderstanding regarding equation (8): the dominating term is $\mathcal{O}(d\log d)$, not $\mathcal{O}(\frac{d\log T}{T})$. The term $\mathcal{O}(\frac{d\log T}{T})$ corresponds to the contribution when the good event $\mathbb{E}_\ell$ fails; this term is negligible, and since  $\frac{\log T}{T} =\mathcal{O}(\log d)$, it is absorbed into the $\mathcal{O}(d\log d)$ term.
>
> >[1] Sanghoon Yu and Min-hwan Oh. *Optimal and practical batched linear bandit algorithm.* In The Forty-Second International Conference on Machine Learning, 2025.
> >[2] Yufei Ruan, Jiaqi Yang, and Yuan Zhou. *Linear bandits with limited adaptivity and learning distributional optimal design.* In Proceedings of the 53rd Annual ACM SIGACT Symposium on Theory of Computing, pp. 74–87, 2021.
> >[3]  Osama A Hanna, Lin Yang, and Christina Fragouli. *Contexts can be cheap: Solving stochastic contextual bandits with linear bandit algorithms.* In The Thirty Sixth Annual Conference on Learning Theory, pp. 1791–1821. PMLR, 2023b.
> >[4] Osama Hanna, Lin Yang, and Christina Fragouli. *Efficient batched algorithm for contextual linear bandits with large action space via soft elimination.* Advances in Neural Information Processing Systems, 36:56772–56783, 2023a.
> >[5] Zihan Zhang, Xiangyang Ji, and Yuan Zhou. *Almost optimal batch-regret tradeoff for batch linear contextual bandits.* In The Thirteenth International Conference on Learning Representations, 2025.
>
> ---

---

> ### Author Response · Authors · 2025-11-21
>
> ---
>
> ## **Computational Bottleneck of Zhang et al. [1]**
>
> The computational bottleneck in Zhang et al. [1], which results in a time complexity scaling with $d^{7/2}$, arises entirely from the first batch. In their Algorithm 1, during the first batch the agent selects an arm **at every round using a G-optimal design**. As discussed in Section 4 of our paper, computing a (near) G-optimal design requires $\mathcal{O}(Kd^3)$ time. Meanwhile, the first batch in Zhang et al. [1] has size  $\mathcal{O}(\sqrt{dT\log(dKT)\log T})$. Multiplying these two factors yields a total computational cost of order $\mathcal{O}(Kd^{7/2}\sqrt{T\log(dKT)\log T})$. This explains the $d^{7/2}$-dependence in their overall complexity and highlights why removing the G-optimal design step leads to a substantial computational advantage in our algorithms.
>
>
> >[1] Zihan Zhang, Xiangyang Ji, and Yuan Zhou. *Almost optimal batch-regret tradeoff for batch linear contextual bandits.* In The Thirteenth International Conference on Learning Representations, 2025.
>
> ---
>
> Please feel free to reach out if you have any further questions or require clarification on any point!

---

### Official Review · Reviewer_rQ4C · 2025-11-01

**Soundness:** 3
**Presentation:** 3
**Contribution:** 1
**Rating:** 2
**Confidence:** 3

**Summary:**

This paper studies batched linear and generalized linear contextual bandits under limited adaptivity. The authors propose three algorithms: BLCE-G (Batched Linear Contextual Bandit with Elimination and near G-optimal design), BLCE (same algorithm without the G-optimal design), and BGLE (Batched Generalized Linear Bandit with Elimination). The claimed contributions are:
- The first linear contextual algorithms achieving minimax-optimal regret in both small-K and large-K regimes with only $O(\log \log T)$ batches,
- Removal of the G-optimal design step while preserving the same statistical guarantees and achieving lowest runtime,
- The first $\kappa$-independent algorithm for batched generalized linear bandits, whose regret depends only on the average curvature $\hat \kappa$ rather than the worst-case curvature $\kappa$.
Experiments show empirical improvements over prior batched baselines in regret and runtime.

**Strengths:**

Please see weakness.

**Weaknesses:**

The main issue of the paper is that the notion of "batched" in the paper differs from standard definitions in the literature. Section 3.2 claims the agent uses a "fixed policy within each batch", but Algorithms 1–2 adaptively update the Gram matrix $H_t$ at every round and select the arm with maximal matrix norm $\|x\|_{H{t-1}^{-1}}$. Thus, the action sequence within a batch depends on the full within-batch context history. In prior works (e.g., Ruan et al., 2021; Hanna et al., 2023; Zhang et al., 2025), "batched" meant memoryless within a batch: only decisions based on statistics from *previous* batches. Here, the algorithm is sequential within the batch. This difference invalidates a direct comparison with the "static grid" baselines summarized in Table 1, which potentially trivializes the problem.

**Questions:**

I have no further questions.

---

> ### Author Response · Authors · 2025-11-21
>
> We thank the reviewer for the time to review our paper and for the feedback. We recognize that there may be a difference in how the notion of "batched" bandits is understood, especially in the *contextual* setting. To our knowledge, the literature has not yet precisely converged on a single formal definition, particularly regarding what aspects of the policy must remain "fixed" within a batch. We therefore appreciate the opportunity to clarify our formulation and its relationship to prior work, and we hope that the discussion below contributes to a clearer and more unified understanding of batched contextual bandits within the existing literature.
>
> ---
>
> ## **Clarifying Our Use of "Fixed Policy Within Each Batch" and Revision**
>
> First, our phrase "fixed policy within each batch" was meant to convey that the *policy parameter* $\hat\theta$ is fixed throughout a batch and updated only at batch boundaries. Incorporating the reviewer's feedback that this wording could be interpreted more narrowly than intended, we have revised the text to make this meaning explicit in the updated version. We thank the reviewer for pointing this out (however, this clarification does not imply that the notion of "fixed" means ``memoryless''; rather, it is intended to avoid confusion by making explicit what is fixed). For a broader discussion of what previous literature considers as "fixed" or "batched", we refer the reviewer to the discussion below.
>
> Under this interpretation of a fixed parameter $\hat\theta$ per batch, updated only at batch boundaries, our algorithms BLCE-G, BLCE, and BGLE are fully consistent with our stated model and with what is commonly regarded as batched bandits in the existing literature, as we elaborate below.
>
> ---

---

> ### Author Response · Authors · 2025-11-21
>
> ---
>
> ## **Batched Bandits Do Not Mean "Memoryless Within a Batch"**
>
> As we begin this discussion, we kindly and sincerely ask the reviewer to consider it with an open mind, and we welcome any further constructive exchange. We would also be happy to incorporate a (concise) version of this discussion into the paper to help clarify a precise working definition of batched contextual bandits, which has so far remained largely implicit in the literature. While some algorithms may appear "memoryless" within a batch, as we discuss below, this memorylessness is a design choice rather than the defining criterion those papers use to characterize the batched setting, which can naturally lead to confusion for readers. A clear formulation and explicit discussion of these distinctions is therefore, in our view, both timely and beneficial for the community.
>
> **The existing batched bandit literature does not equate "batched" with "memoryless within a batch."** Rather, standard formulations and the previously published papers' scope constrain *when* reward feedback can be used to update the policy, without necessarily restricting the learner’s ability to process within-batch contexts (regardless of how specific algorithms are instantiated).
>
> **Evidence 1.**
> For example, Ruan et al. [1] formally define both a batch-learning model and a rare-policy-switch model.
> In the batch learning model, time steps are partitioned into pre-defined batches (a static grid) and the fixed policy is used throughout each batch, with rewards revealed only at the batch boundary. The rare policy switch model is a relaxation of batch learning, where the adaptivity cost is measured by the number of times the learner is allowed to change its action-selection policy.
> Thus, in the formulation of Ruan et al. [1], both batch learning and rare policy switches follow the fixed-policy-within-batch paradigm, differing only in how the batch boundaries are determined. In Section 1.1, Ruan et al. [1] explicitly identify the Rarely Switching OFUL (RS-OFUL) algorithm—introduced in Section 5.1 of Abbasi-Yadkori et al. [5] (a batched variant of the well-known OFUL algorithm)—as an instance of the rare policy switch model. RS-OFUL determines the end of a batch adaptively: it terminates a batch when the determinant of the Gram matrix accumulated up to the current round exceeds $(1+C)$ times the determinant at the previous batch boundary. Consequently, **RS-OFUL updates the Gram matrix at every round, and its within-batch action sequence depends on the full history of within-batch contexts**.
> Nonetheless, **Ruan et al. [1] classify RS-OFUL as a "fixed policy" method under their definition of rare policy switches**. Therefore, our previous statement in Section 3.2 of "a fixed policy within each batch" is not inconsistent with algorithms whose within-batch action sequence depends on the full context history (according to what Ruan et al. [1] defines as "fixed"), such as RS-OFUL and our BLCE-G, BLCE, and BGLE algorithms.
> Moreover, **Ruan et al. [1] do not define batch learning or rare policy switches as "memoryless", and doing so would contradict their own classification of RS-OFUL (Abbasi-Yadkori et al. [5]) as a rare policy switch model, since RS-OFUL is clearly not memoryless**.
>
> **Evidence 2.**
> A similar perspective is present in Hanna et al. [2] defines the notion of batch learning in Section 1.1. In their definition, instead of observing the reward at the end of each round to decide the next action, the learning agent is constrained to split the rounds into a fixed number of $M$ batches, use a predetermined policy within each batch, and it can only observe the action outcomes and switch the policy at the end of each batch. **Hanna et al. [2] clearly classify Abbasi-Yadkori et al. [5]'s RS-OFUL as a batched bandit in their related work section**.
>
> **Evidence 3.**
> In Yu & Oh [4], the most recent work in batched (non-contextual) linear bandits, batch learning is defined in Section 3.2 as a contiguous block of rounds during which the agent must use a fixed policy. In other words, within each batch, the agent is not able to update its policy, effectively deferring feedback from that entire batch until the end. **Yu & Oh [4] also include Abbasi-Yadkori et al. [5]'s RS-OFUL as a batched bandit in their related work section and performed comparisons both theoretically and empirically**.  (Therefore, our previous statement of"fixed policy within each batch” is likewise not inconsistent with algorithms whose within-batch action sequence depends on the full context history. Regardless of the text's interpretation, it is clear that the scope of the batched bandit includes methods incorporating within-batch contexts.) Moreover, none of these papers define batched bandits as "memoryless", and doing so would contradict their own classification of Abbasi-Yadkori et al. [5], since RS-OFUL is clearly not memoryless.

---

> ### Author Response · Authors · 2025-11-21
>
> Taken together, **these state-of-the-art works (Ruan et al. [1], Hanna et al. [2], and Yu \& Oh [4]) all classify RS-OFUL as a batched / limited-adaptivity algorithm. Since our algorithms are even less adaptive than RS-OFUL (we update $\hat\theta$ only at pre-specified batch boundaries), it is logical and consistent to regard our methods as batched as well.** Because RS-OFUL is recognized as a batched bandit method even while processing all within-batch contexts, using readily available within-batch context information, as we do, is fully aligned with existing practice.
> The prior works cited by the reviewer (Ruan et al. [1], Hanna et al. [2], Zhang et al. [3]) simply choose not to exploit within-batch contexts; however, this algorithmic choice does not change the underlying definition of the batched / limited-adaptivity setting. Interpreting this choice as an inherent ``memorylessness'' requirement for batched bandits in general, and using it to discredit our contributions, would, in fact, contradict how Ruan et al. [1] and Hanna et al. [2] (as well as Yu \& Oh [4]) themselves position RS-OFUL, and would be inconsistent with their own statements and comparisons.
>
> **Allowing observed contexts within a batch (without incorporating rewards) is rather common in the batched bandit literature.**
>
> In many of the existing literature studying a batch policy sequence $\pi = (\pi_1,\pi_2,\dots,\pi_T)$,
> each $\pi_t$ may depend on all contexts observed up to time $t$, while reward information from only previous batches may be used for updates. Examples are as follows:
>
> - Yasin Abbasi-Yadkori, Dávid Pál, and Csaba Szepesvári. Improved algorithms for linear stochastic bandits. Advances in Neural Information Processing Systems, 24:2312–2320, 2011.
> - Zijun Gao, Yanjun Han, Zhimei Ren, and Zhengqing Zhou. Batched multi-armed bandits problem. Advances in Neural Information Processing Systems, 32, 2019.
> - Zhimei Ren, Zhengyuan Zhou, and Jayant R Kalagnanam. Batched learning in generalized linear contextual bandits with general decision sets. IEEE Control Systems Letters, 6:37–42, 2020.
> - Yanjun Han, Zhengqing Zhou, Zhengyuan Zhou, Jose Blanchet, Peter W Glynn, and Yinyu Ye. Sequential batch learning in finite-action linear contextual bandits. arXiv preprint arXiv:2004.06321, 2020.
> - Cem Kalkanli and Ayfer Ozgur. Batched thompson sampling. Advances in Neural Information Processing Systems, 34:29984–29994, 2021.
> - Amin Karbasi, Vahab Mirrokni, and Mohammad Shadravan. Parallelizing thompson sampling. Advances in Neural Information Processing Systems, 34:10535–10548, 2021.
> - Osama Hanna, Lin Yang, and Christina Fragouli. Efficient batched algorithm for contextual linear bandits with large action space via soft elimination. Advances in Neural Information Processing Systems, 36:56772–56783, 2023a.
> - Zhimei Ren and Zhengyuan Zhou. Dynamic batch learning in high-dimensional sparse linear contextual bandits. Management Science, 70(2):1315–1342, 2024.
> - Quanquan Gu, Amin Karbasi, Khashayar Khosravi, Vahab Mirrokni, and Dongruo Zhou. Batched neural bandits. ACM/JMS Journal of Data Science, 1(1):1–18, 2024.
> - Rong Jiang and Cong Ma. Batched nonparametric contextual bandits. IEEE Transactions on Information Theory, 2025.
> - Rong Jiang and Cong Ma. The adaptivity barrier in batched nonparametric bandits: Sharp characterization of the price of unknown margin. arXiv preprint arXiv:2511.03708, 2025.
>
> All of these works are published in top venues in their respective fields and/or come from well-established research groups in the bandit community, and there are many more examples.
> Therefore, sequential dependence on contexts within a batch is neither unusual nor non-standard, but rather part of a well-established and practically motivated line of research.
>
> Both formulations—(i) fixed policy within each batch and (ii) allowing observed contexts within each batch while incorporating only past-batch rewards—are widely adopted because they reduce adaptivity cost, not because batched algorithms are required to be memoryless. Thus, our formulation naturally falls within the limited adaptivity framework.
>
> We also note that Zhang et al. [3], a recent work on batched linear contextual bandits, requires maintaining $\mathcal{O}(d \log T)$ matrices and all previous batch estimators at each time step, whereas our method stores only a single Gram matrix per step together with the batch-wise estimates. In both cases, the algorithms process every context and select actions at each step. Imposing strict memorylessness by ignoring readily available within-batch contexts (while still not incorporating rewards) would only increase computational and memory costs, and  appears arbitrary rather than a fundamental characteristic of the batched framework, given the rich existing literature.

---

> ### Author Response · Authors · 2025-11-21
>
> ---
> ## **Static Grid**
>
> With all due respect, there appears to be a misunderstanding. We are happy to clarify this. The reviewer suggests that our comparison with the “static grid” baselines is invalid. **We would like to clarify that “static grid” refers to the case where the batch *schedule* is determined a priori and fixed throughout learning (i.e., if the timesteps of the batch boundaries are fixed in advance, then such a framework is static grid).** It does *not* refer to a specific internal decision rule, such as whether contexts are used or not within each batch. This is precisely how we use the terminology in our manuscript.
>
> Our algorithms use a **static batch schedule**: batch boundaries are fixed in advance (regardless of whether within-batch contexts are used). To see the distinction, consider the "**adaptive grid**", where batch sizes or boundaries are chosen *adaptively* (i.e., variable batch size/boundaries)  at the beginning of each batch. Our methods do **not** use variable (or adaptive) batch boundaries.
>
> ---
> ## **Using Within-Batch Contexts Does Not Trivialize the Problem**
>
> Finally, we respectfully disagree with the interpretation that our setting “potentially trivializes the problem” simply because it leverages within-batch contexts. **There is currently (prior to our work) no theoretical evidence that using within-batch contexts, under the same batch (limited reward-adaptivity) constraints, makes the batched linear contextual bandit problem trivially easier**, as the reviewer’s comment appears to suggest. In fact, among batched linear contextual bandit algorithms, Abbasi-Yadkori et al.’s RS-OFUL [5] is the existing method that fully exploits within-batch contexts (even with variable/adaptive batch boundaries), and its regret in the batched regime is known to be suboptimal. Thus, prior to our work, it was **unknown** whether one could utilize within-batch contexts while still matching or improving optimal regret under limited adaptivity. **Interpreting the mere use of within-batch contexts as inherently making the problem trivial (as if it would automatically lead to improved guarantees) therefore does not accurately reflect the currently known results in the literature and risks unintentionally discrediting meaningful advances in this line of work.**
>
> **Our results show, for the first time, that designing an *optimal* batched algorithm that leverages within-batch contexts is indeed possible.** We remove the need for G-optimal design, achieve minimax optimality in all (small and large arms) regimes in the contextual setting, and do so with **improved computational efficiency and practically implementable algorithms** (see Figure 1 and Table 3 for guidance on which algorithm one would use in practice). **Our algorithmic designs are non-trivial, our regret guarantees are non-trivial, and our analyses are non-trivial.** We therefore do not believe that our setting or results trivialize the batched contextual bandit problem; rather, they advance the understanding of what is achievable under the well-accepted limited-adaptivity constraints.
>
> ---
> We thank the reviewer for this discussion and the opportunity to clarify these points. We hope these clarifications address the reviewer’s concerns, and we would be happy to provide any further details if needed.
> If our responses addressed the reviewer's comments, we would respectfully ask that the evaluation reflect our contributions.
>
> >[1] Yufei Ruan, Jiaqi Yang, and Yuan Zhou. *Linear bandits with limited adaptivity and learning distributional optimal design.* In Proceedings of the 53rd Annual ACM SIGACT Symposium on Theory of Computing, pp. 74–87, 2021.
> >[2] Osama A Hanna, Lin Yang, and Christina Fragouli. *Contexts can be cheap: Solving stochastic contextual bandits with linear bandit algorithms.* In The Thirty Sixth Annual Conference on Learning Theory, pp. 1791–1821. PMLR, 2023b.
> >[3] Zihan Zhang, Xiangyang Ji, and Yuan Zhou. *Almost optimal batch-regret tradeoff for batch linear contextual bandits.* In The Thirteenth International Conference on Learning Representations, 2025.
> >[4] Sanghoon Yu and Min-hwan Oh. Optimal and practical batched linear bandit algorithm. In Forty-Second International Conference on Machine Learning, 2025.
> >[5] Abbasi-Yadkori, Dávid Pál, and Csaba Szepesvári. Improved algorithms for linear stochastic bandits. Advances in Neural Information Processing Systems, 24:2312–2320, 2011.

---

> ### Comment · Reviewer_mddr · 2025-11-25
>
> Indeed, the paper was comparing with previous literature that assumed a more restrictive definition of batch.

---

> ### Author Response · Authors · 2025-11-26
>
> We are happy to provide further clarification. **To avoid any remaining ambiguity, it is useful to distinguish between (i) the *problem formulation* and (ii) the *algorithmic design choices* made within that formulation.** The remark that previous work “assumed a more restrictive definition of batch” potentially conflates these two levels: some of prior algorithms do not utilize contexts, but this does not mean that the formal definition of the batched / limited-adaptivity setting itself requires such restrictions.
>
> To summarize our position succinctly, we provide a short Q&A, with each answer grounded in the literature discussed above.
>
> ---
> **Q1. Has the previous batched bandit literature defined the setting to be “memoryless about contexts”?**
>
> **A1. No.** To the best of our knowledge, no prior work formally defines batched bandits by requiring the learner to be memoryless with respect to within-batch contexts. On the contrary, algorithms such as RS-OFUL (Abbasi-Yadkori et al. [5]), which are explicitly classified as batched / limited-adaptivity by state-of-the-art works (Ruan et al. [1], Hanna et al. [2], and Yu & Oh [4]), process all observed contexts and maintain a Gram matrix that is updated at every round. By these standards, the problem settings constrain *when* reward feedback can be used to update the policy (at batch boundaries or at rare switch times), not whether within-batch contexts can be stored or used in the decision rule. **Thus, context–memorylessness is not part of the formal problem definition.**
>
> ---
> **Q2. Are our proposed methods batched algorithms?**
>
> **A2. Yes.** Under the standards used in the prior literature, where RS-OFUL is commonly treated as a batched/limited-adaptivity algorithm, our methods are likewise batched algorithms. We fix the parameter $\hat\theta$ within each batch and update it only at pre-specified batch boundaries using rewards from previous batches. In this sense, our adaptivity is *no greater* than that of RS-OFUL (which is recognized as batched / rarely-switching), and in fact we update the parameter minimally (only $O(\log\log T)$ times). **It is therefore consistent and logical, in light of Ruan et al. [1], Hanna et al. [2], and Yu & Oh [4], to regard our methods as batched algorithms.**
>
> ---
> **Q3. Was it previously known that using within-batch contexts is beneficial under limited adaptivity?**
>
> **A3. No, this was not established prior to our work. This point is crucial.** Before our work, Abbasi-Yadkori et al. [5] was the linear contextual batched bandit algorithm that explicitly leveraged within-batch contexts, but it is known to be suboptimal in the batched setting. In contrast, Ruan et al. [1], Hanna et al. [2], and Zhang et al. [3] *deliberately avoid* using within-batch contexts and obtain (near-)optimal regret guarantees using their own techniques. **It therefore remained open whether one could (i) utilize within-batch contexts while still (ii) achieving, or even improving upon, optimal regret guarantees under limited adaptivity**. **This contribution is very important, since previously utilizing within-batch contexts was associated with suboptimal guarantees only.**
>
> Our results show, for the first time, that this is indeed possible: **our algorithms both leverage within-batch contexts and *improve* theoretical guarantees, achieving minimax optimality in both large and small arm regimes simultaneously, while also being more computationally efficient than any of the previous methods.** In this sense, we are operating within the consistent batched framework as prior work, but with a different and previously unexplored algorithmic design that turns out to be provably advantageous and practically superior.
>
> ---
> **Moreover, allowing the learner to utilize readily available context information does not diminish our contributions; rather such design should be rewarded.** Our methods **achieve minimax-optimal regret in both large and small $K$ regimes, remove the reliance on G-optimal design in the linear setting, and provide, to our knowledge, the first $\kappa$-independent algorithm for generalized linear contextual batched bandits.** Given that contexts are already observed at every round, **it is natural—and, in our practical view, desirable—to utilize them to obtain statistically and computationally efficient algorithms**, rather than enforcing context–memorylessness as an additional constraint, which would be contrary to the scope in the vast existing literature.
>
> We believe that some of the confusion as to where to draw the line stems from the fact that the clear distinction between the formal definition of batched bandits and specific algorithmic instantiations has not been discussed in depth in prior work. **We see this exchange as a valuable opportunity to clarify and organize these notions so that future contributions can build on a more explicit and well-understood foundation.** Again, we thank the reviewer for this opportunity.

---

### Official Review · Reviewer_mddr · 2025-11-10

**Soundness:** 3
**Presentation:** 3
**Contribution:** 3
**Rating:** 6
**Confidence:** 2

**Summary:**

This paper studied the batched linear contextual bandit problem.
1. It proposed algorithms for linear bandits that matched the minimax optimal regret bound in both small $K$ and large $K$ regime.
2. The proposed algorithms also achieved the minimum possible number of policy switches, among algorithms with nearly minimax optimal regret bound.
3. The algorithms didn't need explicit construction of G-optimal design, and had improved time complexity compared to baselines.
4. The paper also extended the algorithm to generalized linear contextual bandits.

**Strengths:**

1. The paper improved upon previous paper in $\mathrm{poly}(\log d, \log \log T)$ factors in regret term.
2. The paper utilized Sherman-Morrison formula to compute matrix inverse incrementally and reduced time complexity for the algorithm.

**Weaknesses:**

1. The regret bound improvements compared to baselines are quite small.
2. In light of point 1, it would be ideal if the paper could show that $\sqrt{\log \log T}$ or $\sqrt{\log d}$ dependency is inevitable in (batched) linear bandits.

**Questions:**

1. Line 110, lower bound should use $\Omega$ and I believe it shouldn't be $\tilde \Omega$.
2. Empirical results show that the algorithm with G-optimal design (BLCE-G) is actually inferior compared to BLCE. Can you share some insight on why it's the case?

---

> ### Author Response · Authors · 2025-11-21
>
> We sincerely thank you for taking the time to review our paper and for providing thoughtful and valuable feedback. We greatly appreciate your recognition of our work and your constructive comments. Below, we address each of your comments and questions in detail:
>
> ---
>
> ## **Use of the Sherman-Morrison Formula**
>
> We would like to clarify that although we use the Sherman–Morrison formula to incrementally update the matrix inverse and reduce computational overhead, **our complexity improvement does not rely on this technique**. For a fair comparison, we assume that all baseline algorithms can also apply the same Sherman–Morrison updates. The key source of our computational speedup is the complete removal of the G-optimal design subroutine, which constitutes the dominant bottleneck in prior batched (generalized) linear contextual bandit algorithms.
>
> ---
>
> ## **Lower Bound Notation**
>
> We sincerely thank the reviewer for carefully catching this notational inaccuracy. We fully agree with your suggestion that the lower bound should use the standard $\Omega$ notation rather than $\widetilde\Omega$. We have corrected this in Line 110 of the revised manuscript. We greatly appreciate your attention to detail, which helps improve the clarity and precision of our presentation.
>
> ---
>
> ## **Performance Comparison between BLCE-G and BLCE**
>
> To further compare the empirical performance of BLCE-G and BLCE, we additionally conducted four independent experiments, each corresponding to a different fixed context distribution. The distributions considered were the Student-$t$, Beta, Laplace, and Exponential distributions. The results are now included in Appendix Section D.3.
>
> As shown in Figure 4, BLCE-G and BLCE exhibit remarkably similar regret performance, with neither algorithm consistently dominating the other. Across most distributional settings, BLCE achieves slightly lower regret, whereas in the heavy-tailed Student-$t$ distribution case this difference nearly vanishes and BLCE-G matches or occasionally exceeds the performance of BLCE. While theory guarantees that BLCE-G has a strictly smaller minimax regret bound, these guarantees characterize worst-case regimes and do not necessarily arise in benign stochastic environments. Thus, the empirical results do not contradict the theoretical results.
>
> The empirical relationship between BLCE-G and BLCE is more nuanced. While BLCE-G does not consistently exhibit inferior performance, its behavior can be influenced by numerical considerations. In particular, G-optimal design is known to suffer from numerical instability: because no closed-form solution exists, one must run a Frank–Wolfe–type iterative solver until convergence. The convergence tolerance (typically based on the $\ell_2$-norm) introduces an additional hyperparameter, and in practice, even small inaccuracies in the approximate design can noticeably influence the resulting regret. In contrast, BLCE does not rely on G-optimal design and therefore avoids such numerical issues, which can lead to more stable empirical behavior.
>
> ---

---

> ### Author Response · Authors · 2025-11-21
>
> ---
>
> ## **Improvements of the Regret Bound**
>
> Achieving the minimax-optimal regret in both the large-$K$ and small-$K$ regime is highly non-trivial, and these improvements are very meaningful, about which we are more than happy to elaborate.
> As the reviewer noted, our regret bounds improve the dependence on $\text{poly}(\log d, \log T, \log\log T)$ in both the large-$K$ regime $(K \ge \Omega(e^d))$ and the small-$K$ regime $(K \le \mathcal{O}(e^d))$. To the best of our knowledge, no prior algorithm **simultaneously achieves minimax-optimal regret in both regimes**; thus, no previous work optimally covers the full range of arm sizes $K$. For instance, Zhang et al. [1]—the most recent work on batched linear contextual bandits—obtains regret bounds that are actually worse than earlier results in the small-$K$ regime. Their primary contribution lies in partially reducing the use of G-optimal design to improve practicality. In contrast, our algorithms achieve the tightest regret bounds in both regimes while being strictly more practical; in particular, BLCE completely eliminates the need for any G-optimal design.
>
> We also note that Zhang et al. [1], despite offering improved time complexity relative to previous work, still requires maintaining $\mathcal{O}(d\log T)$ matrices and all previous batch estimators at every round, whereas our method maintains **only a single Gram matrix** per round together with the batch estimators.  Furthermore, prior works (Zhang et al. [1], Ruan et al. [2], Hanna et al. [3,4]) fundamentally rely on G-optimal design and i.i.d.-based expectation inequalities—or reduce the contextual setting to a non-contextual one using i.i.d. assumptions—and therefore cannot relax the i.i.d. assumption, which can be somewhat restrictive in certain settings. In contrast, BLCE, which does not require G-optimal design nor i.i.d.-based inequalities, **can leverage the relaxed i.i.d. assumptions** described in Remark 1.
>
> In addition, prior batched algorithms (Zhang et al. [1], Sawarni et al. [5]) adopt batch schedules that depend on problem-specific parameters beyond the horizon $T$.  This often leads to unstable empirical performance, most notably when the first batch becomes extremely large, forcing the agent to explore blindly for an extended period without receiving any reward information. In contrast, our algorithms introduce a **novel batch schedule** that depends solely on $T$, making the method both theoretically optimal and more practical, and resulting in far more predictable and stable behavior across a wide range of problem instances.
>
> Moreover, several prior works rely on elimination-driven exploration (Zhang et al. [1], Ruan et al. [2], Sawarni et al. [5]). When arm elimination does not proceed effectively, these methods can be pushed into overly exploratory behavior, again causing significant empirical instability. Our algorithms avoid this problem by **explicitly incorporating a greedy within-batch exploitation step**, which prevents excessive exploration and consistently yields strong empirical performance without sacrificing theoretical guarantees.
>
> Taken together—improved regret guarantees across all $K$ regimes, strictly lower computational and memory requirements, the ability to relax context assumptions, and substantially more stable and strong empirical performance without compromising any theoretical guarantees—we believe these contributions go well beyond a minor technical refinement.
>
> In the batched generalized linear contextual bandit setting, our improvements go well beyond logarithmic factors. We **remove all dependence on $\kappa$** and, to the best of our knowledge, provide the first batched algorithm that **matches the minimax-optimal regret**. Notably, even in the fully adaptive generalized linear bandit literature, the most recent work of Zhang et al. [6] still incurs $\kappa$-dependence in transient terms. Despite operating under limited adaptivity, the batched setting benefits from the i.i.d. nature of contexts—a structural advantage absent in fully adaptive (and potentially adversarial) settings—which allows us to obtain a completely $\kappa$-independent regret bound even with weaker adaptivity. Moreover, prior work could not achieve the optimal $\widetilde{\mathcal{O}}(d\sqrt{T})$ regret due to the $(\sqrt{d/\hat\kappa} \wedge \sqrt{R_{\dot\mu}\log d})$ term. Our algorithm BGLE attains the minimax-optimal regret while simultaneously reducing computational cost by removing all reliance on G-optimal design.
>
> Finally, because BGLE extends BLCE, our work provides the first **unified framework** that achieves minimax-optimal regret, $\mathcal{O}(\log\log T)$ adaptivity, and practical computational scalability for both linear and generalized linear contextual bandits.  We believe this framework has the potential to naturally extend to broader classes of batched bandit algorithms, enabling theoretically optimal performance while maintaining practical efficiency.

---

> ### Author Response · Authors · 2025-11-21
>
> >[1] Zihan Zhang, Xiangyang Ji, and Yuan Zhou. *Almost optimal batch-regret tradeoff for batch linear contextual bandits.* In The Thirteenth International Conference on Learning Representations, 2025.
> >[2] Yufei Ruan, Jiaqi Yang, and Yuan Zhou. *Linear bandits with limited adaptivity and learning distributional optimal design.* In Proceedings of the 53rd Annual ACM SIGACT Symposium on Theory of Computing, pp. 74–87, 2021.
> >[3] Osama A Hanna, Lin Yang, and Christina Fragouli. *Contexts can be cheap: Solving stochastic contextual bandits with linear bandit algorithms.* In The Thirty Sixth Annual Conference on Learning Theory, pp. 1791–1821. PMLR, 2023b.
> >[4] Osama Hanna, Lin Yang, and Christina Fragouli. *Efficient batched algorithm for contextual linear bandits with large action space via soft elimination.* Advances in Neural Information Processing Systems, 36:56772–56783, 2023a.
> >[5] Ayush Sawarni, Nirjhar Das, Siddharth Barman, and Gaurav Sinha. *Generalized linear bandits with limited adaptivity.* Advances in Neural Information Processing Systems, 37:8329–8369, 2024.
> >[6] Yu-Jie Zhang, Sheng-An Xu, Peng Zhao, and Masashi Sugiyama. *Generalized linear bandits: Almost optimal regret with one-pass update.* The Thirty-Ninth Annual Conference on Neural Information Processing Systems, 2025.
>
> ---
>
> Please feel free to reach out if you have any further questions!

---

### Meta-Review · Area_Chair_SJJn · 2026-01-07

**Summary:**

This paper studies batched linear and generalized linear contextual bandits under limited adaptivity. The authors propose three algorithms: BLCE-G (Batched Linear Contextual Bandit with Elimination and near G-optimal design), BLCE (same algorithm without the G-optimal design), and BGLE (Batched Generalized Linear Bandit with Elimination). While the submission introduces algorithms that achieve minimax-optimal regret with $O(\log \log T)$ batch complexity and removes the G-optimal design bottleneck, significant concerns regarding the fairness of comparison and novelty relative to existing work remain unresolved. Despite the authors' rebuttals, the consensus during the discussion highlights a fundamental divergence from the standard "batched" framework that potentially simplifies the problem.

**Reviewer Concerns:**

Reviewer rQ4C and Reviewer mddr both pointed out that the paper uses a definition of “batched” that allows for adaptation within batches. This is different from the more standard definitions of batched bandits used in the literature (e.g., Ruan et al., 2021; Zhang et al., 2025). By using within-batch context history, the proposed algorithms are more adaptable than the baselines they claim to outperform. More importantly, batched bandit problems are typically contrasted with fully sequential bandit problems. Many motivating applications of batched bandits (such as those cited in this paper’s introduction, including clinical trials and recommender systems) require parallel or batched execution because sequential experimentation is either costly or infeasible. In the setting studied in this paper, however, even within a single batch, the order of rounds still matters: the decision made at round t depends on the contextual information from all immediately preceding steps. As a result, the algorithm must be executed in a fully sequential manner. This mismatch leads to unfair theoretical and empirical comparisons, as the underlying settings and constraints are fundamentally different. The authors are encouraged to reframe the comparison and admit that the gain should be (at least partially) attributed to the additional historical context and the fully sequential execution style leveraged by the proposed algorithm.

Reviewer mddr also noted that the regret improvements are relatively small. Considering the concerns about the unfair comparison setting, these small gains are not enough to justify the claimed significance of the results. During my reading of the paper, I also noticed that it has large similarities to Ren et al. (2024) in terms of its main contribution (optimal batched linear bandits), algorithm design (elimination-based), and proof structures. The submission does not provide a rigorous comparative discussion or differentiate itself from this highly relevant prior work. This raises doubts about the novelty and contribution of this work.

**Reviewer Scores:**

Reviewer rQ4C: 2 (Maintained rejection based on the inconsistent batch definition).

Reviewer mddr: 6 (Expressed concern that the paper compares against a more restrictive batch definition, which undermines the claimed speedup and regret gains).

Reviewer 4X1Q: 8 (Appreciated the removal of G-optimal design but did not fully address the definitional fairness issue).

Reviewer np75: 6 (Satisfied by responses to their own questions, and changed to 8 in the comment section).

---

### Decision · Program_Chairs · 2026-01-26

Reject